# Streaming Algorithms and Lower Bounds for Estimating Correlation Clustering Cost

**Sepehr Assadi**[1]    **Vihan Shah**[1]    **Chen Wang**[2]

[1]University of Waterloo    [2]Rutgers University

sepehr@assadi.info

vihan.shah@uwaterloo.ca

wc497@cs.rutgers.edu

## Abstract

Correlation clustering is a fundamental optimization problem at the intersection of machine learning and theoretical computer science. Motivated by applications to big data processing, recent years have witnessed a flurry of results on this problem in the streaming model. In this model, the algorithm needs to process the input $n$-vertex graph by making one or few passes over the stream of its edges and using a limited memory, much smaller than the input size.

All previous work on streaming correlation clustering has focused on semi-streaming algorithms with $\Omega(n)$ memory, whereas in this work, we study streaming algorithms with much smaller memory requirements of only polylog$(n)$ bits. This stringent memory requirement is in the same spirit of classical streaming algorithms that instead of recovering a full solution to the problem—which can be prohibitively large with such small memory as is the case in our problem—, aimed to learn certain statistical properties of their inputs. In our case, this translates to determining the "(correlation) clusterability" of input graphs, or more precisely, estimating the cost of the optimal correlation clustering solution.

As our main result, we present two novel algorithms that in only polylog$(n)$ space are able to estimate the optimal correlation clustering cost up to some constant multiplicative factor plus some extra additive error. One of the algorithms outputs a 3-multiplicative approximation plus $o(n^2)$ additive approximation, and the other one further reduces the additive error at the cost of increasing the multiplicative factor to some large constant. We then present new lower bounds that justify the mix of both multiplicative and additive error approximations in our algorithms.

## 1   Introduction

Correlation clustering is a fundamental optimization problem at the intersection of machine learning and theoretical computer science. This problem was introduced by the work of [BBC04][1], with motivation to document clustering, as follows: we have a complete graph $G = (V, E)$ whose edges are labeled by either $(+)$ or $(-)$, and the objective, known as *disagreement minimization*, is to cluster the vertices so that the number of $(+)$ edges across clusters and $(-)$ edges inside the same clusters are minimized. Correlation clustering has since found broad applications in areas such as document categorization [BBC04], webpage segmentation [CKP08], microscopy imaging [ZYH14], and community detection [VGW18, SDE+21], to name a few.

There is an abundant body of literature studying polynomial time algorithms for correlation clustering. [BBC04, CMSY15] showed that there exists a 2.06-approximation algorithm in polynomial time and

37th Conference on Neural Information Processing Systems (NeurIPS 2023).

---

[1]Historically, some forms of the problem were mentioned in earlier works like [GW89].

the problem is NP-hard and even APX-hard. Furthermore, [ACN05] gave a simple (combinatorial) poly-time 3-approximation algorithm that is widely used in practice. Very recently, breakthrough results by [CLN22, CALLN23] achieved 1.73 approximation in polynomial time. In addition, efficient algorithms are also explored for several variants of this problem like the agreement maximization objective [BBC04, Swa04, CGW03], weighted graphs [Swa04, CGW03, EF03], fixed number of clusters [GG06], fair clustering [AEKM20, SZ22], and others.

In recent years, with the rapid development of the 'big data era', there has been a growing interest in algorithms for correlation clustering under *sublinear* models. In general, learning algorithms under sublinear models are able to output the answer without processing or storing the *entire input*. For instance, [BGK13, AW22] gave algorithms to (approximately) learn the clustering in sublinear time, and [CLM$^+$21, BCMT22] designed algorithms in the Massively Parallel Computation (MPC) model. Another widely popular model of sublinear algorithms—the focus of our paper—is the *graph streaming model*. In this model, the edges of the input graph are given to the algorithm one by one in a stream and the memory of the algorithm is desired to be substantially smaller than the input. Here, [ACG$^+$15] designed a streaming algorithm with $\tilde{O}(n)$ memory[2] that achieves 3-approximation with $O(\log \log(n))$ passes – here, $n$ is the number of vertices of the graph and thus the input size is $\Theta(n^2)$ bits. The number of passes of these algorithms were later improved to $O(1)$ by [CLM$^+$21] and a single pass by [AW22], albeit with much larger, yet still a constant, approximation factor. Very recently, [BCMT23] further improved the approximation ratio of single-pass streaming algorithms to a 5-approximation in polynomial time and $(1 + \varepsilon)$-approximation in exponential-time.

The aforementioned line of work in [ACG$^+$15, CLM$^+$21, AW22, BCMT23] focused on the $\widetilde{O}(n)$-memory regime, otherwise known as the *semi-streaming* memory. Allowing for $\Omega(n)$ memory in these algorithms is necessary given that even outputting the solution, namely, the clustering of the input labeled graph, requires this much memory. Yet, in many application, $\Omega(n)$ memory can still be quite large, and the implementations can require significant resource. In such cases, it is highly desirable to determine the "clusterability" of the input graph *before* running the actual clustering algorithm. If even the optimal cost for the input is high, it implies the clustering cannot provide any meaningful outcome, and we should not waste resources on these instances. This type of "value estimation" problem is extensively studied in the streaming literature, see e.g. [KKS15, GT19, ACL$^+$22, BOS22, DKPP22] for several examples in this context. This raises the following fundamental question:

> *How well can we **estimate** the optimal correlation clustering **cost** with* polylog$(n)$-*space streaming algorithms?*

Despite the vast body of work on streaming correlation clustering in general, this question has received almost no attention so far. Indeed, to the best of our knowledge, the only prior work here is that of [ACG$^+$15] who proved that any (finite) *purely multiplicative* factor approximation of the cost is not possible in $o(n)$ space. This result however does *not* rule out any *additive error* approximation (say, in the spirit of [BGK13] for local query algorithms).

We remedy this state of affairs in this paper. Our main algorithmic results show that one can obtain a 3-approximation plus $o(n^2)$ additive approximation to the cost of optimal correlation clustering in only polylog$(n)$ space – the additive approximation can be further reduced at the cost of increasing the multiplicative approximation to some large constant. We then complement these results with new streaming lower bounds that further justify the necessity of the additive errors in our algorithms. Throughout, we will present algorithms in the manner of insertion-only streams. However, we shall note that our algorithms can be extended to *dynamic* streams wherein the edges of the graph can be both inserted and deleted during the stream.

**Our Contributions**

To state our results, we need the following notation. Let OPT be the optimal cost for correlation clustering. We say an algorithm achieves an $(\alpha, \beta)$-approximation of OPT if it gives an $\alpha$-multiplicative with a $\beta$ additive approximation, namely, outputs a number ALG such that

$$\mathsf{OPT} \leq \mathsf{ALG} \leq \alpha \cdot \mathsf{OPT} + \beta.$$

---

[2]Here, and throughout, we use $\tilde{O}(\cdot)$ to hide polylog terms on the parameters.

Our first main result is a single-pass streaming algorithm that achieves an $(O(1), \delta n^2)$-approximation with high probability in $\text{poly}(\log n, 1/\delta)$ space.

> **Result 1.** There is a single-pass streaming algorithm that outputs an $(O(1), \delta n^2)$-approximation of the optimal correlation clustering cost with high probability[a] and uses $O(\text{polylog}(n)/\delta^5)$ space.
>
> ――――――――――
> [a]Here, and throughout, with high probability means with probability at least $1 - 1/n$.

The $\delta^{-5}$-dependence of our algorithm ensures that by setting $\delta = n^{-0.19}$ we can reduce the additive error down to $n^{1.81} = o(n^2)$ and still achieve an $o(n)$-space algorithm. On the flip side however, the leading constant multiplicative factor is quite large in this algorithm – as we will see shortly, the algorithm is built on the sparse-dense decomposition idea from [AW22], which inevitably incurs a (worst-case) constant of at least $10^7$. Our next result addresses this drawback: we present another algorithm that achieves a $(3, \delta n^2)$-approximation in expectation with polylog$n$ space at the cost of an exponential dependence on $\delta$ in the space.

> **Result 2.** There is a single-pass streaming algorithm that outputs a $(3, \delta n^2)$-approximation of the optimal correlation clustering cost in expectation and uses $2^{O(1/\delta)} \cdot \text{polylog}(n)$ space.

Compared to the algorithm in Result 1, the algorithm in Result 2 is more suitable for instances whose optimal correlation clustering costs are large (e.g. $\mathsf{OPT} = \Omega(n^2)$). In such a case, we can pick $\delta$ to be a small constant, and achieve a $(3 + \delta)$-multiplicative approximation with $\text{polylog}(n)$ space.

Since both of our upper bounds contain the $\delta n^2$ additive error, we would naturally wonder to what extent these additive terms are necessary. We present two new lower bounds that partially address this question, and show that the additive errors are necessary to a large extent. Our first lower bound shows that if *only* additive error is allowed, there is no $\text{polylog}(n)$-space streaming algorithm in a single pass achieves additive error substantially better than $O(n^2)$.

> **Result 3.** No single-pass streaming algorithm with $\text{polylog}(n)$ space can output a $(1, n^{2-\varepsilon})$ approximation of the optimal correlation clustering cost with a sufficiently large constant probability of success for some $\varepsilon = o(1)$.

Result 3 can also be interpreted as the *additive* counterpart of the lower bounds in [ACG+15], which focus instead on the purely multiplicative approximation. Since our upper bound allows both approximations, we further provide a second lower bound showing that a $\Theta(n)$ additive error is necessary even if both multiplicative and additive errors are allowed.

> **Result 4.** No single-pass streaming algorithm with even $o(\sqrt{n})$ space can output a $(1.19, O(n))$ approximation of the optimal correlation clustering cost with a sufficiently large constant probability of success.

We now discuss the applicable scenarios for our algorithms. To test instances with low vs. high correlation clustering costs (e.g. $o(n^2)$ vs. $\Omega(n^2)$), it suffices to run our algorithm in Result 1 with $\delta = 1/\log(n)$ which uses $\widetilde{O}(1)$ space. On the other hand, to separate instances that are 'partially clusterable' (e.g. optimal cost of $n^2/1000$) vs. instances that are 'not clusterable at all' (e.g. optimal cost of $n^2/5$), it suffices to run our algorithm in Result 2 with $\delta = O(1)$ a small constant which uses $\widetilde{O}(1)$ space. The only case our algorithms are not able to deal with is to separate multiple 'well-clusterable' instances which does not seem that motivated in practice either.

**Experiments**   To further validate our algorithms, we conduct experiments of our algorithms with the stochastic block model (SBM) extensively studied in the literature (e.g., [HLL83, ABH16, ZT23, Abb17a]). For correlation clustering, we can use a variate of the model that plants clusters with sizes $\Omega(n)$ and samples $(+)$ edges between the vertices in the same cluster with a large probability and $(-)$ edges with a low probability, and vice versa. Our implementation on the simulations of graph streams from the Stochastic Block Model shows that our algorithms consistently obtain approximations within a factor of 3 for the optimal clustering cost, while only storing $0.04\% \sim 3.6\%$ [3] total edges when the graphs is moderately large. Furthermore, our algorithms are able to distinguish instances that are "well-cluserable" vs. "badly-clusterable" using a very small memory.

――――――――――
[3]$0.04\%$ fraction of edges are obtained by implementing the algorithm of Result 2 in a two-pass manner

## 2 Preliminaries

We now introduce the notation and the problem definition. For standard technical tools, please see the supplementary material.

**Notation.** We use $G = (V, E^+ \cup E^-)$ to denote a labeled complete graph arriving in a stream. For any vertex $v$, we denote by $N^+(v)$ all vertices that have an $(+)$ edge from $v$, and we let $N^+[v] = \{v\} \cup N^+(v)$. Similarly, we denote by $N^-(v)$ all vertices that have an $(-)$ edge from $v$, and we let $N^-[v] = \{v\} \cup N^-(v)$. We use $N^+(u) \triangle N^+(v)$ (resp. $N^-(u) \triangle N^-(v)$) to denote the disjoint neighborhoods of $u$ and $v$, i.e. $N^+(u) \triangle N^+(v) = (N^+(u) \cup N^+(v)) \setminus (N^+(u) \cap N^+(v))$. For a fixed set of vertices $A \subseteq V$, we further let $E^+(v, A)$ be the set of $(+)$ edges between $v$ and vertices in $A$, and $E^-(v, A)$ be the set of $(-)$ edges between $v$ and vertices in $A$.

We use $G^+ = (V, E^+)$ to denote the positive subgraph of $G$ with all the $(+)$ edges, and we define $G^- = (V, E^-)$ analogously. Note that since we work with labelled complete graph, the information of $G^-$ edges can be uniquely inferred from the positive subgraph $G^+$. As such, when we work with $G^+$ only, we call a $(+)$ edge $(u, v) \in E^+$ as an *edge in $G^+$*. Similarly, we call a $(-)$ edge $(u, v) \in E^-$ an *non-edge in $G^+$*. We may omit $G^+$ when the context is clear.

For a fixed cluster $\mathcal{C}$ on a labeled complete graph $G$, we use $\mathsf{cost}(\mathcal{C})$ to denote the total cost of correlation clustering on $G$ by $\mathcal{C}$ (we omit $G$ since it is obvious by the context). Furthermore, we slightly abuse the notation to reload $\mathsf{cost}(\cdot)$ as a function of cost on subgraphs of $G$ in the following occasions: 1) For an induced subgraph $H \subseteq G$, we let $\mathsf{cost}(\mathcal{C}, H)$ be the cost of $\mathcal{C}$ induced by the edges whose both endpoints are in $H$; 2) For two vertex sets $A \cup B \subseteq V$, we let $\mathsf{cost}(\mathcal{C}, (A, B))$ be the cost of $\mathcal{C}$ induced by the edges with exactly one endpoint in $A$ and one endpoint in $B$.

Finally, for a single edge $(u, v)$, we use the notation $\mathsf{edge\text{-}cost}(\mathcal{C}, (u, v))$ to denote the cost induced by a single edge $(u, v)$ for clustering $\mathcal{C}$. We may write a short-hand notation $\mathsf{edge\text{-}cost}((u, v))$ when the context is clear that $\mathcal{C}$ is used.

**Problem Definition**

We now give the formal description of the problem.

**Problem 1** (Correlation Clustering Value Estimation)**.** Given a labeled complete graph $G = (V, E^- \cup E^+)$ and a clustering $\mathcal{C}$ that partitions $V$ into disjoint set of vertices $C_1, C_2, \cdots, C_k$, the cost of disagreement minimization correlation clustering on $\mathcal{C}$ is defined as

$$\mathsf{cost}(\mathcal{C}) := \left|\{(u, v) \in E^+ \mid \exists i \neq j \ s.t. \ u \in C_i, v \in C_j\}\right| + \left|\{(u, v) \in E^- \mid \exists i \ s.t. \ u, v \in C_i\}\right|$$

Let $\mathsf{OPT}$ be the minimum $\mathsf{cost}$ over all possible clusterings. The Correlation Clustering Value Estimation problem asks for a number $\mathsf{ALG}$ such that $\mathsf{OPT} \leq \mathsf{ALG} \leq f(\mathsf{OPT})$ for some function $f(x) \geq x$. If $f(\mathsf{OPT}) = \alpha \cdot \mathsf{OPT} + \beta \cdot n^2$ for $\alpha$ and $\beta$, we say that $\mathsf{ALG}$ is a $(\alpha, \beta)$-approximation of the value of correlation clustering in this scenario.

We study algorithms for correlation clustering value estimation under the *graph streaming* model, where edges arrive one after another in a stream with the labels. We give a more formal definition of the streaming model (and the more general dynamic streaming model) in the supplementary material.

## 3 An Algorithm based on Sparse-dense Decomposition

In this section, we present our first streaming algorithm that achieves a $(O(1), \delta n^2)$-approximation for testing the value of correlation clustering in $O(\mathrm{polylog}(n)/\mathrm{poly}(\delta))$ space, as long as $\delta \geq 1/\mathrm{polylog}(n)$. We utilize the idea in [AW22] to test a $(O(1), \delta n^2)$-approximate value of the sparse-dense decomposition-based correlation clustering cost, which in turn is an $O(1)$ approximation of the optimal cost. Our algorithm uses a memory of $O(\mathrm{polylog}(n)/\mathrm{poly}(\delta))$ bits, which is efficient for large-scale inputs. More formally, the guarantee of our algorithm is as follows.

**Theorem 1.** *There is a (dynamic) streaming algorithm that with high probability gives a $(O(1), \delta n^2)$-approximation for the correlation clustering value and take space $O\left(\frac{\log^2(n)}{\delta^5}\right)$ words.*

On a high level, our algorithm for Theorem 1 uses the idea to approximate the value of optimal correlation clustering with sparse-dense decomposition (see supplementary material). It is shown in

[AW22] that once we can find such a decomposition on $G^+$, we can achieve an $O(1)$-approximation by simply putting every sparse vertex into a singleton cluster and gathering each almost-clique $K_i$ in a separate cluster. Therefore, an approximation of the number of the aforementioned edges in a *fixed* decomposition will result in a good estimation of the cost.

However, since the sparse-dense decomposition is not unique, it is unclear how to estimate the edges for a fixed decomposition. On the other hand, an algorithm with polylog($n$) space is necessarily oblivious of the decomposition since it takes $\Theta(n)$ bits to write it down. To overcome the problem, we forgo the strict notion of the sparse-dense decomposition, and utilize the notion of $\varepsilon$-*sparse edges* and $\varepsilon$-*dense non-edges* (of $G^+$) instead. We now formally define these notions as follows.

**Definition 1.** Fix an arbitrary graph $G = (V, E)$ (which is *not* necessarily labeled and complete) and a vertex pair $(u, v) \in G$, we say

1. $(u, v)$ is an $\varepsilon$-sparse edge (resp. non-edge) if $(u, v) \in E$ (resp. $(u, v) \notin E$) and $|N(v) \triangle N(u)| \geq \varepsilon \cdot \max\{\deg(u), \deg(v)\}$.

2. $(u, v)$ is an $\varepsilon$-dense edge (resp. non-edge) if $(u, v) \in E$ (resp. $(u, v) \notin E$) and $|N(v) \triangle N(u)| \leq \varepsilon \cdot \max\{\deg(u), \deg(v)\}$.

Note that the definitions of the $\varepsilon$-sparse and $\varepsilon$-dense edges/non-edges are generic and *not* restricted to the labeled complete graph (or even the correlation clustering application). Similarly, the sketching tools we design in this section are also generic: we will use them in the context of correlation clustering later. We now prove the existence of the following tools.

**Lemma 3.1.** *There exists a dynamic streaming algorithm* **Tool-spr** *with parameters* $\varepsilon, \delta$ *that given any graph* $G = (V, E)$ *in a stream and a pair of vertices* $u, v \in V$, *satisfying the promise* $\deg^+(u) \geq \delta n$ *and* $\deg^+(v) \geq \delta n$, *with high probability outputs 'Yes" if* $(u, v)$ *is at least* $\varepsilon$-*sparse and "No" if* $(u, v)$ *is not* $\frac{\varepsilon}{8}$-*sparse using* $O(1/\varepsilon^2 \delta)$ *words of space.*

**Lemma 3.2.** *There exists a dynamic streaming algorithm* **Tool-dns** *with parameters* $\varepsilon, \delta$ *that given any graph* $G = (V, E)$ *in a stream and a pair of vertices* $u, v \in V$, *satisfying the promise* $\deg^+(u) \geq \delta n$ *and* $\deg^+(v) \geq \delta n$, *with high probability outputs 'Yes" if* $(u, v)$ *is at most* $\varepsilon$-*dense and "No" if* $(u, v)$ *is not* $8\varepsilon$-*dense using* $O(1/\varepsilon^2 \delta)$ *words of space.*

We defer the proofs of Lemmas 3.1 and 3.2 to the supplementary material, and use them as blackboxes in the rest of the paper. We now discuss how to use the $\varepsilon$-sparse edges and $\varepsilon$-dense non-edges (defined in Definition 1) in $G^+$ to estimate the correlation clustering cost. In what follows, we use $E^+_{\varepsilon\text{-sparse}}$ to denote the set of $\varepsilon$-sparse (positive) edges, and $E^-_{\varepsilon\text{-dense}}$ to denote the set of $\varepsilon$-dense non-edges. Furthermore, for each vertex $v$, we use $E^-_{\varepsilon\text{-dense}, v}$ to denote the set of $\varepsilon$-dense non-edges incident on $v$. We let $m^+_{\varepsilon\text{-sparse}} := \left| E^+_{\varepsilon\text{-sparse}} \right|$ denote the number of $\varepsilon$-sparse edges, $m^-_{\varepsilon\text{-dense}} := \left| E^-_{\varepsilon\text{-dense}} \right|$ denote the number of $\varepsilon$-dense non-edges and let $m^-_{\varepsilon\text{-dense}, v} := \left| E^-_{\varepsilon\text{-dense}, v} \right|$ denote the number of $\varepsilon$-dense non-edges incident on $v$. Finally, we define $\widehat{m}^-_{\varepsilon\text{-dense}}$ as follows.

**Definition 2.** For each vertex $v$ define $\widehat{m}^-_{\varepsilon\text{-dense}, v} := \min\{m^-_{\varepsilon\text{-dense}, v}, \deg^+(v)\}$. Furthermore, let $\widehat{m}^-_{\varepsilon\text{-dense}} := \sum_{v \in V} \widehat{m}^-_{\varepsilon\text{-dense}, v}$.

The intuition behind $\widehat{m}^-_{\varepsilon\text{-dense}, v}$ is to count the non-edges (of $G^+$) in $E^-_{\varepsilon\text{-dense}, v}$ for *at most* $\deg^+(v)$ *times*. Our estimator for $\varepsilon$-dense non-edges will estimate $\widehat{m}^-_{\varepsilon\text{-dense}}$ instead of $m^-_{\varepsilon\text{-dense}}$ for the following reason: it suffices to estimate $\widehat{m}^-_{\varepsilon\text{-dense}}$ since the number of non-edges inside each almost-clique is at most $\deg^+(v)$; on the other hand, if we estimate *all* $\varepsilon$-dense non-edges, there could be a very large overhead since the non-edges *between* almost-cliques are also counted. We note that if $\widehat{m}^-_{\varepsilon\text{-dense}, v} = m^-_{\varepsilon\text{-dense}, v}$ for all $v \in V$ then $\widehat{m}^-_{\varepsilon\text{-dense}}$ is twice $m^-_{\varepsilon\text{-dense}}$ (since we are double counting edges). But this is a 2-approximation in the worst case, and we do this to make calculations easier.

We prove the following lemmas that establish the connections between the aforementioned sets of edges and the edges from the sparse-dense decomposition. Due to space limits, we directly present the properties of our estimators, and defer the proof to the supplementary material.

**Lemma 3.3.** *Suppose* $G = (V, E)$ *is any labeled graph and* $V = V_{sparse} \sqcup K_1 \sqcup \ldots \sqcup K_k$ *is an* $\varepsilon$-*sparse-dense decomposition of* $G^+$ *for* $0 \leq \varepsilon \leq 1/360$ *and* $\eta_0 \leq 1/20$. *Let* $C_{SDD}$ *be the cost of*

*correlation clustering when the clusters are the almost cliques and sparse vertices are in singleton clusters. Then we have* $\frac{2}{\eta_0 \cdot \varepsilon} \cdot m^+_{\eta_0 \varepsilon\text{-sparse}} + m^+_{\varepsilon\text{-sparse}} + \widehat{m}^-_{4\varepsilon\text{-dense}} \geq C_{SDD} \geq \mathsf{OPT}$.

We upper bound the cost of $m^+_{\varepsilon\text{-sparse}}$ and $\widehat{m}^-_{\varepsilon\text{-dense}}$ by a function of $\mathsf{OPT}$ using a charging argument.

**Lemma 3.4.** *Suppose $G = (V, E)$ is any labeled graph and $\mathsf{OPT}$ be the optimal correlation clustering cost, and let $\varepsilon \leq \frac{1}{32}$ and $\beta \leq \frac{1}{2\varepsilon}$, then $m^+_{\beta\varepsilon\text{-sparse}} \leq \frac{2}{\beta\varepsilon} \cdot \mathsf{OPT}$ and $\widehat{m}^-_{\beta\varepsilon\text{-dense}} \leq 8 \cdot \mathsf{OPT}$.*

By Lemmas 3.3 and 3.4 if we are able to exactly recover $m^+_{\varepsilon\text{-sparse}}$ and $\widehat{m}^-_{\varepsilon\text{-dense}}$ then we would get an $O(1)$ approximation of $\mathsf{OPT}$. Such a task seems difficult, however, using Lemmas 3.1 and 3.2 we design our tools for estimating $m^+_{\varepsilon\text{-sparse}}$ and $\widehat{m}^-_{\varepsilon\text{-dense}}$ with an additive error.

**Lemma 3.5.** *There exists a dynamic streaming algorithm with parameters $\varepsilon, \delta$ that given any graph $G = (V, E)$ in a stream returns a value $Z_{sp}$ that is at least $m^+_{\varepsilon\text{-sparse}}$ and at most $m^+_{\varepsilon/8\text{-sparse}} + \delta n^2$ with high probability and uses $O(\log n/\varepsilon^2 \delta^3)$ words of space.*

**Lemma 3.6.** *There exists a dynamic streaming algorithm with parameters $\varepsilon, \delta$ that given any graph $G = (V, E)$ in a stream returns a value $Z_{den}$ that is at least $\widehat{m}^-_{\varepsilon\text{-dense}}$ and at most $\widehat{m}^-_{8\varepsilon\text{-dense}} + \delta n^2$ with high probability and uses $O(\log^2 n/\varepsilon^2 \delta^5)$ words of space.*

**Finalizing the proof sketch of Theorem 1.** We are given a parameter $\delta$ as input and we want the additive error to be at most $\delta n^2$. We fix $\varepsilon = 1/360$, $\eta_0 = 1/20$ and $\delta' = \delta \cdot (2 + \frac{2}{\eta_0 \varepsilon})^{-1}$.

Run the algorithm in Lemma 3.5 with parameters $\varepsilon$ and $\delta'$, and let the output be $Z^\varepsilon_{sp}$. Also, run the algorithm in Lemma 3.5 with parameters $\eta_0 \varepsilon$ and $\delta'$, and let the output be $Z^{\eta_0 \varepsilon}_{sp}$. Run the algorithm in Lemma 3.6 with parameters $4\varepsilon$ and $\delta'$, and let the output be $Z_{den}$.

Lemmas 3.3 to 3.6 imply that the cost $Z_{CC}$ we output satisfies:

$$\mathsf{OPT} \leq Z_{CC} := Z^\varepsilon_{sp} + (2/\eta_0 \varepsilon) \cdot Z^{\eta_0 \varepsilon}_{sp} + Z_{den} \leq O(1) \cdot \mathsf{OPT} + \delta n^2.$$

The space taken is $O(\log n/\delta^3)$ for both copies of Lemma 3.5 and $O(\log^2 n/\delta^5)$ for Lemma 3.6 giving a total space of $O(\log^2 n/\delta^5)$ words. This proves Theorem 1.

## 4 An Algorithm based on Pivot

In this section, we give our second streaming algorithm that is a $(3 + \gamma, \delta n^2)$-approximation for the correlation clustering value for any choice of $\delta$ and $\gamma < 1/2$. The algorithm works in $O(\text{polylog}(n))$ space as long as $\delta \geq \Omega(\frac{1}{\log\log(n)})$. Consider the following formal statement:

**Theorem 2.** *There is a (dynamic) streaming algorithm that with high probability gives a $(3 + \gamma, \delta n^2)$-approximation for the correlation clustering value and takes space $O\left(\frac{2^{7/6\delta} \cdot \log^2 n}{\gamma \cdot \delta^5}\right)$ words.*

Our algorithm in Theorem 2 is inspired by the Local Cluster algorithm from [BGK13]. The Local Cluster algorithm samples $1/\delta$ random vertices in a set $U$ and computes the greedy maximal independent set (MIS) $M$ of $U$ to get the cluster centers $p_1, p_2, \ldots, p_t$. The clusters generated then are $N^+[p_i] - \cup_{j=1}^{i-1} N^+[p_j]$ and all the remaining vertices (called unclustered vertices) are clustered in their own singleton cluster. [BGK13] proved that the expected cost of this clustering is at most $3 \cdot \mathsf{OPT} + \frac{\delta}{2} n^2$. More formally, they showed the following:

**Proposition 4.1** ([BGK13]). *The expected cost of Local Cluster is at most $3 \cdot \mathsf{OPT} + \frac{\delta}{2} n^2$.*

To simulate the Local Cluster algorithm in a streaming manner, while forming a cluster we need to know which of its neighbors are already clustered in previous clusters which requires new sketching tools (on $G^+$) that are different from the ones we used in Section 3. These tools estimate the number of non-edges **within** or the number of edges **going out of** the neighborhood of a vertex ($u$) outside of the ($+$) neighborhood of a known set of vertices ($S$). In other words, fix a graph $G = (V, E)$, a vertex $u$, and a set $S$, we want to estimate the number of non-edges within or the number of edges going out of $N[u] - N[S]$. We again use the generic form to present the sketching tools and do not specify ($+$) edges. The formal definitions can be given as follows.

**Definition 3.** A non-edge $(x, y)$ is **within** $N[u] - N[S]$ iff: $i)$ $x \notin N[S]$ and $ii)$ $y \notin N[S]$ and $iii)$ $x \in N[u]$ and $y \in N[u]$.

**Definition 4.** An edge $(x, y)$ is **going out of** $N[u] - N[S]$ iff $i)$ $x \notin N[S]$ and $ii)$ $y \notin N[S]$ and $iii)$ $x \in N[u]$ or $y \in N[u]$ but not both.

**Definition 5.** An edge $(x, y)$ is **unclustered** w.r.t. $S$ iff $i)$ $x \notin N[S]$ and $ii)$ $y \notin N[S]$.

We also define $m_{ne}(u, S)$ as the number of non-edges **within** $N[u] - N[S]$, $m_e(u, S)$ as the number of edges **going out of** $N[u] - N[S]$, and $m_u(S)$ as the number of **unclustered** edges w.r.t. $S$. We write $m_{ne}, m_e$, and $m_u$ when $u$ and $S$ are clear from context.

Exactly recovering $m_u, m_{ne}$ and $m_e$ is difficult, so we have the following lemma about estimating the number of non-edges **within** $N[u] - N[S]$, the edges **going out of** $N[u] - N[S]$, and the **unclustered** edges with respect to $S$:

**Lemma 4.2.** *There exist streaming algorithms called* NE-Tool, E-Tool, *and* U-Tool *that given $u$ and $S$ before the stream, respectively compute with high probability $i)$. the number of non-edges **within**; $ii)$. the number of edges **going out of** $N[u] - N[S]$; and $iii)$. the number of edges with **unclustered** w.r.t. $S$ with an overestimation of at most $\delta n^2$ and take space $O(1/\delta^2)$ words.*

Using the tools we have built so far, we present a $(3, \delta n^2)$ approximation algorithm in expectation.

---

**Algorithm 1.** Simulation of the Local Cluster algorithm
**Input:** $G = (V, E^+ \cup E^-)$ in a (dynamic) stream
**Output:** $(3, \delta n^2)$-approximation to the correlation clustering value in expectation
**Pre-Processing:**

1. Sample a set $U$ of $1/\delta$ random vertices
2. Let $\pi$ be a random permutation of the vertices in $U$

**During the Stream:**

1. Store $(+)$ all edges between vertices of $U$
2. For all $u \in U$ and $S \subseteq U$ compute NE-Tool$(u, S)$, E-Tool$(u, S)$, U-Tool$(S)$ with parameter $\delta^2/6$ using the $G^+$ subgraph.

**Post-Processing:**

1. Compute the greedy MIS $M := p_1, p_2, \ldots, p_t$ of $U$ in the order of $\pi$ (using edges stored between vertices of $U$).
2. $S_0 = \emptyset$. $\tilde{Z} = 0$.
3. For $i = 1$ to $t$:
   - $\tilde{Z} = \tilde{Z} + $ NE-Tool$(p_i, S_{i-1}) + $ E-Tool$(p_i, S_{i-1})$.
   - $S_i := S_{i-1} \cup \{p_i\}$
4. Output $Z := \tilde{Z} + $ U-Tool$(M)$

---

Algorithm 1 estimates $m_u, m_{ne}$ and $m_e$ with at most $\delta n^2$ additive error. We now show that with the exact values of $m_u, m_{ne}$ and $m_e$, we get the exact clustering value of the Local Cluster algorithm.

**Claim 4.3.** *The clustering value of Local Cluster is equal to $m_u + \sum_i m_{ne}(p_i, S_{i-1}) + m_e(p_i, S_{i-1})$.*

Combining Lemma 4.2 and Claim 4.3 gives us the following performance guarantees for Algorithm 1.

**Lemma 4.4.** *The output $Z$ of Algorithm 1 is at least* OPT *with high probability and is at most* $3$OPT $+ \delta n^2$ *in expectation.*

*Proof.* Using Claim 4.3 we know that if the tools worked with no error, Algorithm 1 would give the exact clustering cost of Local Cluster. Also, we know that the tools do not underestimate and overestimate by $\delta^2 n^2/6$ with high probability (Lemma 4.2). We first note that $Z$ is at least OPT because of the above conditioning and the fact that the clustering cost of Local Cluster is at least OPT. Using Proposition 4.1 we know that the expected cost of the clustering when choosing $\frac{1}{\delta}$ random pivots is at most $3$OPT $+ \frac{\delta}{2} n^2$. Therefore, the clustering cost of Algorithm 1 is between OPT and $3$OPT $+ \frac{\delta}{2} n^2$ plus the overestimate.

We now calculate the overestimate. We use parameter $\delta^2/6$ for the tools thus the additive error in each tool is at most $\delta^2 n^2/6$. There are $1/\delta$ pivots implying a total additive error of at most $\delta n^2/3$ over all the copies of NE-Tool and E-Tool. U-Tool has an error of at most $\delta^2 n^2/6$ implying that the overall error is at most $\delta n^2/2 + \delta n^2/3 + \delta^2 n^2/6 \leq \delta n^2$ giving a $(3, \delta n^2)$ approximation in expectation.

We note that we condition on the high probability events for all copies of the tools and union bound over the failure probabilities. Thus, for the overall failure probability to be small we need $2^{1/\delta} \leq \text{poly}(n)$. This requirement is trivially statisfied by the fact that $2^{1/\delta}$ is on the space bound (Lemma 4.5), and the space bound for any streaming algorithm is at most $O(n^2)$. $\qquad\square$

We now show the space bound of Algorithm 1.

**Lemma 4.5.** *The space of Algorithm 1 is* $O\left(\frac{2^{1/\delta} \cdot \log n}{\delta^5}\right)$ *words.*

*Proof.* Each copy of the tools with parameter $\delta^2/6$ takes $O(\log n/\delta^4)$ words of space, and we compute the tools for all $v \in U$ and $S \subseteq U$. Thus, the space used is $O(2^{1/\delta} \cdot \log n/\delta^5)$ words. Storing edges between vertices in $U$ takes space at most $1/\delta^2$ words which is a lower order term. $\quad\square$

Lemmas 4.4 and 4.5 together give us a $(3, \delta n^2)$ approximation in expectation using $O\left(\frac{2^{1/\delta} \cdot \log n}{\delta^5}\right)$ space. We now prove Theorem 2. We run Algorithm 1 $\frac{60 \log n}{\gamma}$ times in parallel and let $Z_{min}$ be the minimum cost over all iterations. $Z_{min}$ is a $(3 + \gamma, \frac{7}{6}\delta n^2)$-approximation with high probability.

**Claim 4.6.** $\mathsf{OPT} \leq Z_{min} \leq (3 + \gamma)\mathsf{OPT} + \frac{7}{6}\delta n^2$ *with high probability.*

Thus, we get a $(3 + \gamma, \frac{7}{6}\delta n^2)$-approximation with high probability. Each parallel repetition of the algorithm takes $O\left(\frac{2^{1/\delta} \cdot \log n}{\delta^5}\right)$ words of space. Repeating $O(\frac{\log n}{\gamma})$ times and re-scaling $\delta$ by a factor of $\frac{7}{6}$ gives a total space bound of $O\left(\frac{2^{7/6\delta} \cdot \log^2 n}{\gamma \cdot \delta^5}\right)$ words. This completes the proof of Theorem 2.

## 5   A Lower Bound for $O(n^{2-\varepsilon})$ Additive error

In this section, we show that if we *only* allow additive error, any streaming algorithm with polylogarithm memory cannot cross an error barrier of $\Omega(n^{2-\varepsilon})$ for some $\varepsilon = o(1)$. Here, and throughout, we will refer this lower bound as the *almost-quadratic lower bound*. The lower bound is weaker than the linear lower bound of Section 6 in terms of the multiplicative factor since it only works for $c = 1$. However, it is much stronger in the additive sense: the upper bounds obtained by our algorithms are $O(n^2)$, and the almost-quadratic lower bound matches this term up to an $O(n^\varepsilon)$ factor – this provides a strong justification of the additive error in our algorithms.

The formal statement of the almost-quadratic lower bound is as follows.

**Theorem 3.** *There exists a constant $C$, such that any single-pass streaming algorithm that estimates the optimal value $\mathsf{OPT}$ of correlation clustering by a $C \cdot n^{2-\varepsilon}$ purely additive error (i.e., an estimated value that is at most $\left(\mathsf{OPT} + C \cdot n^{2-\varepsilon}\right)$) with probability at least $\frac{99}{100}$ has to use a memory of $\Omega\left(n^\varepsilon\right)$ bits, even on labeled complete graphs.*

Note that Theorem 3 does *not* require the stream to be dynamic, which is in contrast to our upper bound results that work for dynamic streams. We obtain the almost-quadratic lower bound by a new reduction from the INDEX problem. On the high level, the instance we construct 'hides' an $\Omega(n^{2-\varepsilon})$ gap between the yes and no cases inside a case-invariant $\Omega\left(n^2\right)$ cost. The reduction can be viewed as a more involved variant of the space lower bound for the *exact* correlation clustering in a very recent work [AAD$^+$23]. In a nutshell, we modify their construction to 'boost' the gap between yes and no cases, and apply a new trick to separate the *values* of clustering. Due to space limits, we defer the proof of Theorem 3 to the supplementary material.

# 6 A Lower Bound for $O(n)$ Additive error

In this section, we show that any dynamic streaming algorithm that gets a $(c, \varepsilon n)$-approximation for $c < 1.2$ and $O(n)$ additive error needs $\Omega\left(\sqrt{n}\right)$ bits of space. Here, and throughout, we will call this lower bound the *linear lower bound*. Formally, we have:

**Theorem 4.** *Let $c \in [1, \frac{6}{5})$ and $\varepsilon \in (0, \frac{6}{5} - c)$. Any single-pass streaming algorithm that estimates the optimal value* $\mathsf{OPT}$ *of correlation clustering by a $(c, \varepsilon n)$-approximation (i.e., an estimated value that is at most $(c \cdot \mathsf{OPT} + \varepsilon \cdot n)$) with probability at least $\frac{99}{100}$ has to use a memory of $\Omega\left(\sqrt{n}\right)$ bits, even on labeled (complete) graphs.*

Similar to the case in Theorem 3, Theorem 4 does *not* require the stream to be dynamic. Comparing with the *almost-quadratic lower bound* we show in Section 5, the lower bound in Theorem 4 is weaker in the additive sense. However, it allows the multiplicative approximation of the algorithm to be $> 1$, while the lower bound in Section 5 only rules out algorithms with purely additive errors.

Our lower bound uses the celebrated machinery from Boolean Hidden Hypermatching (BHH) and Gap Cycle Counting (GCC) pioneered by [VY11]. The Gap Cycle Counting (GCC) lower bound states that for any algorithm to distinguish whether a graph consists of cycles with length $2t$ or cycles with length $4t$ for some $t \geq 2$, a memory of $\Omega\left(n^{1-\frac{1}{t}}\right)$ bits is necessary. On the high level, our plan is to show that for graphs similar to the ones prescribed in the GCC problem, the values of correlation clustering are different by an additive gap of $O(n)$. Therefore, by a reduction argument, any algorithm that breaks this barrier of additive gap requires $\omega\left(\text{polylog}(n)\right)$ memory.

Due to space limits, we defer the lower bound construction to the supplementary material.

# 7 Experiments

We describe in this section the experiments of algorithms on the Stochastic Block Model and the Erdos-Renyi random graphs. These experiments show that for a very natural family of graphs, our algorithms can achieve a very competitive performance on the estimated cost values, and the performances are often much better than the worst-case theoretical analysis. Furthermore, our algorithms are capable of separating "well-clusterable" vs. "badly-clusterable" instances.

**Experiment Settings** Limited by space, we sketch the key settings of our experiment, and defer the full discussion to the full version in the supplementary material. As we have discussed, we perform our experiments on the data generated from the well-studied Stochastic Block Model (SBM) that plants ground-truth clusters with sizes $\Omega(n)$, samples $(+)$ edge between vertex pairs $(x_i, x_j)$ in the same planted cluster with probability $p > 0.5$, and samples $(+)$ edges between vertex pairs $(x_i, x_j)$ in different clusters with probability $1 - p$. The SBM captures a lot of real-world scenarios, including social networks [HLL83], community detection [Abb17b], graph clustering [LW19], and Bioinformatics [MGC21], to name a few.

We test SBM instances that are reasonably "cluserable", i.e., we set $p = 0.8$ with vertices $n = 500$, $n = 1000$, and $n = 2000$. Furthermore, we compare the costs estimated by algorithms for the SBM and the Erdos-Renyi random graphs $G(n, p)$ with $p = 0.5$. In this regime, the Erdos-Renyi graph does not appear to have any clustering property, and the cost is high. Under the $n = 1000$ setting, we tested whether our algorithms can distinguish the costs between "good" (SBM with $p = 0.95$) and "bad" (Erdos-Renyi with $p = 0.5$) instances – a property that can be extremely useful in practice.

We implement our algorithms based on our descriptions in Section 3 and Section 4 with some relaxation of parameters. Notably, for the pivot-based algorithm, we slightly relax the requirement to allow 2-pass over the stream; as such, we can use the first pass to perform greedy MIS on the sampled vertex set, so we do not have to pay the $2^{O(1/\delta)}$ factor in the space.

We evaluate the performances mainly based on two metrics: the multiplicative factor of the cost estimation (which we call the "competitive ratio") and the fraction of the edges used. To overcome the possible effects of random seed, we fix our random seed from $0$ to $14$, and run 15 experiments. We include the error bars and the curves of the competitive ratios and fraction of the edges. For the experiment to distinguish SBM and Erdos-Renyi graphs, we simply plot the two types of costs w.r.t. experiment runs, and give the distributions of the costs. All the experiments were conducted on two macbooks and public Colab clusters.

**Experimental Results**    We first show that our algorithms are insensitive to the choice of parameters – this property is evident by the figures we used in the full version in the supplementary material. As such, we focus on the settings with $\varepsilon = 0.01$ for the SDD-based algorithm and $\delta = 0.1$ for the pivot-based algorithm for the rest of this section.

Due to the space limit, we only discuss our algorithm based on sparse dense decomposition (SDD). The left 3 plots in Figure 1 give the plots of the cost estimation for the SDD-based algorithm on $n = 2000$. The figures show that the approximation factor for this algorithm is roughly between 1 and 2, our algorithm consistently uses less than 4% of the edges in the graph.

The results for our SDD-based algorithm to distinguish between SBM instances with $p = 0.95$ and ER instances with $p = 0.5$ with the SDD-based algorithm is shown in the right two plots of Figure 1. From the figure, it can be observed that the the SDD-based algorithm outputs drastically different clustering costs of the SBM instances vs. the Erdos-Renyi instances. We can observe from the left plot that the supports of the costs are *disjoint*. As such, by a simple linear threshold, our SDD algorithm is able to perfectly distinguish between both types of instances while using less than 10% of the edges ($n = 1000$ case).

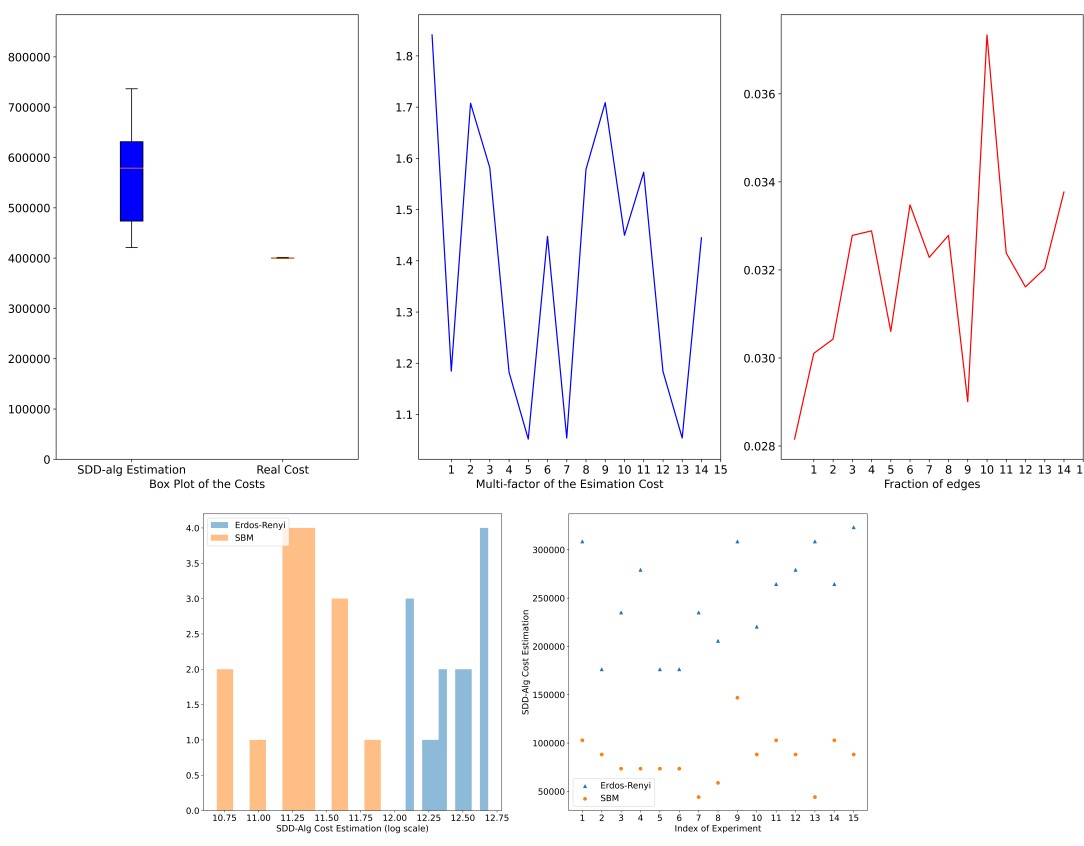

Figure 1: The performance of the SDD-based algorithm for cost estimation and instance clusterability distinguishing. Top 3 plots: the cost estimation of the SDD-based algorithm over 15 runs; Bottom 2 plots: the costs of SBM and ER graphs estimated by the SDD-based algorithm over 15 runs.

## Acknowledgments and Disclosure of Funding

Research supported in part by the Alfred P. Sloan Research Fellowship, a Faculty of Mathematics Research Chair Award from University of Waterloo, an NSF CAREER Grant CCF-2047061, a gift from Google Research, and a Rutgers Research Council Fulcrum Award.

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
