# Streaming Algorithms and Lower Bounds for Estimating Correlation Clustering Cost

Sepehr Assadi*       Vihan Shah *       Chen Wang*

## Abstract

Correlation clustering is a fundamental optimization problem at the intersection of machine learning and theoretical computer science. Motivated by applications to big data processing, recent years have witnessed a flurry of results on this problem in the streaming model. In this model, the algorithm needs to process the input $n$-vertex graph by making one or few passes over the stream of its edges and using a limited memory, much smaller than the input size.

All previous work on streaming correlation clustering has focused on semi-streaming algorithms with $\Omega(n)$ memory, whereas in this work, we study streaming algorithms with much smaller memory requirements of only polylog$(n)$ bits. This stringent memory requirement is in the same spirit of classical streaming algorithms that instead of recovering a full solution to the problem—which can be prohibitively large with such small memory as is the case in our problem—, aimed to learn certain statistical properties of their inputs. In our case, this translates to determining the "(correlation) clusterability" of input graphs, or more precisely, estimating the cost of the optimal correlation clustering solution.

As our main result, we present two novel algorithms that in only polylog$(n)$ space are able to estimate the optimal correlation clustering cost up to some constant multiplicative factor plus some extra additive error. One of the algorithms outputs a 3-multiplicative approximation plus $o(n^2)$ additive approximation, and the other one further reduces the additive error at the cost of increasing the multiplicative factor to some large constant. We then present new lower bounds that justify the mix of both multiplicative and additive error approximations in our algorithms.

---

*({sepehr.assadi,vihan.shah98,chen.wang.cs}@rutgers.edu) Department of Computer Science, Rutgers University. Research supported in part by a

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

 with planted clustering and some "noise." In particular, we arbitrarily label each vertex with a clustering label to form clusters with size $\Omega(n)$. Then, for two vertices $v_i$ and $v_j$, if they belong to the same planted cluster, we add a $(+)$ edge $(x_i, x_j)$ with probability $p$, and a $(-)$ edge with probability $1 - p$ for some $p > 0.5$. On the other hand, if $x_i$ and $x_j$ belong to different clusters, we add a $(+)$ edge with probability $1 - p$ and a $(-)$ edge with probability $p$. Stochastic Block Models are interesting when $p$ is sufficiently large, and the instance will be very "clusterable" with a small cost. In such a case, we can safely assume the planted clusters are optimal, and we can compute the correlation clustering cost by adding the $(+)$ edges crossing different planted clusters and $(-)$ edges inside the same ones.

We implement our algorithms on the simulations of graph streams obtained by the Stochastic Block Model. Our simulation results show that our algorithms consistently obtain approximations within a factor at most 4 of the optimal clustering cost, while only storing roughly 0.04% and 3.6% [3] of all possible edges respectively, when the graphs is moderately large (2000 vertices, 1999000 edges). We further tested our algorithms' ability to distinguish "well-clusterable" vs. "badly-clusterable" instances – the latter can be obtained by sampling Erdos-Renyi random graphs with $p = 0.5$, where optimal correlation clustering cost is roughly to put every vertex in a singleton cluster. We observe that our algorithms can always distinguish the costs between the SBM and Erdos-Renyi graphs when the probability $p$ in SBM is high enough: both estimation algorithms obtain *disjoint* cost distributions among the two types of graphs.

## 1.2   Our Techniques

**Upper bound of Result 1.**   Our first upper bound result borrows the idea of sparse-dense decomposition-based algorithm from [AW22]. In a nutshell, the sparse-dense decomposition tech-

---

[3]0.04% fraction of edges are obtained by implementing the algorithm of Result 2 in a two-pass manner

nique used in [AW22] categorizes the vertices as two types based on the $G^+$ subgraph: the *sparse vertices* $V_{\mathrm{sparse}}$ and *almost cliques* $K_1, \ldots, K_k$. For each sparse vertex $v$, a significant portion of its neighboring vertices $u \in N^+(v)$ have considerably different $(+)$ adjacent vertices than $v$, i.e. $N^+(v)$ and $N^+(u)$ have a large disjoint part. On the other hand, for each almost clique $K_i$, the vertices inside $K_i$ are densely connected with $(+)$ edges, and there are only a few $(+)$ edges going outside. [AW22] shows that by putting each almost-clique in a separate cluster and treating vertices in $V_{\mathrm{sparse}}$ as singletons, we can get an $O(1)$-approximation of the optimal cost.

The sparse-dense decomposition technique naturally requires $\Omega(n)$ space to write down the clustering. However, if we can estimate the *cost* of a sparse-dense decomposition-based clustering in smaller space, we can still obtain $O(1)$-approximation of the optimal cost. Indeed, for any vertex whose number of $(+)$ edges are sufficiently high, i.e. at least $\delta n$, a sample size of $O(1/\mathrm{poly}(\delta))$ is sufficient to test whether the vertex is sparse with high constant probability. As such, it seems promising to test whether each high-degree (in $G^+$) vertex belongs to $V_{\mathrm{sparse}}$, and count the costs induced by the set of edges as prescribed in [AW22]. Furthermore, the low-degree vertices can incur at most $\delta n^2$ additive error, and we can simply add this term in the final output.

The above idea is quite similar to the strategy we actually used in the algorithm, albeit there are some subtle yet important differences. We note that the sparse-dense decomposition in [AW22] is *not* unique – leaving alone the inherent randomness used in the algorithm, the sparse-dense assignments of some vertices are affected by the ordering of vertices. As such, we proceed differently by using the notion of sparse and dense *edges* instead.

Roughly speaking, we say a $(+)$ edge is sparse if its endpoints possess substantially different neighborhoods in $G^+$. Similarly, we say a non-edge in $G^+$ (i.e., a $(-)$ edge) to be dense if its endpoints share a large overlap of the neighborhoods in $G^+$. Intuitively, a vertex is sparse if it has many incident sparse edges; similarly, a vertex in some almost-clique only induces costs on its dense non-edges. It turns out that if we count in $G^+$ all the sparse edges and *at most* $\deg^+(v)$ dense non-edges for each vertex $v$, the resulting value is guaranteed to be between $\mathsf{OPT}$ and $O(1) \cdot \mathsf{OPT}$.

The final missing piece is to estimate in small space the number of desired sparse edges and dense non-edges as prescribed above. Similar to the estimation of vertices, we note that a sample size of $O(1/\mathrm{poly}(\delta))$ is sufficient to estimate the sparse-dense information on vertex pairs with both $\deg^+(v) \geq \delta n$. Furthermore, we can estimate the count with small variance by randomly sampling $O_\delta(\mathrm{polylog}(n))$ pairs. Finally, by simply forgoing the vertices with small degrees in $G^+$, the induced additive error is at most $\delta n^2$, as desired.

**Upper bound of Result 2.** Our algorithm in Result 2 is inspired by the pivot-based local algorithm from [BGK13]. The algorithmic procedure in [BGK13] is quite straightforward (although the analysis is highly non-trivial): we uniformly at random sample $O(1/\delta^2)$ vertices $U$ together with an ordering $\pi$ of the vertices. We then perform a greedy Maximal Independent Set (MIS) for vertices in $U$ on $G^+$ according to the ordering of $\pi$. By an analysis that is similar in spirit to [ACN05], one can show that the correlation clustering cost on the vertices induced by the MIS is at most $3\mathsf{OPT}$ in expectation. Furthermore, by pruning the MIS of the graph $G^+$, there are at most $\delta n^2$ $(+)$ edges remaining, for which we can afford to put in singleton clusters with an additive cost of $\delta n^2$.

Our algorithm takes the idea to test the cost of this pivot-based algorithm in limited space. With techniques similar to what we used in Result 1, we can sample $O(1/\delta^2)$ vertices uniformly at random; and for each $v \in U$, we can test the number of $(-)$ edges inside $N^+(v)$ and number of $(+)$ edges leaving $N^+(v)$, provided $v$ has at least $\delta n$ incident $(+)$ edges. However, simply adding up these numbers for each vertex in $U$ may lead to an *overestimation* of the cost since there can be large overlaps between the vertices in $U$ and their positive neighborhoods.

To handle the above problem, we count the number of edges that induce costs more carefully. For a vertex $v$ in the MIS induced by $U$, suppose we know the set of vertices $S$ in the MIS *before*

$v$ by the ordering of $\pi$. If we can test the number of cost-induced edges that do *not* incident on any vertices in $N^+[S] \cup S$, where $N^+[S]$ stands for the union of the positive neighbors of vertices in $S$, then we will not encounter the overestimation issue. Indeed, we can approximately count the number of cost-incurring edges with such constraints by storing whether they have $(+)$ edges to $S$. Since $S$ is at most of $O(1/\delta^2)$ size, the blow-up on the space is at most $O(1/\text{poly}(\delta))$.

Finally, we need to handle the fact that we do not know such $S$ for each vertex $v$, and we cannot use an ordered MIS on-the-fly as a later insertion/deletion may completely change the MIS. As such, for each vertex $v \in U$, we simply keep *all possible* $S$ sets during the stream to store the cost information, which introduces an extra $2^{O(1/\delta)}$ factor to the space. By the end of the stream, we can compute the real MIS in $U$ offline (by storing all the $O_\delta(\text{polylog}(n))$ edges), and use the corresponding $S$ set for each vertex to compute the cost value.

**Lower bounds.** Our lower bounds are mostly adapted from well-known communication problems, albeit with some novel tricks. In particular, our lower bound in Result 3 borrows idea from a recent reduction from INDEX in [AAD$^+$23], which was used to give an $\Omega(n^2)$ space lower bound to return the *exact* correlation clustering cost in a single pass. We prove that the same space-error trade-off holds even if we duplicate each vertex multiple times. As such, we can obtain a lower bound of $\Omega(n^\varepsilon)$ for any algorithm with only $O(n^{2-\varepsilon})$ additive error. Our lower bound in Result 4 follows from a reduction from the famous Gap-Cycle Counting (GCC) problem. En route to proving the lower bound, we slightly tweak the original instance of Gap-cycle Counting, and show that the lower bound works also for *odd* cycles, which turns out to be necessary to separate the correlation clustering costs.

## 2 Preliminaries

We introduce the necessary notation, the problem definition, and the standard technical tools we are going to use in this section.

### 2.1 Notation

We use $G = (V, E^+ \cup E^-)$ to denote a labeled complete graph arriving in a stream. For any vertex $v$, we denote by $N^+(v)$ all vertices that have an $(+)$ edge from $v$, and we let $N^+[v] = \{v\} \cup N^+(v)$. Similarly, we denote by $N^-(v)$ all vertices in the vertex set $V$ that have an $(-)$ edge from $v$, and we let $N^-[v] = \{v\} \cup N^-(v)$. We use $N^+(u) \triangle N^+(v)$ (resp. $N^-(u) \triangle N^-(v)$) to denote the disjoint neighborhoods of $u$ and $v$, i.e.

$$N^+(u) \triangle N^+(v) = (N^+(u) \cup N^+(v)) \setminus (N^+(u) \cap N^+(v)).$$

For a fixed set of vertices $A \subseteq V$, we further let $E^+(v, A)$ be the set of $(+)$ edges between $v$ and vertices in $A$, and $E^-(v, A)$ be the set of $(-)$ edges between $v$ and vertices in $A$.

We use $G^+ = (V, E^+)$ to denote the positive subgraph of $G$ with all the $(+)$ edges, and we define $G^- = (V, E^-)$ analogously. Note that since we work with a labeled complete graph, the information of $G^-$ edges can be uniquely inferred from the positive subgraph $G^+$. As such, when we work with $G^+$ only, we call a $(+)$ edge $(u, v) \in E^+$ as an *edge in* $G^+$. Similarly, we call a $(-)$ edge $(u, v) \in E^-$ an *non-edge in* $G^+$. We may omit $G^+$ when the context is clear.

For a fixed cluster $\mathcal{C}$ on a labeled complete graph $G$, we use $\text{cost}(\mathcal{C})$ to denote the total cost of correlation clustering on $G$ by $\mathcal{C}$ (see Problem 1 for the formal definition of cost). Furthermore, we slightly abuse the notation to reload $\text{cost}(\cdot)$ as a function of cost on subgraphs of $G$ in the following occasions:

1. For an induced subgraph $H \subseteq G$, we let $\text{cost}(\mathcal{C}, H)$ be the cost of $\mathcal{C}$ induced by the edges with both endpoints in $H$;

2. For two vertex sets $A, B \subseteq V$, we let $\mathsf{cost}(\mathcal{C}, (A, B))$ be the cost of $\mathcal{C}$ induced by the edges with exactly one endpoint in $A$ and one endpoint in $B$.

Finally, for a single edge $(u, v)$, we use the notation $\mathsf{edge\text{-}cost}(\mathcal{C}, (u, v))$ to denote the cost induced by a single edge $(u, v)$ for clustering $\mathcal{C}$. We may write a short-hand notation $\mathsf{edge\text{-}cost}((u, v))$ when the context is clear that $\mathcal{C}$ is used.

## 2.2 Problem Definition

We now give the formal description of the problem.

**Problem 1** (Correlation Clustering Value Estimation)**.** Given a labeled complete graph $G = (V, E^- \cup E^+)$ and a clustering $\mathcal{C}$ that partitions $V$ into disjoint set of vertices $C_1, C_2, \cdots, C_k$, the cost of disagreement minimization correlation clustering on $\mathcal{C}$ is defined as

$$\mathsf{cost}(\mathcal{C}) := \left| \{ (u, v) \in E^+ \mid \exists i \neq j \text{ s.t. } u \in C_i, v \in C_j \} \right| + \left| \{ (u, v) \in E^- \mid \exists i \text{ s.t. } u, v \in C_i \} \right|$$

Let $\mathsf{OPT}$ be the minimum $\mathsf{cost}$ over all possible valid clusterings. The Correlation Clustering Value Estimation problem asks for a number $\mathsf{ALG}$ such that $\mathsf{OPT} \leq \mathsf{ALG} \leq f(\mathsf{OPT})$ for some function $f(x) \geq x$.

If $f(\mathsf{OPT}) = \alpha \cdot \mathsf{OPT} + \beta \cdot n^2$ for $\alpha$ and $\beta$, we say that $\mathsf{ALG}$ is a $(\alpha, \beta)$-approximation of the value of correlation clustering in this scenario.

## 2.3 The Streaming Model

We study algorithms for correlation clustering value estimation under the *graph streaming* model, where edges arrive one after another in a stream with the labels. We introduce the graph streaming model for the labeled complete graph for both insertion-only and dynamic streams.

**Insertion-only labeled complete graph stream.** Given a graph $G = (V, E^+ \cup E^-)$, the insertion-only graph stream of $G$ is a length-$T$ sequence of tuples $\langle \sigma_1, \sigma_2, \cdots, \sigma_T \rangle$ such that $T = \binom{n}{2}$, and each $\{\sigma_t\}_{t=1}^{T}$ is consistent of

$$\sigma_t = ((u_t, v_t), x_t),$$

where $(u_t, v_t)$ specifies a unique vertex pair and $x_t$ gives the label of $(u_t, v_t)$.

**Dynamic labeled complete graph stream.** Given a graph $G = (V, E^+ \cup E^-)$, the dynamic graph stream of $G$ is a length-$T$ sequence of tuples $\langle \sigma_1, \sigma_2, \cdots, \sigma_T \rangle$ such that $T \geq \binom{n}{2}$, and each $\{\sigma_t\}_{t=1}^{T}$ is consistent of

$$\sigma_t = ((u_t, v_t), \Delta_t),$$

where $(u_t, v_t)$ specifies a vertex pair and $\Delta_t$ gives the *update* of the label on $(u_t, v_t)$ with the following rules:

- If $(u_t, v_t)$ does *not* have any $(+)$ or $(-)$ edges, $\Delta_t$ can be the insertion of $(+)$ or $(-)$ labels, but cannot be a deletion operation.

- If $(u_t, v_t)$ has at least one $(+)$ label, $\Delta_t$ can be the insertion or deletion of a $(+)$ edge, but it cannot specify any operation for $(-)$ edges.

- If $(u_t, v_t)$ has at least one $(-)$ label, $\Delta_t$ can be the insertion or deletion of a $(-)$ edge, but it cannot specify any operation for $(+)$ edges.

In other words, the dynamic stream is allowed to perform insertions and deletions whenever it is *consistent* with the existing labels on the vertex pair. Finally, the stream has to provide a labeled complete graph in the end, i.e. every pair of vertices has exactly one $(+)$ or $(-)$ edge.

We note that our algorithms work even when given a dynamic stream of just the positive edges (all pairs of vertices that have no edge at the end of the stream have a $(-)$ edge). We further note that our algorithms work even when the complete graph in the end has $\text{poly}(n)$ $(+)$ edges (or $(-)$ edges) between a pair of vertices (the multiple edges are thought of as just one edge and not parallel edges). The space blowup in this case is only a multiplicative factor of $\log n$.

For a pair of vertices, knowing whether it has a $(+)$ or $(-)$ edge at the end of the stream can be done with a 1 bit counter. We just need to remember if the last insertion between that pair of vertices was a $(+)$ or $(-)$ edge. If the stream contains only $(+)$ edges then a counter mod 2 does the job. So in conclusion we just need 1 bit. In the case where we have $\text{poly}(n)$ edges between a pair of vertices in the end, we use a counter of size $O(\log n)$ bits.

## 2.4 Standard Technical Tools

**Concentration inequalities.** We use the following standard multiplicative and additive forms of Chernoff bounds.

**Proposition 2.1** (Multiplicative Chernoff bound; c.f. [DP09]). *Suppose $X_1, \ldots, X_m$ are $m$ independent random variables with range $[0,1]$ each. Let $X := \sum_{i=1}^{m} X_i$ and $\mu_L \leq \mathbb{E}[X] \leq \mu_H$. Then, for any $\varepsilon > 0$, there is*

$$\Pr(X > (1+\varepsilon) \cdot \mu_H) \leq \exp\left(-\frac{\varepsilon^2 \cdot \mu_H}{3+\varepsilon}\right) \quad \text{and} \quad \Pr(X < (1-\varepsilon) \cdot \mu_L) \leq \exp\left(-\frac{\varepsilon^2 \cdot \mu_L}{2+\varepsilon}\right).$$

**Proposition 2.2** (Additive Chernoff bound; c.f. [DP09]). *Suppose $X_1, \ldots, X_m$ are $m$ independent random variables with range $[0,1]$ each. Let $X := \sum_{i=1}^{m} X_i$ and $\mu_L \leq \mathbb{E}[X] \leq \mu_H$. Then, for any $t > 0$, there is*

$$\Pr(X > \mu_H + t) \leq \exp\left(-\frac{2t^2}{m}\right) \quad \text{and} \quad \Pr(X < \mu_L - t) \leq \exp\left(-\frac{2t^2}{m}\right).$$

**Sampling from graph streams.** For a graph $G = (V, E^+ \cup E^-)$, it is straightforward to sample an edge uniformly at random in an insertion-only graph stream. For dynamic graphs, the standard tool is the $\ell_0$ sampler ([FIS08, LW16b]) that takes $O(\log^3(n))$ space and samples an edge with probability at least $1 - 1/\text{poly}(n)$. However, in our algorithms, we sample *vertices* and store the edges between the sampled vertices. As such, we do *not* have to use the $\ell_0$ samplers and we directly keep track of the label of the last update – this only takes $O(1)$ words for each pair of vertices. Finally, since sampling techniques can be viewed as a *sketching matrix* in the stream, we use the name *sketching tools* to denote the sampling subroutines as standard.

**Sparse-dense decomposition.** Following the definition in [AW22], an $\varepsilon$-sparse-dense decomposition can be defined as follows.

**Definition 1** ([AW22]). Given a graph $G = (V, E)$, an $\varepsilon$-sparse-dense decomposition $V_{\text{sparse}} \sqcup K_1 \sqcup \ldots \sqcup K_k$ consists of:

- **Sparse vertices** $V_{\text{sparse}}$: Any vertex $v \in V_{\text{sparse}}$ has at least $\eta_0 \cdot \varepsilon \cdot \deg(v)$ neighbors $u$ such that: $|N(v) \triangle N(u)| \geq \eta_0 \cdot \varepsilon \cdot \max\{\deg(u), \deg(v)\}$.

- **Dense vertices partitioned into almost-cliques** $K_1, \ldots, K_k$: For every $i \in [k]$, each $K_i$ has the following properties. Let $\Delta(K_i)$ be the maximum degree (in $G$) of the vertices in $K_i$, then:

*i*). Every vertex $v \in K_i$ has at most $\varepsilon \cdot \Delta(K_i)$ *non-neighbors* inside $K_i$;

*ii*). Every vertex $v \in K_i$ has at most $\varepsilon \cdot \Delta(K_i)$ *neighbors* outside $K_i$;

*iii*). Size of each $K_i$ satisfies $(1 - \varepsilon) \cdot \Delta(K_i) \leq |K_i| \leq (1 + \varepsilon) \cdot \Delta(K_i)$.

[AW22] proved that for sufficiently small $\varepsilon$ and $\eta_0$, such a decomposition always exists – this can be formalized as follows.

**Proposition 2.3.** *For $\varepsilon \leq \frac{1}{360}$ and $\eta_0 \leq \frac{1}{20}$, for any input graph $G = (V, E)$, the $\varepsilon$-sparse-dense decomposition of $G$ defined as in Definition 1 always exists.*

We will use Proposition 2.3 on the subgraph $G^+ = (V, E^+)$ of the labeled complete graph to obtain lower bounds for the correlation clustering costs.

**Communication complexity and streaming lower bounds.** We use reduction arguments from two-party one-way communication complexity to prove streaming lower bounds in this paper. The framework works as follows: in a two-party one-way communication game between Alice and Bob, a communication protocol $\Pi$ can always be simulated by a streaming algorithm if the players can sample the input. Concretely, Alice can simply run the streaming algorithm locally and send the memory to Bob, where Bob can run the rest of the streaming algorithm and output the answer. As such, the one-way communication complexity implies the memory lower bound for single-pass streaming algorithms used in the reduction.

# 3 An Algorithm based on Sparse-dense Decomposition

In this section, we present our first streaming algorithm that achieves a $(O(1), \delta n^2)$-approximation for testing the value of correlation clustering in $O(\text{polylog}(n)/\text{poly}(\delta))$ space, as long as $\delta \geq \frac{1}{\text{polylog}(n)}$. We utilize the idea in [AW22] to test a $(O(1), \delta n^2)$-approximate value of the sparse-dense decomposition-based correlation clustering cost, which in turn is an $O(1)$ approximation of the optimal cost. Our algorithm uses a memory of $O(\text{polylog}(n)/\text{poly}(\delta))$ bits, which is efficient for large-scale inputs. More formally, the guarantee of our algorithm is as follows.

**Theorem 1.** *There is a (dynamic) streaming algorithm that with high probability gives a $(O(1), \delta n^2)$-approximation for the correlation clustering value and take space $O\left(\frac{\log^2(n)}{\delta^5}\right)$ words.*

On a high level, our algorithm for Theorem 1 uses the idea to approximate the value of optimal correlation clustering with sparse-dense decomposition prescribed as in Definition 1. It is shown in [AW22] that once we can find such a decomposition on $G^+$, we can achieve an $O(1)$-approximation by simply putting every sparse vertex into a singleton cluster and gathering each almost-clique $K_i$ in a separate cluster. In this way, the costs are essentially only induced by 3 types of edges: 1. the positive edges incident on a sparse vertex (call them *sparse-vertex edges*); 2. the positive edges that connect different almost-cliques (call them *cross almost-clique edges*); and 3. negative edges inside the almost-cliques. Therefore, an approximation of the number of the aforementioned edges in a *fixed* decomposition will result in a good estimation of the cost.

However, since the sparse-dense decomposition is not unique, it is unclear how to estimate the edges for a fixed decomposition. On the other hand, an algorithm with polylog($n$) space is necessarily oblivious to the decomposition which takes $\Theta(n)$ bits to write down. To overcome this problem, we forgo the strict notion of the sparse-dense decomposition, and utilize the notion of $\varepsilon$-*sparse edges* and $\varepsilon$-*dense non-edges* (of $G^+$) instead. Roughly speaking, the two notions focus on 'local' edge properties as opposed to the decomposition outputs. We will eventually prove that these two notions

capture edges that possibly induce costs in *every* sparse-dense decomposition, and the overestimation of the cost is still within an $O(1)$ factor.

The rest of this section is structured as follows. In Section 3.1, we first describe self-contained sketching tools to approximately test whether an edge (or a non-edge) is $\varepsilon$-sparse or $\varepsilon$-dense. Subsequently, in Section 3.2, we show our desired estimators towards the test of the cost. And finally, we put the two parts together and present the algorithm in Section 3.3.

## 3.1 Sketching tools for sparse and dense edges

In this section, we present two sketching tools to test edges with certain properties. In particular, the first one to detect edges whose endpoints have sufficiently disjoint neighborhoods, and the second one to detect non-edges whose endpoints have overlapping neighborhoods. Formally, we define the notions of $\varepsilon$-sparse and $\varepsilon$-dense edges/non-edges as follows.

**Definition 2.** Fix an arbitrary graph $G = (V, E)$ (which is *not* necessarily labeled and complete) and a vertex pair $(u, v) \in G$, we say

1. $(u, v)$ is an $\varepsilon$-sparse edge (resp. non-edge) if $(u, v) \in E$ (resp. $(u, v) \notin E$) and $|N(v) \triangle N(u)| \geq \varepsilon \cdot \max\{\deg(u), \deg(v)\}$.

2. $(u, v)$ is an $\varepsilon$-dense edge (resp. non-edge) if $(u, v) \in E$ (resp. $(u, v) \notin E$) and $|N(v) \triangle N(u)| \leq \varepsilon \cdot \max\{\deg(u), \deg(v)\}$.

Note that the definitions of the $\varepsilon$-sparse and $\varepsilon$-dense edges/non-edges are generic and *not* restricted to the labeled complete graph (or even the correlation clustering application). Similarly, the sketching tools we design in this section are also generic: we will use them in the context of correlation clustering later.

### 3.1.1 The tool for $\varepsilon$-sparse edges

In this section, we present a tool to test whether an edge is (approximately) $\varepsilon$-sparse with high probability. We start with defining the goal of our tool as the following problem.

**Problem 2.** Consider a graph $G = (V, E)$ specified in a (dynamic) stream and let $u, v$ be a pair of vertices known before the stream. The goal is to

1. Output "Yes" if $(u, v)$ is *at least* $\varepsilon$-sparse.

2. Output "No" if $(u, v)$ is *not* $\frac{\varepsilon}{8}$-sparse.

with high probability assuming the promise that both $u$ and $v$ have degree at least $\delta \cdot n$. i.e. $\deg(u) \geq \delta \cdot n$, $\deg(v) \geq \delta \cdot n$.

We remark that by our definition of $\varepsilon$-sparse edges, if an edge $(u, v)$ is *not* $\frac{\varepsilon}{8}$-sparse, it is *not* sparse for any parameters *larger* than $\frac{\varepsilon}{8}$. To keep the argument simple, we do not optimize the leading constant for $\varepsilon$. Our tool is as follows.

---

**Algorithm 1. Tool-spr: Tool to test if $(u, v)$ is $\varepsilon$-sparse**

**Input:** $G = (V, E)$ in a (dynamic) stream and vertices $u, v \in V$ before the stream.

**Promise:** $\deg^+(u) \geq \delta n$ and $\deg^+(v) \geq \delta n$.

**Pre-Processing:**

---

1. Sample $k := \frac{200}{\varepsilon^2 \delta} \cdot \log(n)$ vertices $z_1, z_2, \ldots, z_k$ uniformly at random into set $S$.

**During the Stream:**

1. For every $i \in [k]$, store a counter $C_u(i)$ for the number of edges between $u$ and $z_i$. Also, store a counter $C_v(i)$ for the number of edges between $v$ and $z_i$.

2. Store the degrees of $u$ and $v$.

**Post-Processing:**

1. Let $N_{u,v}^S := |(N(u) \triangle N(v)) \cap S|$.

2. If $N_{u,v}^S \geq \frac{\varepsilon}{4} \cdot |(N(u) \cup N(v)) \cap S|$, output "Yes"; otherwise, output "No".

---

We prove the following lemma about Algorithm 1.

**Lemma 3.1.** *Algorithm 1 with parameters $\varepsilon, \delta$ and input $u, v$ satisfying the promise $\deg^+(u) \geq \delta n$ and $\deg^+(v) \geq \delta n$ solves Problem 2 with high probability by outputting 'Yes" if $(u, v)$ is at least $\varepsilon$-sparse and "No" if $(u, v)$ is not $\frac{\varepsilon}{8}$-sparse using $O(1/\varepsilon^2\delta)$ words of space.*

Let $E_S$ denote the set of edges where $C_u(i) > 0$ or $C_v(i) > 0$. Let $N_u^S := |N(u) \cap S|$, $N_v^S := |N(v) \cap S|$, and we know $N_{u,v}^S = |(N(u) \triangle N(v)) \cap S|$. Observe that $N_{u,v}^S$ can be easily computed using the stored information. We start by showing that the space complexity of **Tool-spr** is small.

**Claim 3.2.** *The space taken by **Tool-spr** is at most $O\left(\frac{1}{\varepsilon^2\delta}\right)$ words.*

*Proof.* For each $i \in [k]$, we store two $O(1)$ bit counters for $z_i$. Hence, the total space taken is at most $O(|S|) = O\left(\frac{1}{\varepsilon^2\delta}\right)$ words. □

We now show the correctness of the tool. Consistent with the desired properties, the algorithm has the following guarantees for the output:

**Claim 3.3.** *The answers for **Tool-spr** on a single (valid) edge $(u, v)$ satisfy the following with high probability:*

- *If $|N(u) \triangle N(v)| \geq \varepsilon \cdot \max\{\deg(u), \deg(v)\}$, **Tool-spr** outputs "Yes".*

- *If $|N(u) \triangle N(v)| \leq \frac{\varepsilon}{8} \cdot \max\{\deg(u), \deg(v)\}$, **Tool-spr** outputs "No".*

The rest of Section 3.1.1 is to prove Claim 3.3. As the first step, we show that since the degrees of $u$ and $v$ are sufficiently large, a large number of vertices in $S$ should be in $N(u)$ and $N(v)$ with high probability. Formally, we show that

**Claim 3.4.** *With high probability, we have $N_u^S \geq \frac{100}{\varepsilon^2} \cdot \log(n)$ and $N_v^S \geq \frac{100}{\varepsilon^2} \cdot \log(n)$.*

*Proof.* Let us analyze the quantity $N_u^S$ (the analysis for $N_v^S$ is identical). For each vertex $i \in [k]$, define random variable $N_u^i$ as the indicator for whether $z_i$ is in $N(u)$. Thus, we have $N_u^S = \sum_i N_u^i$ as the total number of vertices in $S$ that are neighbors of $u$. The assumption, $\deg(u) \geq \delta \cdot n$ implies $\mathbb{E}\left[N_u^i\right] = \Pr(z_i \in N(u)) \geq \delta$. Therefore, in expectation, the number of $z_i$'s in $N(u)$ is

$$\mathbb{E}\left[N_u^S\right] = \sum_i \mathbb{E}\left[N_u^i\right] \geq |S| \cdot \delta = \frac{200}{\varepsilon^2} \cdot \log(n). \qquad (|S| = \frac{200}{\varepsilon^2\delta} \cdot \log(n))$$

Note that $N_u^S$ is a summation of independent indicator random variables. Therefore, by the Chernoff bound(Proposition 2.1), we can get

$$\Pr\left(N_u^S \le \frac{100}{\varepsilon^2} \cdot \log(n)\right) = \Pr\left(N_u^S \le (0.5) \cdot \mathbb{E}\left[N_u^S\right]\right)$$

$$\le \exp\left(-\frac{\mathbb{E}\left[N_u^S\right]}{4 \cdot 3}\right)$$

$$\le n^{-10}.$$

To get the desired result, one can apply a union bound over the failure probabilities of the events that $N_u^S$ and $N_v^S$ are large enough. □

Now, with a sufficiently large neighborhoods in $S$ for both $u$ and $v$, we are able to prove the desired outputs by applying Chernoff bounds. Formally, we can obtain the following claims.

**Claim 3.5.** *Conditioning on the event of Claim 3.4, if $(u, v)$ is an $\varepsilon$-sparse edge, **Tool-spr** answers "Yes" with high probability.*

*Proof.* Consider an edge $(u, v)$ that is $\varepsilon$-sparse, and a sample $z_i \in N(u) \cup N(v)$. The probability for $z_i$ to be in $N(u) \triangle N(v)$ is:

$$\Pr\left(z_i \in N(u) \triangle N(v) | z_i \in N(u) \cup N(v)\right) = \frac{|N(u) \triangle N(v)|}{|N(u) \cup N(v)|}$$

$$\ge \frac{\varepsilon \cdot \max\{\deg(u), \deg(v)\}}{|N(u) \cup N(v)|} \quad \text{(definition of $\varepsilon$-sparse edge)}$$

$$\ge \frac{\varepsilon}{2}. \qquad (|N(u) \cup N(v)| \le 2 \cdot \max\{\deg(u), \deg(v)\})$$

Therefore, taking a vertex sample $z_i \in (N(u) \cup N(v)) \cap S$, we can define $N_{u,v}^i$ as the indicator random variable for whether $z_i \in (N(u) \triangle N(v)) \cap S$. We have $N_{u,v}^S = \sum_i N_{u,v}^i$ and its expected value is:

$$\mathbb{E}\left[N_{u,v}^S\right] = \sum_i \mathbb{E}\left[N_{u,v}^i\right]$$

$$= \sum_{z_i \in (N(u) \cup N(v)) \cap S} \Pr\left(z_i \in N(u) \triangle N(v)\right)$$

$$\ge \frac{\varepsilon}{2} \cdot |(N(u) \cup N(v)) \cap S|.$$

By the additive Chernoff bound (Proposition 2.2), the probability for $N_{u,v}^S$ to be less than $\frac{\varepsilon}{4} \cdot |(N(u) \cup N(v)) \cap S|$ is at most

$$\Pr\left(N_{u,v}^S \le \frac{\varepsilon}{4} \cdot |(N(u) \cup N(v)) \cap S|\right) \le \exp\left(-\frac{\varepsilon^2}{2} \cdot |(N(u) \cup N(v)) \cap S|\right)$$

$$\le \exp\left(-\frac{\varepsilon^2}{2} \cdot \frac{100}{\varepsilon^2} \cdot \log(n)\right)$$

$$(|(N(u) \cup N(v)) \cap S| \ge \frac{100}{\varepsilon^2} \cdot \log(n) \text{ by conditioning on Claim 3.4})$$

$$\le n^{-10},$$

which is the desired bound. □

**Claim 3.6.** *Conditioning on the event of Claim 3.4, if $(u,v)$ is not a $\frac{\varepsilon}{8}$-sparse edge, **Tool-spr** answers "No" with high probability.*

*Proof.* Similar to the proof of Claim 3.5, consider an edge $(u,v)$ that is *not* an $\frac{\varepsilon}{8}$-sparse edge, and a sample $z_i \in N(u) \cup N(v)$. The probability that $z_i$ is in $N(u) \triangle N(v)$ is:

$$\Pr\left(z \in N(u) \triangle N(v) | z_i \in N(u) \cup N(v)\right) = \frac{|N(u) \triangle N(v)|}{|N(u) \cup N(v)|}$$
$$\leq \frac{\varepsilon/8 \cdot \max\{\deg(u), \deg(v)\}}{|N(u) \cup N(v)|} \quad \text{(definition of } \varepsilon\text{-sparse edge)}$$
$$\leq \frac{\varepsilon}{8}. \quad (\max\{\deg(u), \deg(v)\} \leq |N(u) \cup N(v)|)$$

Therefore, taking a vertex sample $z_i \in (N(u) \cup N(v)) \cap S$, we can define $N_{u,v}^i$ as the indicator random variable for whether $z_i \in (N(u) \triangle N(v)) \cap S$. We have $N_{u,v}^S = \sum_i N_{u,v}^i$ and its expected value is:

$$\mathbb{E}\left[N_{u,v}^S\right] = \sum_i \mathbb{E}\left[N_{u,v}^i\right]$$
$$= \sum_{z_i \in (N(u) \cup N(v)) \cap S} \Pr\left(z_i \in N(u) \triangle N(v)\right)$$
$$\leq \frac{\varepsilon}{8} \cdot |(N(u) \cup N(v)) \cap S|.$$

Again, by the additive form of Chernoff bound, the probability for $N_{u,v}^S$ to be more than $\frac{\varepsilon}{4} \cdot |(N(u) \cup N(v)) \cap S|$ is at most

$$\Pr\left(N_{u,v}^S \geq \frac{\varepsilon}{4} \cdot |(N(u) \cup N(v)) \cap S|\right) \leq \exp\left(-\frac{\varepsilon^2}{2} \cdot |(N(u) \cup N(v)) \cap S|\right)$$
$$\leq \exp\left(-\frac{\varepsilon^2}{2} \cdot \frac{100}{\varepsilon^2} \cdot \log(n)\right)$$
$$(|(N(u) \cup N(v)) \cap S| \geq \frac{100}{\varepsilon^2} \cdot \log(n) \text{ conditioning on Claim 3.4})$$
$$\leq n^{-10},$$

as claimed. □

*Proof of Claim 3.3.* We can now apply a union bound over the events of Claim 3.4, Claim 3.5 and Claim 3.6, and all the events stated in the claims hold. Therefore, with high probability, **Tool-spr** works in the desired way. □

Claim 3.3 and Claim 3.2 together prove Lemma 3.1.

### 3.1.2 The tool for $\varepsilon$-dense non-edges

In this section, we present a tool to test whether a non-edge is (approximately) $\varepsilon$-dense with high probability. We start with defining the goal of our tool as the following problem.

**Problem 3.** Consider a graph $G = (V, E)$ specified in a (dynamic) stream and let $u, v$ be a pair of vertices known before the stream. The goal is to

1. Output "Yes" if $(u,v)$ is *at most* $\varepsilon$-dense.

2. Output "No" if $(u, v)$ is *not* $8\varepsilon$-dense.

with high probability assuming the promise that both $u$ and $v$ have degree at least $\delta \cdot n$ i.e. $\deg(u) \geq \delta \cdot n$, $\deg(v) \geq \delta \cdot n$.

Note that contrary to the rules of $\varepsilon$-sparse edges, if a non-edge is *not* $8\varepsilon$-dense, it is *not* dense for any parameter *smaller* than $8\varepsilon$. Our dense tool algorithm is as follows.

---

**Algorithm 2. Tool-dns: Tool to test if non-edge $(u, v)$ is $\varepsilon$-dense**

**Input:** $G = (V, E)$ in a (dynamic) stream and vertices $u, v \in V$ before the stream.

**Promise:** $\deg^+(u) \geq \delta n$ and $\deg^+(v) \geq \delta n$.

**Pre-Processing:**

1. Sample $k := \frac{200}{\varepsilon^2 \delta} \cdot \log(n)$ vertices $z_1, z_2, \ldots, z_k$ uniformly at random into set $S$.

**During the Stream:**

1. For every $i \in [k]$ store a counter $C_u(i)$ for the number of edges between $u$ and $z_i$. Also, store a counter $C_v(i)$ for the number of edges between $v$ and $z_i$.

2. Store the degrees of $u$ and $v$.

**Post-Processing:**

1. Let $N^S_{u,v} := |(N(u) \triangle N(v)) \cap S|$.

2. If $N^S_{u,v} \leq 2\varepsilon \cdot |(N(u) \cup N(v)) \cap S|$, output "Yes"; otherwise, output "No".

---

We prove the following lemma about Algorithm 2.

**Lemma 3.7.** *Algorithm 2 with parameters $\varepsilon, \delta$ and input $u, v$ satisfying the promise $\deg^+(u) \geq \delta n$ and $\deg^+(v) \geq \delta n$ solves Problem 3 with high probability by outputting 'Yes" if $(u, v)$ is at most $\varepsilon$-dense and "No" if $(u, v)$ is not $8\varepsilon$-dense using $O(1/\varepsilon^2 \delta)$ words of space.*

Let $E_S$ denote the set of edges where $C_u(i) > 0$ or $C_v(i) > 0$. Let $N^S_u := |N(u) \cap S|$, $N^S_v := |N(v) \cap S|$, and we know $N^S_{u,v} = |(N(u) \triangle N(v)) \cap S|$. It is easy to see that $N^S_{u,v}$ can be computed using the stored information. We start by showing that the space complexity of **Tool-dns** is small.

**Claim 3.8.** *The space taken by **Tool-dns** is at most $O\left(\frac{1}{\varepsilon^2 \delta}\right)$ words.*

*Proof.* For each $i \in [k]$, we store two $O(1)$ bit counters for $z_i$. Hence, the total space taken is at most $O(|S|) = O\left(\frac{1}{\varepsilon^2 \delta}\right)$ words. $\qquad\square$

We now show the correctness of **Tool-dns** in the same manner as **Tool-spr**. The formal statement is as follows:

**Claim 3.9.** *The answers for **Tool-dns** on a single (valid) non-edge $(u, v)$ satisfy the following with high probability:*

- *If $|N(u) \triangle N(v)| \leq \varepsilon \cdot \max\{\deg(u), \deg(v)\}$, **Tool-dns** outputs "Yes".*

- *If $|N(u) \triangle N(v)| \geq 8\varepsilon \cdot \max\{\deg(u), \deg(v)\}$, **Tool-dns** outputs "No".*

We first note that since the vertex sampling process of **Tool-dns** is identical to that of **Tool-spr**, one can directly apply Claim 3.4 to argue that with high probability the number of neighbors of $u$ and $v$ intersecting with $S$ is large. We now show that 'completeness' and 'soundness' of **Tool-dns** as the following claims.

**Claim 3.10.** *Conditioning on the event of Claim 3.4, if $(u, v)$ is an $\varepsilon$-dense non-edge, **Tool-dns** answers "Yes" with high probability.*

*Proof.* Consider a non-edge $(u, v)$ that is $\varepsilon$-dense and a sample $z_i \in N(u) \cup N(v)$. The probability for $z_i$ to be in $N(u) \triangle N(v)$ is:

$$\Pr(z_i \in N(u) \triangle N(v) | z_i \in N(u) \cup N(v)) = \frac{|N(u) \triangle N(v)|}{|N(u) \cup N(v)|}$$

$$\leq \frac{\varepsilon \cdot \max\{\deg(u), \deg(v)\}}{|N(u) \cup N(v)|}$$

$$\text{(definition of } \varepsilon\text{-dense non-edge)}$$

$$\leq \varepsilon. \qquad (\max\{\deg(u), \deg(v)\} \leq |N(u) \cup N(v)|)$$

Therefore, taking a vertex sample $z_i \in (N(u) \cup N(v)) \cap S$, we can define $N_{u,v}^i$ as the indicator random variable for whether $z_i \in (N(u) \triangle N(v)) \cap S$. We have $N_{u,v}^S = \sum_i N_{u,v}^i$ and its expected value is:

$$\mathbb{E}\left[N_{u,v}^S\right] = \sum_i \mathbb{E}\left[N_{u,v}^i\right]$$

$$= \sum_{z_i \in (N(u) \cup N(v)) \cap S} \Pr(z_i \in N(u) \triangle N(v))$$

$$\geq \varepsilon \cdot |(N(u) \cup N(v)) \cap S|.$$

Again, by the additive form of Chernoff bound (Proposition 2.2), the probability for $N_{u,v}^S$ to be more than $2\varepsilon \cdot |(N(u) \cup N(v)) \cap S|$ is:

$$\Pr\left(N_{u,v}^S \geq 2\varepsilon \cdot |(N(u) \cup N(v)) \cap S|\right) \leq \exp\left(-2\varepsilon^2 \cdot |(N(u) \cup N(v)) \cap S|\right)$$

$$\leq \exp\left(-2\varepsilon^2 \cdot \frac{100}{\varepsilon^2} \cdot \log(n)\right)$$

$$(|(N(u) \cup N(v)) \cap S| \geq \frac{100}{\varepsilon^2} \cdot \log(n) \text{ conditioning on Claim 3.4})$$

$$\leq n^{-10},$$

as claimed. □

**Claim 3.11.** *Conditioning on the event of Claim 3.4, if $(u, v)$ is not an $8\varepsilon$-dense non-edge (i.e. $|N(v) \triangle N(u)| \geq 8\varepsilon \cdot \max\{\deg(u), \deg(v)\}$), **Tool-dns** answers "No" with high probability.*

*Proof.* Consider a non-edge $(u, v)$ that is not $8\varepsilon$-dense and a sample $z_i \in N(u) \cup N(v)$. The probability for $z_i$ to be in $N(u) \triangle N(v)$ is:

$$\Pr(z \in N(u) \triangle N(v)) = \frac{|N(u) \triangle N(v)|}{|N(u) \cup N(v)|}$$

$$\geq \frac{8\varepsilon \cdot \max\{\deg(u), \deg(v)\}}{|N(u) \cup N(v)|} \qquad \text{(definition of } \varepsilon\text{-dense non-edge)}$$

$$\geq 4\varepsilon. \qquad (|N(u) \cup N(v)| \leq 2 \cdot \max\{\deg(u), \deg(v)\})$$

Therefore, taking a vertex sample $z_i \in (N(u) \cup N(v)) \cap S$, we can define $N_{u,v}^i$ as the indicator random variable for whether $z_i \in (N(u) \triangle N(v)) \cap S$. We have $N_{u,v}^S = \sum_i N_{u,v}^i$ and its expected value is:

$$\mathbb{E}\left[N_{u,v}^S\right] = \sum_i \mathbb{E}\left[N_{u,v}^i\right]$$

$$= \sum_{z_i \in (N(u) \cup N(v)) \cap S} \Pr\left(z \in N(u) \triangle N(v)\right)$$

$$\geq 4\varepsilon \cdot |(N(u) \cup N(v)) \cap S|.$$

By the additive Chernoff bound (Proposition 2.2), the probability for $N_{u,v}^S$ to be less than $2\varepsilon \cdot |(N(u) \cup N(v)) \cap S|$ is at most

$$\Pr\left(N_{u,v}^S \leq 2\varepsilon \cdot |(N(u) \cup N(v)) \cap S|\right) \leq \exp\left(-8\varepsilon^2 \cdot |(N(u) \cup N(v)) \cap S|\right)$$

$$\leq \exp\left(-8\varepsilon^2 \cdot \frac{100}{\varepsilon^2} \cdot \log(n)\right)$$

$$(|(N(u) \cup N(v)) \cap S| \geq \tfrac{100}{\varepsilon^2} \cdot \log(n) \text{ conditioning on Claim 3.4})$$

$$\leq n^{-10},$$

as claimed. $\qquad\square$

*Proof of Claim 3.9.* Similar to the proof of Claim 3.3, we can now apply a union bound among the events of Claim 3.4, Claim 3.10 and Claim 3.11, and conclude that with high probability, **Tool-dns** works in the desired way. $\qquad\square$

Claim 3.9 and Claim 3.8 together prove Lemma 3.7.

## 3.2 Using $\varepsilon$-sparse Edges and $\varepsilon$-dense Non-edges for Correlation Clustering

In this section, we discuss how to use the $\varepsilon$-sparse edges and $\varepsilon$-dense non-edges (defined in Definition 2) in $G^+$ to estimate the correlation clustering cost. In what follows, we use $E_{\varepsilon\text{-sparse}}^+$ to denote the set of $\varepsilon$-sparse (positive) edges, and $E_{\varepsilon\text{-dense}}^-$ to denote the set of $\varepsilon$-dense non-edges. Furthermore, for each vertex $v$, we use $E_{\varepsilon\text{-dense},\,v}^-$ to denote the set of $\varepsilon$-dense non-edges incident on $v$. We let $m_{\varepsilon\text{-sparse}}^+ := \left|E_{\varepsilon\text{-sparse}}^+\right|$ denote the number of $\varepsilon$-sparse edges, $m_{\varepsilon\text{-dense}}^- := \left|E_{\varepsilon\text{-dense}}^-\right|$ denote the number of $\varepsilon$-dense non-edges and let $m_{\varepsilon\text{-dense},v}^- := \left|E_{\varepsilon\text{-dense},\,v}^-\right|$ denote the number of $\varepsilon$-dense non-edges incident on $v$. Finally, we define $\widehat{m}_{\varepsilon\text{-dense}}^-$ as follows.

**Definition 3.** For each vertex $v$ define

$$\widehat{m}_{\varepsilon\text{-dense},v}^- := \min\{m_{\varepsilon\text{-dense},v}^-, \deg^+(v)\},$$

where $\deg^+(v)$ is the number of edges incident on $v$ (in $G^+$). Furthermore, let $\widehat{m}_{\varepsilon\text{-dense}}^-$ be defined as

$$\widehat{m}_{\varepsilon\text{-dense}}^- := \sum_{v \in V} \widehat{m}_{\varepsilon\text{-dense},v}^-.$$

The intuition behind $\widehat{m}_{\varepsilon\text{-dense},v}^{-}$ is to count the non-edges (of $G^+$) in $E_{\varepsilon\text{-dense},\,v}^{-}$ for *at most* $\deg^+(v)$ *times*. Our estimator for $\varepsilon$-dense non-edges will estimate $\widehat{m}_{\varepsilon\text{-dense}}^{-}$ instead of $m_{\varepsilon\text{-dense}}^{-}$ for the following reason: it suffices to estimate $\widehat{m}_{\varepsilon\text{-dense}}^{-}$ since the number of non-edges inside each almost-clique is at most $\deg^+(v)$; on the other hand, if we estimate *all* $\varepsilon$-dense non-edges, there could be a very large overhead since the non-edges *between* almost-cliques are also counted. We note that if $\widehat{m}_{\varepsilon\text{-dense},v}^{-} = m_{\varepsilon\text{-dense},v}^{-}$ for all $v \in V$ then $\widehat{m}_{\varepsilon\text{-dense}}^{-}$ is twice $m_{\varepsilon\text{-dense}}^{-}$ (since we are double counting edges). But this is at most a 2-approximation in the worst case, and we are doing this to make calculations easier. To proceed with the properties of $\widehat{m}_{\varepsilon\text{-dense}}^{-}$, we first give the following observations.

**Observation 3.12.** *For any $\varepsilon$, the quantity of $\widehat{m}_{\varepsilon\text{-dense}}^{-}$ is upper-bounded by $\sum_{v \in V} \deg^+(v)$.*

The proof of Observation 3.12 is trivial since we 'cap' the number of non-edges for a vertex with $\deg^+(v)$. We now give another observation to characterize the upper and lower bounds of $\widehat{m}_{\varepsilon\text{-dense}}^{-}$ w.r.t. different sets of non-edges.

**Observation 3.13.** *Consider the quantity $\widehat{m}_{\beta\varepsilon\text{-dense}}^{-}$ and $\widehat{m}_{\beta'\varepsilon\text{-dense}}^{-}$ for $\beta' > \beta$, the following properties are true:*

1. *$\widehat{m}_{\beta\varepsilon\text{-dense}}^{-} \leq \widehat{m}_{\beta'\varepsilon\text{-dense}}^{-}$ for $\beta' > \beta$.*

2. *Consider* any *non-edge set $E'$ that satisfies the following: i) $e$ is at most $\beta\varepsilon$-dense for all $e \in E'$; ii) the number non-edges in $E'$ that are incident on $u$ is at most $\deg^+(u)$ for all $u \in V(E')$. Then, we have $|E'| \leq \widehat{m}_{\beta\varepsilon\text{-dense}}^{-}$.*

3. *Consider* any *non-edge set $E''$ obtained by the following process:* "Go over all vertices in $V$ one by one and for vertex $v$ add $\deg^+(v)$ non-edges that are at most $\beta'\varepsilon$-dense to $E''$ (if the remaining number is less than $\deg^+(v)$ then add all such remaining non-edges)". *Then, we have $\widehat{m}_{\beta'\varepsilon\text{-dense}}^{-} \leq 2\,|E''|$.*

*Proof.* The first property of Observation 3.13 is evident since we 'relax' the parameter. We have $E_{\beta\varepsilon\text{-dense}}^{-} \subseteq E_{\beta'\varepsilon\text{-dense}}^{-}$ implying $\widehat{m}_{\beta\varepsilon\text{-dense}}^{-} \leq \widehat{m}_{\beta'\varepsilon\text{-dense}}^{-}$. For the second property, we know that for a vertex $v$, the number of non-edges of $E'$ incident on $v$ ($|E'(v)|$) can be at most $\deg^+(v)$. $|E'(v)|$ can also trivially be upper bounded by $m_{\beta\varepsilon\text{-dense},v}^{-}$ since $E'$ only contains $\beta\varepsilon$-dense non-edges implying that $|E'(v)| \leq \widehat{m}_{\beta\varepsilon\text{-dense},v}^{-}$. Thus, we have

$$|E'| \leq \sum_{v \in V} |E'(v)| \leq \sum_{v \in V} \widehat{m}_{\beta\varepsilon\text{-dense},v}^{-} = \widehat{m}_{\beta\varepsilon\text{-dense}}^{-}.$$

Finally, for the third property, let $E''(v)$ be the number of non-edges in $E''$ incident on $v$. We know that $|E''(v)|$ is exactly $\deg^+(v)$ except in case there are fewer $\beta'\varepsilon$-dense non-edges incident on $v$ then we add all of them. Thus, $|E''(v)| = \min(\deg^+(v), m_{\beta'\varepsilon\text{-dense},v}^{-}) = \widehat{m}_{\beta'\varepsilon\text{-dense},v}^{-}$ implying that

$$\widehat{m}_{\beta'\varepsilon\text{-dense}}^{-} = \sum_{v \in V} \widehat{m}_{\beta'\varepsilon\text{-dense},v}^{-} = \sum_{v \in V} |E''(v)| \leq 2\,|E''|.$$

We have an inequality because there might be a $\beta'\varepsilon$-dense non-edge $(u,v)$ which is in $E''(u)$ but not in $E''(v)$ because of the $\deg^+(v)$ upper bound (so $(u,v)$ is not counted twice). $\qquad\square$

We now establish the relationship between the $\varepsilon$-sparse edges, $\varepsilon$-dense non-edges, and the costs of correlation clustering.

**Correlation clustering properties of $\varepsilon$-sparse edges and $\varepsilon$-dense non-edges**

The key properties for the connections between the $\varepsilon$-sparse edges/$\varepsilon$-dense non-edges and the correlation clustering follow the idea in [AW22]. In particular, we will show that

- If a clustering pays the cost of all the $\alpha \cdot \varepsilon$-sparse edges and sufficiently many $\beta \cdot \varepsilon$-dense non-edges for each vertex $v$ (as defined by $\widehat{m}^-_{\beta\varepsilon\text{-dense}}$), where $\alpha$ and $\beta$ are suitable parameters, the estimation never drops below OPT;

- If a clustering only pays the cost of edges that are *at least* $\alpha' \cdot \varepsilon$-sparse and $\deg(v)$ number of non-edges that are *at most* $\beta' \cdot \varepsilon$-dense for each $v$ (as defined by $\widehat{m}^-_{\beta'\varepsilon\text{-dense}}$), where $\alpha'$ and $\beta'$ are suitable parameters, the estimation is upper-bounded by $O(1) \cdot$ OPT.

To continue, we remind the readers that we insist the definition of $\varepsilon$-sparse edges and $\varepsilon$-dense non-edges to be on $G^+$, and use the notion of non-edge and $(-)$ edge as the same meaning.

We prove the following lemmas that establish the connections between the aforementioned set of edges and the costs of correlation clustering.

**Lemma 3.14.** *Suppose $G = (V, E)$ is any labeled graph and $V = V_{sparse} \sqcup K_1 \sqcup \ldots \sqcup K_k$ is an $\varepsilon$-sparse-dense decomposition of $G^+$ for $0 \leq \varepsilon \leq 1/360$ and $\eta_0 \leq 1/20$. Let edge-cost$(e)$ denote the cost for a fix clustering $\mathcal{C}$ pays for edge $e \in E^+ \cup E^-$. Then*

a) *The cost of $(+)$ edges $(u, v) \in E^+$ such that $u \in V_{sparse}$ or $v \in V_{sparse}$ is at most $\frac{2}{\eta_0 \varepsilon}$ times of the number of all the $\varepsilon$-sparse $(+)$ edges. I.e.,*

$$\sum_{\substack{(u,v) \in E^+ \\ u \in V_{sparse} \ or \ v \in V_{sparse}}} \text{edge-cost}((u,v)) \leq \frac{2}{\eta_0 \cdot \varepsilon} \cdot m^+_{\eta_0\varepsilon\text{-}sparse}.$$

b) *The cost of $(+)$ edges $(u, v) \in E^+$ such that $u \in K_i$ and $v \in K_j$ for some $i \neq j$ is at most the number of all the $\varepsilon$-sparse $(+)$ edges. I.e.,*

$$\sum_{\substack{(u,v) \in E^+ \\ u \in K_i, v \in K_j, \\ i \neq j}} \text{edge-cost}((u,v)) \leq m^+_{\varepsilon\text{-}sparse}.$$

c) *The cost of $(-)$ edges $(u, v)$ $((u, v) \in E^-)$ such that $u, v \in K_i$ for the same $i$ is at most $\widehat{m}^-_{4\varepsilon\text{-}dense}$. I.e.,*

$$\sum_{\substack{(u,v) \in E^- \\ u, v \in K_i}} \text{edge-cost}((u,v)) \leq \widehat{m}^-_{4\varepsilon\text{-}dense}.$$

*Proof.* We prove the three properties in order.

**Proof of properties $a)$ and $b)$.** Property $a)$ is true by the definition of sparse vertices. Fix any sparse vertex $v \in V_{sparse}$. By definition, $v$ has at least $\eta_0 \cdot \varepsilon \cdot \deg^+(v)$ neighbors $u_1, u_2, \ldots$ such that $(u_i, v)$ is an $\eta_0\varepsilon$-sparse $(+)$ edge. This implies that $m^+_{\eta_0\varepsilon\text{-sparse},v} \geq \eta_0 \cdot \varepsilon \cdot \deg^+(v)$. Thus, we have

$$\sum_{\substack{(u,v) \in E^+ \\ u \in V_{sparse} \ or \ v \in V_{sparse}}} \text{edge-cost}((u,v)) \leq \sum_{v \in V_{sparse}} \deg^+(v) \leq \sum_{v \in V} \frac{1}{\eta_0 \cdot \varepsilon} \cdot m^+_{\eta_0\varepsilon\text{-sparse},v} = \frac{2}{\eta_0 \cdot \varepsilon} \cdot m^+_{\eta_0\varepsilon\text{-sparse}}$$

since edge-cost($(u,v)$) = 1 for all the edges in the sum. Note that the first inequality is because we may double count edges between two sparse vertices.

To prove property *b)*, it suffices to show that every edge $(u,v) \in E^+$ such that $u \in K_i$ and $v \in K_j$ for some $i \neq j$ is an $\varepsilon$-sparse edge. Fix $u \in K_i$ and $v \in K_j$. We have:

$$
\begin{aligned}
\left| N^+(u) - N^+(v) \right| &\geq \left| (N^+(u) \cap K_i) - (N^+(v) \cap K_i) \right| \\
&\geq |K_i| - \varepsilon \cdot \Delta(K_i) - \varepsilon \cdot \Delta(K_j) \\
&\geq \Delta(K_i) - \varepsilon \cdot \Delta(K_i) - \varepsilon \cdot \Delta(K_i) - \varepsilon \cdot \Delta(K_j).
\end{aligned}
$$

Similarly, we have $|N^+(v) - N^+(u)| \geq \Delta(K_j) - 2\varepsilon \cdot \Delta(K_j) - \varepsilon \cdot \Delta(K_i)$. Therefore, we have:

$$
\begin{aligned}
\left| N^+(u) \triangle N^+(v) \right| &= \left| N^+(u) - N^+(v) \right| + \left| N^+(v) - N^+(u) \right| \\
&\geq \Delta(K_i) - 3\varepsilon \cdot \Delta(K_i) + \Delta(K_j) - 3\varepsilon \cdot \Delta(K_j) \\
&\geq (1 - 3\varepsilon) \cdot \max \left\{ \deg(u), \deg(v) \right\} \\
&\geq \varepsilon \cdot \max \left\{ \deg(u), \deg(v) \right\}
\end{aligned}
$$

where the last inequality is by the choice of $\varepsilon$. Therefore, $(u,v)$ must be an $\varepsilon$-sparse edge.

**Proof of property *c)*.** To prove property *c)*, we start with proving $\Delta(K_i) \leq \deg(u)/(1-2\varepsilon)$ using Definition 1. We have $\deg(u) \geq |K_i| - \varepsilon \Delta(K_i) \geq (1-\varepsilon)\Delta(K_i) - \varepsilon \Delta(K_i)$.

We first show that every non-edge vertex pair $(u,v)$ such that $u, v \in K_i$ for the same $i$ is at most $4\varepsilon$-dense. To see this, note that for each vertex $u \in K_i$, the number of its non-neighbors in $K_i$ is at most $\varepsilon \cdot \Delta(K_i) \leq \frac{\varepsilon}{1-2\varepsilon} \cdot \max \left\{ \deg(u), \deg(v) \right\} \leq 2\varepsilon \cdot \max \left\{ \deg(u), \deg(v) \right\}$ (using Definition 1 and the choice of $\varepsilon \leq 1/4$). Therefore, number of vertices in $N^+(u) - N^+(v)$ is at most $2\varepsilon \cdot \max \left\{ \deg(u), \deg(v) \right\}$. The same bound holds for $N^+(v) - N^+(u)$ implying that $(u,v)$ is $4\varepsilon$-dense.

We now show that the number of non-edges in $K_i$ incident on $u$ is at most $\deg^+(v)$. All non-edges in $K_i$ are counted towards the cost. The number of non-edges in $K_i$ that are incident on $u$ is at most $\varepsilon \cdot \Delta(K_i) \leq \frac{\varepsilon}{1-2\varepsilon} \cdot \deg^+(v) \leq \deg^+(v)$.

We now conclude the proof of property *c)*. Let $E'$ be the set of non-edges within the almost cliques $K_i$ for $i \in [k]$. We know that every non-edge in $E'$ is at most $4\varepsilon$-dense. Fix a vertex $u$ in some almost-clique $K_i$. By the above arguments we know that the number of non-edges of $E'$ incident of $u$ is at most $\deg^+(u)$. Using Observation 3.13 with $\beta = 4$, the total cost of the non-edges inside almost-cliques $|E'|$ is upper bounded by $\widehat{m}^-_{4\varepsilon\text{-dense}}$. $\qquad \square$

We now upper bound the cost of $m^+_{\varepsilon\text{-sparse}}$ and $\widehat{m}^-_{\varepsilon\text{-dense}}$ by a function of OPT using a charging argument. Formally, we present the following lemma:

**Lemma 3.15.** *Suppose $G = (V, E)$ is any labeled graph and* OPT *be the optimal correlation clustering cost, and let $\beta, \varepsilon$ be such that $0 < \beta \leq \frac{1}{2\varepsilon}$, there is*

$$
m^+_{\beta\varepsilon\text{-sparse}} \leq \frac{2}{\beta\varepsilon} \cdot \mathsf{OPT};
$$

$$
\widehat{m}^-_{\beta\varepsilon\text{-dense}} \leq 8 \cdot \mathsf{OPT}.
$$

The proof of Lemma 3.15 is considerably technical, but the idea is straightforward following the charging arguments in [AW22, CLM$^+$21]. As such, we postpone it to Appendix A.

## 3.3 The Sparse-dense Decomposition-based Algorithm

We are now ready to give the final algorithm for Problem 1 and prove Theorem 1. We will use the tools in Section 3.1 to estimate the number of $\varepsilon$-sparse edges and $\varepsilon$-dense non-edges. We know by Lemmas 3.14 and 3.15 that suitable scalings of the number of $\varepsilon$-sparse edges and $\varepsilon$-dense non-edges when added together is at least $\mathsf{OPT}$ and at most $O(1) \cdot \mathsf{OPT}$. Thus estimating these costs to within $\pm \delta \cdot n^2$ should give us the approximation we want. We start by showing an estimator for $m^+_{\varepsilon\text{-sparse}}$.

---

**Algorithm 3. Estimate $m^+_{\varepsilon\text{-sparse}}$**

**Input:** $G = (V, E)$ in a (dynamic) stream.

**Pre-Processing:**

1. Sample $k = \frac{16 \log n}{\delta^2}$ pairs of vertices $(u_1, v_1), (u_2, v_2), \ldots, (u_k, v_k)$.

**During the Stream:**

1. For all $i \in [k]$, store the degrees of $u_i$ and $v_i$ and store a counter for the number of edges between $u_i$ and $v_i$.

2. Also, store **Tool-spr**$(u_i, v_i)$ with parameters $\varepsilon$ and $\delta/4$ for all $i \in [k]$.

**Post-Processing:**

1. Let $i = 1$ to $k$.

2. For pair $(u_i, v_i)$ if degree of $u_i$ or $v_i$ is less than $\delta n/4$ then set $X_i = 0$.

3. If $(u_i, v_i)$ is an edge and if **Tool-spr**$(u_i, v_i)$ returns $\varepsilon$-sparse then set $X_i = 1$ otherwise $X_i = 0$.

4. After going over all edges, output $Z_{sp} = \delta \cdot n^2/2 + \frac{\binom{n}{2}}{k} \sum_{i=1}^{k} X_i$.

---

We prove the following lemma about Algorithm 3.

**Lemma 3.16.** *Algorithm 3 with parameters $\varepsilon$ and $\delta$ returns a value $Z_{sp}$ that is at least $m^+_{\varepsilon\text{-sparse}}$ and at most $m^+_{\varepsilon/8\text{-sparse}} + \delta n^2$ with high probability and uses $O(\log n/\varepsilon^2 \delta^3)$ words of space.*

A vertex $v$ is said to have a low degree if $\deg(v) < \delta \cdot n/4$. An edge is said to be low-degree if one of its endpoints is a low-degree vertex. There are very few low-degree edges thus, their contribution to the cost is small and can be ignored.

**Claim 3.17.** *The total cost of low-degree edges is at most $\delta n^2/4$.*

*Proof.* A low-degree vertex has at most $\delta n/4$ edges. Also, there are at most $n$ low-degree vertices thus there are at most $\delta n^2/4$ low-degree edges implying that their cost is at most $\delta n^2/4$. $\qquad \square$

We first bound the space of Algorithm 3.

**Claim 3.18.** *Algorithm 3 uses $O(\log n/\varepsilon^2 \delta^3)$ words of space.*

*Proof.* **Tool-spr**$(u_i, v_i)$ takes $O(1/\varepsilon^2 \delta)$ words of space. The space taken for storing the degrees and edges are lower-order terms. Repeating $k$ times gives us a space-bound of $O(\log n/\varepsilon^2 \delta^3)$ words. $\qquad \square$

We now show that Algorithm 3 gives us an estimate of $m^+_{\varepsilon\text{-sparse}}$.

**Claim 3.19.** *The output $Z_{sp}$ of Algorithm 3 is at least $m^+_{\varepsilon\text{-sparse}}$ and at most $m^+_{\varepsilon/8\text{-sparse}} + \delta n^2$ with high probability.*

*Proof.* Let $X = \frac{1}{k} \sum_{i=1}^{k} X_i$. We first lower bound $X$ by $m^+_{\varepsilon\text{-sparse}}$. We have

$$\mathbb{E}[X_i] = \Pr(X_i = 1) \geq \frac{m^+_{\varepsilon\text{-sparse}} - \delta n^2/4}{\binom{n}{2}} = \mu_L$$

since $X_i$ is 1 for all edges that are not low-degree and are $\varepsilon$-sparse. We know $\mathbb{E}[X] = \mathbb{E}[X_i]$ by linearity of expectation. By Hoeffding's inequality (Proposition 2.2) we have:

$$\Pr(X < \mu_L - \delta/2) \leq \Pr(X < \mathbb{E}[X] - \delta/2) \leq \exp\left(-2k \cdot \delta^2/4\right) \leq n^{-8}.$$

Thus, with high probability $X \geq \mu_L - \delta/2$ implying

$$\binom{n}{2} \cdot X \geq m^+_{\varepsilon\text{-sparse}} - \delta n^2/2.$$

The calculation is analogous when we want to prove the upper bound on $X$. We have

$$\mathbb{E}[X_i] = \Pr(X_i = 1) \leq \frac{m^+_{\varepsilon/8\text{-sparse}}}{\binom{n}{2}} = \mu_H$$

since $X_i$ is 0 for all edges that are not $\varepsilon/8$-sparse. By Hoeffding's inequality (Proposition 2.2) we have:

$$\Pr(X > \mu_H + \delta/2) \leq \Pr(X > \mathbb{E}[X] + \delta/2) \leq \exp\left(-2k \cdot \delta^2/4\right) \leq n^{-8}.$$

Thus, with high probability $X \leq \mu_H + \delta/2$ implying

$$\binom{n}{2} \cdot X \leq m^+_{\varepsilon/8\text{-sparse}} + \delta n^2/2.$$

Therefore, with high probability:

$$m^+_{\varepsilon\text{-sparse}} \leq Z_{sp} \leq m^+_{\varepsilon/8\text{-sparse}} + \delta n^2.$$

Finally, the bounds above are true with high probability and the tools succeed with high probability, thus a union bound gives us success with high probability. $\square$

Claim 3.18 and Claim 3.19 together prove Lemma 3.16.

We now show how to estimate the value of $\widehat{m}^-_{\varepsilon\text{-dense}}$. We start by estimating $\widehat{m}^-_{\varepsilon\text{-dense},v}$ (Definition 3). Consider the following algorithm:

---

**Algorithm 4. Estimate $\widehat{m}^-_{\varepsilon\text{-dense},v}$**

**Input:** $G = (V, E)$ in a (dynamic) stream and a vertex $v \in V$.

**Promise:** $\deg^+(v) \geq \delta n$.

**Pre-Processing:**

---

1. Sample $k = \left( \frac{16 \log n}{\delta^2} \right)$ vertices $u_1, u_2, \ldots, u_k$ uniformly at random and independently.

**During the Stream:**

1. Store $\deg^+(v)$.

2. Store a counter for the number of edges between $u_i$ and $v$ for all $i \in [k]$.

3. Finally, store **Tool-dns**$(u_i, v)$ with parameters $\varepsilon$ and $\delta$ for all $i \in [k]$.

**Post-Processing:**

1. Let $i = 1$ to $k$.

2. If $(u_i, v)$ is a non-edge and if **Tool-dns**$(u_i, v)$ returns $\varepsilon$-dense then $Y_i = 1$.

3. After going over all sampled vertices, return $Z_v = \min(\deg^+(v), \frac{n}{k} \sum_{i=1}^{k} Y_i) + \delta n/2$.

We prove the following lemma about Algorithm 4.

**Lemma 3.20.** *Algorithm 4 with parameters $\varepsilon$ and $\delta$ returns a value $Z_v$ that is at least $\widehat{m}^-_{\varepsilon\text{-}dense,v}$ and at most $\widehat{m}^-_{8\varepsilon\text{-}dense,v} + \delta n$ with high probability and uses $O(\log n/\varepsilon^2 \delta^3)$ words of space.*

We first bound the space of Algorithm 4.

**Claim 3.21.** *Algorithm 4 uses $O(\log n/\varepsilon^2 \delta^3)$ words of space.*

*Proof.* **Tool-spr**$(u_i, v_i)$ takes $O(1/\varepsilon^2 \delta)$ words of space. The space taken for storing the degrees and edges are lower-order terms. Repeating $k$ times gives us a space-bound of $O(\log n/\varepsilon^2 \delta^3)$ words. $\square$

We now show that Algorithm 4 gives us an estimate of $\widehat{m}^-_{\varepsilon\text{-}dense,v}$.

**Claim 3.22.** *The output $Z_v$ of Algorithm 4 is at least $\widehat{m}^-_{\varepsilon\text{-}dense,v}$ and at most $\widehat{m}^-_{8\varepsilon\text{-}dense,v} + \delta n$ with high probability.*

*Proof.* Let $Y = \frac{1}{k} \sum_{i=1}^{k} Y_i$. We first lower bound $Y$ by $m^-_{\varepsilon\text{-}dense,v}$. We have

$$\mathbb{E}[Y_i] = \Pr(Y_i = 1) \geq \frac{m^-_{\varepsilon\text{-}dense,v}}{n} = \mu_L$$

since $Y_i$ is 1 for all edges that $\varepsilon$-dense. We know $\mathbb{E}[Y] = \mathbb{E}[Y_i]$ by linearity of expectation. By Hoeffding's inequality (Proposition 2.2) we have:

$$\Pr(Y < \mu_L - \delta/2) \leq \Pr(Y < \mathbb{E}[Y] - \delta/2) \leq \exp\left(-2k \cdot \delta^2/4\right) \leq n^{-8}.$$

Thus, with high probability $Y \geq \mu_L - \delta/2$ implying

$$n \cdot Y \geq m^-_{\varepsilon\text{-}dense,v} - \delta n/2.$$

The calculation is analogous when we want to prove the upper bound on $Y$. We have

$$\mathbb{E}[Y_i] = \Pr(Y_i = 1) \leq \frac{m^-_{8\varepsilon\text{-}dense,v}}{n} = \mu_H$$

since $Y_i$ is 0 for all edges that are not $8\varepsilon$-dense. By Hoeffding's inequality (Proposition 2.2) we have:

$$\Pr\left(Y > \mu_H + \delta/2\right) \leq \Pr\left(Y > \mathbb{E}\left[Y\right] + \delta/2\right) \leq \exp\left(-2k \cdot \delta^2/4\right) \leq n^{-8}.$$

Thus, with high probability $Y \leq \mu_H + \delta/2$ implying

$$n \cdot Y \leq m^-_{8\varepsilon\text{-dense},v} + \delta n/2.$$

The output $Z_v$ is $\delta n/2$ plus the minimum of $n \cdot Y$ and $\deg^+(v)$. So we have:

$$
\begin{aligned}
Z_v &= \min(\deg^+(v), nY) + \delta n/2 \\
&\geq \min(\deg^+(v), m^-_{\varepsilon\text{-dense},v} - \delta n/2) + \delta n/2 \\
&\geq \min(\deg^+(v), m^-_{\varepsilon\text{-dense},v}) - \delta n/2 + \delta n/2 \\
&= \widehat{m}^-_{\varepsilon\text{-dense},v}.
\end{aligned}
$$

Similarly, we have $Z_v \leq \widehat{m}^-_{8\varepsilon\text{-dense},v} + \delta n$. Therefore, with high probability:

$$\widehat{m}^-_{\varepsilon\text{-dense}} \leq Z_v \leq \widehat{m}^-_{8\varepsilon\text{-dense}} + \delta n.$$

Finally, the bounds above are true with high probability and the tools succeed with high probability, thus a union bound gives us success with high probability. □

Claim 3.21 and Claim 3.22 together prove Lemma 3.20.
We are now ready to estimate $\widehat{m}^-_{\varepsilon\text{-dense}}$ as defined in Definition 3.

---

**Algorithm 5. Estimate $\widehat{m}^-_{\varepsilon\text{-dense}}$**

**Input:** $G = (V, E)$ in a (dynamic) stream.

**Pre-Processing:**

1. Sample $k = \frac{64 \log n}{\delta^2}$ vertices $v_1, v_2, \ldots, v_k$ uniformly at random and independently.

**During the Stream:**

1. For all $i \in [k]$ store $\deg(v_i)$ and let $Y_i$ be the estimate of $\widehat{m}^-_{\varepsilon\text{-dense},v_i}$ using Algorithm 4 with parameters $\varepsilon$ and $\delta/4$.

**Post-Processing:**

1. For $i \in [k]$, if $\deg^+(v_i) < \delta n/4$ then set $Y_i = \delta n/4$.

2. Output $Z_{den} = \frac{n}{k} \sum_{i=1}^{k} Y_i + \delta \cdot n^2/2$.

---

We prove the following lemma about Algorithm 5.

**Lemma 3.23.** *Algorithm 5 with parameters $\varepsilon$ and $\delta$ returns a value $Z_{den}$ that is at least $\widehat{m}^-_{\varepsilon\text{-dense}}$ and at most $\widehat{m}^-_{8\varepsilon\text{-dense}} + \delta n^2$ with high probability and uses $O(\log^2 n/\varepsilon^2 \delta^5)$ words of space.*

We first bound the space of Algorithm 5.

**Claim 3.24.** *Algorithm 5 uses $O(\log^2 n/\varepsilon^2\delta^5)$ words of space.*

*Proof.* Algorithm 4 takes $O(\log n/\varepsilon^2\delta^3)$ words of space. The space taken for storing the degrees is a lower-order term. Repeating $k$ times gives us a space-bound of $O(\log^2 n/\varepsilon^2\delta^5)$ words. $\qquad\square$

We make the following claim on the bounds of $Z_{den}$.

**Claim 3.25.** *The output $Z_{den}$ of Algorithm 5 is at least $\widehat{m}^-_{\varepsilon\text{-}dense}$ and at most $\widehat{m}^-_{8\varepsilon\text{-}dense} + \delta n^2$ with high probability.*

*Proof.* We know by Lemma 3.20 that with high probability we have $\widehat{m}^-_{\varepsilon\text{-dense},v_i} \leq Y_i \leq \widehat{m}^-_{8\varepsilon\text{-dense},v_i} + \delta n/4$ for all $i \in [k]$ (since we use parameter $\delta/4$ in Algorithm 5). This holds for $i \in [k]$ with $\deg^+(v_i) < \delta n/4$ because $Y_i = \delta n/4 \geq \deg^+(v_i) \geq \widehat{m}^-_{\varepsilon\text{-dense},v_i}$ and $Y_i = \delta n/4 \leq \widehat{m}^-_{8\varepsilon\text{-dense},v_i} + \delta n/4$. We condition on the high probability event of Lemma 3.20 for all $k$ copies used in Algorithm 5.

We first show the lower bound on $Y$. To do this we lower bound the expectation of $Y_i$. We have $\mathbb{E}[Y_i] \geq \frac{1}{n}\sum_{j=1}^n \widehat{m}^-_{\varepsilon\text{-dense},j} = \frac{1}{n}\cdot\widehat{m}^-_{\varepsilon\text{-dense}} = \mu_L$ since $v_i$ is uniform over all vertices. Scaling $Y$ and applying Hoeffding's inequality (Proposition 2.2) gives us:

$$\Pr\left(Y/n < \mu_L/n - \delta/2\right) \leq \Pr\left(Y/n < \mathbb{E}[Y/n] - \delta/2\right) \leq \exp\left(-2k\cdot\delta^2/4\right) \leq n^{-8}.$$

Thus, with high probability $Y \geq \mu_L - \delta n/2$ implying

$$n\cdot Y \geq \widehat{m}^-_{\varepsilon\text{-dense}} - \delta n^2/2.$$

The calculation is analogous when we want to prove the upper bound on $Y$. We know that $Y_i \leq \widehat{m}^-_{8\varepsilon\text{-dense},v_i} + \delta n/4$ for all $i \in [k]$. We have $\mathbb{E}[Y_i] \leq \frac{1}{n}\cdot\widehat{m}^-_{8\varepsilon\text{-dense}} + \delta n/4 = \mu_H$ since $v_i$ is uniform over all vertices. Thus, $\mathbb{E}[Y] = \mathbb{E}[Y_i]$ by linearity of expectation. By Hoeffding's inequality (Proposition 2.2) we have:

$$\Pr\left(Y/n > \mu_H/n + \delta/4\right) \leq \Pr\left(Y/n > \mathbb{E}[Y/n] - \delta/4\right) \leq \exp\left(-2k\cdot\delta^2/4^2\right) \leq n^{-8}.$$

Thus, with high probability $Y/n \leq \mu_H/n + \delta/4$ implying

$$n\cdot Y \geq \widehat{m}^-_{8\varepsilon\text{-dense}} + \delta n^2/2.$$

We conclude that $Z_{den} = n\cdot Y + \delta n^2/2$ has the following bounds with high probability.

$$\widehat{m}^-_{\varepsilon\text{-dense}} \leq Z_{den} \leq \widehat{m}^-_{8\varepsilon\text{-dense}} + \delta n^2.$$

Finally, the bounds above are true with high probability and the tools succeed with high probability, thus a union bound gives us success with high probability. $\qquad\square$

Claim 3.24 and Claim 3.25 together prove Lemma 3.23.

**Finalizing the proof of Theorem 1.** We are given a parameter $\delta$ as input and we want the additive error to be at most $\delta n^2$. We fix $\varepsilon = 1/360$, $\eta_0 = 1/20$ and $\delta' = \delta\cdot(2 + \frac{2}{\eta_0\varepsilon})^{-1}$.

Run Algorithm 3 with parameters $\varepsilon$ and $\delta'$, and let the output be $Z^\varepsilon_{sp}$. Also, run Algorithm 3 with parameters $\eta_0\varepsilon$ and $\delta'$, and let the output be $Z^{\eta_0\varepsilon}_{sp}$. Lemma 3.16 and Lemma 3.14 imply that $Z^\varepsilon_{sp} + (2/\eta_0\varepsilon)\cdot Z^{\eta_0\varepsilon}_{sp}$ is an upper bound on the cost of the positive edges:

$$Z^\varepsilon_{sp} + (2/\eta_0\varepsilon)\cdot Z^{\eta_0\varepsilon}_{sp} \geq m^+_{\varepsilon\text{-sparse}} + (2/\eta_0\varepsilon)\cdot m^+_{\eta_0\varepsilon\text{-sparse}}$$

$$\geq \sum_{\substack{(u,v)\in E^+ \\ u\in V_{\text{sparse}}\text{ or }v\in V_{\text{sparse}}}} \text{edge-cost}((u,v)) + \sum_{\substack{(u,v)\in E^+ \\ u\in K_i, v\in K_j, \\ i\neq j}} \text{edge-cost}((u,v)).$$

Also, by Lemma 3.16 and Lemma 3.15 we have:

$$Z_{sp}^{\varepsilon} + (2/\eta_0\varepsilon) \cdot Z_{sp}^{\eta_0\varepsilon} \leq \left( m_{\varepsilon/8\text{-sparse}}^+ + \delta'n^2 \right) + (2/\eta_0\varepsilon) \cdot \left( m_{\eta_0\varepsilon/8\text{-sparse}}^+ + \delta'n^2 \right)$$
$$\leq \frac{16}{\varepsilon} \mathsf{OPT} + \frac{2}{\eta_0\varepsilon} \cdot \frac{16}{\eta_0\varepsilon} \mathsf{OPT} + (1 + 2/\eta_0\varepsilon) \cdot \left( \delta'n^2 \right).$$

Run Algorithm 5 with parameters $4\varepsilon$ and $\delta'$, and let the output be $Z_{den}$. Lemma 3.23 and Lemma 3.14 imply that $Z_{den}$ is an upper bound on the cost of the negative edges:

$$Z_{den} \geq \widehat{m}_{4\varepsilon\text{-dense}}^- \geq \sum_{\substack{(u,v)\in E^- \\ u,v\in K_i}} \mathsf{edge\text{-}cost}(\,(u,v)).$$

Also, by Lemma 3.23 and Lemma 3.15 we have:

$$Z_{den} \leq \widehat{m}_{32\varepsilon\text{-dense}}^- + \delta'n^2 \leq 8\mathsf{OPT} + \delta'n^2.$$

We can now easily approximate the correlation clustering cost using Lemma 3.15. We have:

$$Z_{CC} := Z_{sp}^{\varepsilon} + (2/\eta_0\varepsilon) \cdot Z_{sp}^{\eta_0\varepsilon} + Z_{den} \geq m_{\varepsilon\text{-sparse}}^+ + (2/\eta_0\varepsilon) \cdot m_{\eta_0\varepsilon\text{-sparse}}^+ + \widehat{m}_{4\varepsilon\text{-dense}}^- \geq \mathsf{OPT}$$

and

$$Z_{CC} = Z_{sp}^{\varepsilon} + (2/\eta_0\varepsilon) \cdot Z_{sp}^{\eta_0\varepsilon} + Z_{den}$$
$$\leq \frac{16}{\varepsilon} \mathsf{OPT} + \frac{32}{\eta_0^2\varepsilon^2} \mathsf{OPT} + (1 + 2/\eta_0\varepsilon) \cdot \delta'n^2 + 8\mathsf{OPT} + \delta'n^2$$
$$= \left( \frac{32}{\eta_0^2\varepsilon^2} + \frac{16}{\varepsilon} + 8 \right) \cdot \mathsf{OPT} + \left( 2 + \frac{2}{\eta_0\varepsilon} \right) \delta'n^2.$$

By replacing $\delta' = \delta \cdot (2 + \frac{2}{\eta_0\varepsilon})^{-1}$ we get an additive error of at most $\delta n^2$. Substituting $\varepsilon = 1/360$ and $\eta_0 = 1/20$ gives us:

$$\mathsf{OPT} \leq Z_{CC} \leq O(1) \cdot \mathsf{OPT} + \delta n^2.$$

The space taken is $O(\log n/\delta^3)$ for both copies of Algorithm 3 and $O(\log^2 n/\delta^5)$ for Algorithm 5 giving a total space of $O(\log^2 n/\delta^5)$ words. This proves Theorem 1.

# 4  An Algorithm based on Pivot

In this section, we give our second streaming algorithm which is a $(3+\gamma, \delta n^2)$-approximation for the correlation clustering value for any choice of $\delta$ and $\gamma < 1/2$. The algorithm works in $O(\text{polylog}(n))$ space as long as $\delta \geq \Omega(\frac{1}{\log\log(n)})$. Consider the following formal statement:

**Theorem 2.** *There is a (dynamic) streaming algorithm that with high probability gives a $(3+\gamma, \delta n^2)$-approximation for the correlation clustering value and takes space $O\left( \frac{2^{7/6\delta} \cdot \log^2 n}{\gamma \cdot \delta^5} \right)$ words.*

Our algorithm in Theorem 2 is inspired by [BGK13], and it requires new sketching tools (on $G^+$) that is different from the ones we used in Section 3. These tools estimate the number of non-edges **within** or the number of edges **going out of** the neighborhood of a vertex $(u)$ outside of the $(+)$ neighborhood of a known set of vertices $(S)$. In other words, fix a graph $G = (V, E)$, a vertex $u$, and a set $S$, we want to estimate the number of non-edges within or the number of edges going out of $N[u] - N[S]$. We again use the generic form to present the sketching tools and do not specify $(+)$ edges. The formal definitions can be given as follows.

**Definition 4.** A non-edge $(x, y)$ is **within** $N[u] - N[S]$ iff: *i*) $x \notin N[S]$ and *ii*) $y \notin N[S]$ and *iii*) $x \in N[u]$ and $y \in N[u]$.

**Definition 5.** An edge $(x, y)$ is **going out of** $N[u] - N[S]$ iff *i*) $x \notin N[S]$ and *ii*) $y \notin N[S]$ and *iii*) $x \in N[u]$ or $y \in N[u]$ but not both.

**Definition 6.** An edge $(x, y)$ is **unclustered** w.r.t. $S$ iff *i*) $x \notin N[S]$ and *ii*) $y \notin N[S]$.

We also define $m_{ne}(u, S)$ as the number of non-edges **within** $N[u] - N[S]$, $m_e(u, S)$ as the number of edges **going out of** $N[u] - N[S]$, and $m_u(S)$ as the number of **unclustered** edges w.r.t. $S$. We write $m_{ne}, m_e$, and $m_u$ when $u$ and $S$ are clear from context. Consider the following lemma about estimating the number of non-edges **within** $N[u] - N[S]$, the edges **going out of** $N[u] - N[S]$, and the **unclustered** edges with respect to $S$:

**Lemma 4.1.** *There exist streaming algorithms called* NE-Tool, E-Tool, *and* U-Tool *that given* $u$ *and* $S$ *before the stream, respectively compute with high probability i). the number of non-edges **within**; ii). the number of edges **going out of** $N[u] - N[S]$; and iii). the number of edges with **unclustered** w.r.t. $S$ with an overestimation of at most $\delta n^2$ and take space $O(1/\delta^2)$ words.*

In the next sections, we show in detail the novel sketching tools NE-Tool and E-Tool that are predecessor-aware, i.e. given a fixed set of vertices $S$ as the predecessors (in the MIS of the algorithm of Theorem 2), they can estimate the desired edges outside the neighborhood of $S$.

## 4.1   A Predecessor-aware Non-Edge Sketching Tool

In this section, we describe a tool to estimate the number of non-edges within the neighborhood of a vertex outside a set of vertices $(m_{ne}(u, S))$. Consider the formal description of the problem:

**Problem 4.** The input graph $G = (V, E)$ is given in a (possibly dynamic) stream. The input also contains a vertex $u$ and a set $S$ given before the stream. The goal is to estimate the number of non-edges **within** $N[u] - N[S]$ $(m_{ne}(u, S))$ with an overestimation of at most $\delta n^2$.

**Lemma 4.2.** *There is a (dynamic) streaming algorithm that solves Problem 4 with high probability using $O(1/\delta^2)$ words of space.*

We now show the streaming algorithm of Lemma 4.2:

---

**Algorithm 6.** NE-Tool

**Input:** $G = (V, E)$ in a (dynamic) stream and $u \in V$ and $S \subseteq V$ before the stream.

**Pre-Processing:**

1. Sample $k = \frac{10 \log n}{\delta^2}$ pairs of vertices $x_i, y_i$ uniformly at random

**During the Stream:**

1. For pair $(x_i, y_i)$, store the following information:

    (a)  A counter $C(i)$ for the number of edges between $x_i$ and $y_i$.

    (b)  A counter $C_x(i, S)$ for the number of edges between $x_i$ and $S$ and a counter $C_y(i, S)$ for the number of edges between $y_i$ and $S$.

    (c)  A counter $C_x(i, u)$ for the number of edges between $x_i$ and $u$ and a counter $C_y(i, u)$ for the number of edges between $y_i$ and $u$.

---

> **Post-Processing:**
>
> 1. For pair $i$ set $Z_i = 1$ iff $(x_i, y_i)$ is a non-edge **within** $N[u] - N[S]$ and set $Z_i = 0$ otherwise.
>
> 2. Let $\tilde{Z} = \binom{n}{2} \cdot \frac{1}{k} \cdot \left( \sum_i Z_i \right)$
>
> 3. Output $\tilde{Z} + \binom{n}{2}\delta$

It is easy to see that the stored information is enough to check whether $(x_i, y_i)$ is a non-edge **within** $N[u] - N[S]$. We now show that $\tilde{Z}$ is a good approximation for $m_{ne}$.

**Claim 4.3.** *With high probability, we have* $m_{ne} - \binom{n}{2}\delta \le \tilde{Z} \le m_{ne} + \binom{n}{2}\delta$.

*Proof.* $Z_i$ is a random variable since the choice of $x_i$ and $y_i$ is random. $x_i, y_i$ could be any of $\binom{n}{2}$ pairs but $Z_i$ is 1 only for the $m_{ne}$ pairs (non-edges within $N[u] - N[S]$). Thus, $\mathbb{E}\left[Z_i\right] = \Pr\left(Z_i = 1\right) = m_{ne}/\binom{n}{2}$. Let $\bar{Z} = \frac{1}{k} \cdot \left( \sum_i Z_i \right)$. Thus, $\mathbb{E}\left[\bar{Z}\right] = m_{ne}/\binom{n}{2}$. Using Hoeffding's inequality (Proposition 2.2) we get:

$$
\begin{aligned}
\Pr\left( \left| \bar{Z} - \mathbb{E}\left[\bar{Z}\right] \right| \ge \delta \right) &\le 2\exp\left( -2\delta^2 k \right) \\
&\le 2\exp\left( -2 \cdot 10 \log n \right) \\
&\le n^{-10}.
\end{aligned}
$$

So with high probability, we have

$$
\begin{aligned}
\left| \bar{Z} - \mathbb{E}\left[\bar{Z}\right] \right| &\le \delta \\
\frac{m_{ne}}{\binom{n}{2}} - \delta \le \bar{Z} &\le \frac{m_{ne}}{\binom{n}{2}} + \delta \\
m_{ne} - \binom{n}{2}\delta \le \tilde{Z} &\le m_{ne} + \binom{n}{2}\delta
\end{aligned}
$$

proving the claim. $\qquad\square$

The output of Algorithm 6 is $\tilde{Z} + \binom{n}{2}\delta$ implying $m_{ne} \le \tilde{Z} + \binom{n}{2}\delta \le m_{ne} + \delta n^2$ using Claim 4.3. This proves the estimation guarantee of `NE-Tool` in Lemma 4.2. We now show that the space used is $O(1/\delta^2)$ words.

**Claim 4.4.** *The space used by Algorithm 6 is* $O(1/\delta^2)$ *words.*

*Proof.* In one iteration, we store a constant number of counters which takes a space of $O(1)$ bits implying that the space used by the algorithm over all iterations is $O(1/\delta^2)$ words. $\qquad\square$

This proves the space-bound of `NE-Tool` in Lemma 4.2.

## 4.2 A Predecessor-aware Edge Sketching Tool

In this section, we describe a tool to estimate the number of edges going out of the neighborhood of a vertex outside a set of vertices $(m_e(u, S))$. Consider the formal description of the problem:

**Problem 5.** The input graph $G = (V, E)$ is given in a (possibly dynamic) stream. The input also contains a vertex $u$ and a set $S$ given before the stream. The goal is to estimate the number of edges **going out of** $N[u] - N[S]$ $(m_e(u, S))$ with an overestimation of at most $\delta n^2$.

**Lemma 4.5.** *There is a (dynamic) streaming algorithm that solves* Problem 5 *with high probability using $O(1/\delta^2)$ words of space.*

We now show the streaming algorithm of Lemma 4.5:

---

**Algorithm 7.** `E-Tool`

**Input:** $G = (V, E)$ in a (dynamic) stream and $u \in V$ and $S \subseteq V$ before the stream.

**Pre-Processing:**

1. Sample $k = \frac{10 \log n}{\delta^2}$ pairs of vertices $x_i, y_i$ at random

**During the Stream:**

1. For pair $(x_i, y_i)$, store the following information:

   (a) A counter $C(i)$ for the number of edges between $x_i$ and $y_i$.

   (b) A counter $C_x(i, S)$ for the number of edges between $x_i$ and $S$ and a counter $C_y(i, S)$ for the number of edges between $y_i$ and $S$.

   (c) A counter $C_x(i, u)$ for the number of edges between $x_i$ and $u$ and a counter $C_y(i, u)$ for the number of edges between $y_i$ and $u$.

**Post-Processing:**

1. For pair $i$ set $Z_i = 1$ iff $(x_i, y_i)$ is an edge **going out of** $N[u] - N[S]$ and set $Z_i = 0$ otherwise.

2. Let $\tilde{Z} = \binom{n}{2} \cdot \frac{1}{k} \cdot (\sum_i Z_i)$

3. Output $\tilde{Z} + \binom{n}{2}\delta$

---

It is easy to see that the stored information is enough to check whether $(x_i, y_i)$ is an edge **going out of** $N[u] - N[S]$.

We first show that $\tilde{Z}$ is a good approximation for $m_e$.

**Claim 4.6.** *With high probability, we have $m_e - \binom{n}{2}\delta \leq \tilde{Z} \leq m_e + \binom{n}{2}\delta$.*

*Proof.* $Z_i$ is a random variable since the choice of $x_i$ and $y_i$ is random. $x_i, y_i$ could be any of $\binom{n}{2}$ pairs but $Z_i$ is 1 only for the $m_e$ pairs (edges going out of $N[u] - N[S]$). Thus, $\mathbb{E}[Z_i] = \Pr(Z_i = 1) = m_e / \binom{n}{2}$. Let $\bar{Z} = \frac{1}{k} \cdot (\sum_i Z_i)$. Thus, $\mathbb{E}[\bar{Z}] = m_e / \binom{n}{2}$. Using Hoeffding's inequality (Proposition 2.2) we get:

$$\Pr\left(\left|\bar{Z} - \mathbb{E}[\bar{Z}]\right| \geq \delta\right) \leq 2 \exp\left(-2\delta^2 k\right)$$
$$\leq 2 \exp\left(-2 \cdot 10 \log n\right)$$
$$\leq n^{-10}.$$

So with high probability, we have

$$\left|\bar{Z} - \mathbb{E}\left[\bar{Z}\right]\right| \leq \delta$$

$$\frac{m_e}{\binom{n}{2}} - \delta \leq \bar{Z} \leq \frac{m_e}{\binom{n}{2}} + \delta$$

$$m_e - \binom{n}{2}\delta \leq \tilde{Z} \leq m_e + \binom{n}{2}\delta$$

proving the claim. □

The output of Algorithm 7 is $\tilde{Z} + \binom{n}{2}\delta$ implying $m_e \leq \tilde{Z} + \binom{n}{2}\delta \leq m_e + \delta n^2$ using Claim 4.6. This proves the estimation guarantee of E-Tool in Lemma 4.5. We now show that the space used is $O(1/\delta^2)$ words.

**Claim 4.7.** *The space used by Algorithm 7 is $O(1/\delta^2)$ words.*

*Proof.* In one iteration, we store a constant number of counters which takes a space of $O(1)$ bits implying that the space used by the algorithm over all iterations is $O(1/\delta^2)$ words. □

This proves the space-bound of E-Tool in Lemma 4.5.

## 4.3 Unclustered edge Sketching Tool

In this section, we describe a tool to estimate the number of edges incident on unclustered vertices given a set $S$ of cluster centers. A cluster contains the cluster center $u$ and its neighboring vertices $N(u)$. Thus, vertices that are not in $N[S]$ are called unclustered vertices and edges incident on those vertices are called unclustered edges. Consider the formal description of the problem:

**Problem 6.** The input graph $G = (V, E)$ is given in a (possibly dynamic) stream. The input also contains a set of vertices $S$, given before the stream, that represent cluster centers. The goal is to estimate the number of unclustered edges $m_u$ i.e. edges that have both endpoints outside $N[S]$ with an overestimation of at most $\delta n^2$.

**Lemma 4.8.** *There is a (dynamic) streaming algorithm that solves Problem 6 with high probability using $O(1/\delta^2)$ words of space.*

We now show the streaming algorithm of Lemma 4.8:

---

**Algorithm 8. U-Tool**

**Input:** $G = (V, E)$ in a (dynamic) stream and $S \subseteq V$ before the stream.

**Pre-Processing:**

1. Sample $k = \frac{10 \log n}{\delta^2}$ pairs of vertices $x_i, y_i$ at random

**During the Stream:**

1. For pair $(x_i, y_i)$, store the following information:

   (a) A counter $C(i)$ for the number of edges between $x_i$ and $y_i$.

   (b) A counter $C_x(i, S)$ for the number of edges between $x_i$ and $S$ and a counter $C_y(i, S)$ for the number of edges between $y_i$ and $S$.

---

**Post-Processing:**

1. For pair $i$ set $Z_i = 1$ iff $(x_i, y_i)$ is an unclustered edge and set $Z_i = 0$ otherwise.

2. Let $\tilde{Z} = \binom{n}{2} \cdot \frac{1}{k} \cdot (\sum_i Z_i)$

3. Output $\tilde{Z} + \binom{n}{2}\delta$

---

It is easy to see that the stored information is enough to check whether $(x_i, y_i)$ is an unclustered edge. First check if $(x_i, y_i)$ is an edge $C(i) > 0$. If $x_i \S$ or $y_i \in S$ then $(x_i, y_i)$ is not an unclustered edge. Finally, if at least one of $C_x(i, S), C_y(i, S)$ is 0 $(x_i, y_i)$ is an unclustered edge. We first show that $\tilde{Z}$ is a good approximation for $m_u$.

**Claim 4.9.** *With high probability, we have* $m_u - \binom{n}{2}\delta \leq \tilde{Z} \leq m_u + \binom{n}{2}\delta$.

*Proof.* $Z_i$ is a random variable since the choice of $x_i$ and $y_i$ is random. $x_i, y_i$ could be any of $\binom{n}{2}$ pairs but $Z_i$ is 1 only for the $m_u$ pairs (unclustered edges). Thus, $\mathbb{E}[Z_i] = \Pr(Z_i = 1) = m_u / \binom{n}{2}$. Let $\bar{Z} = \frac{1}{k} \cdot (\sum_i Z_i)$. Thus, $\mathbb{E}[\bar{Z}] = m_u / \binom{n}{2}$. Using Hoeffding's inequality (Proposition 2.2) we get:

$$\Pr\left(\left|\bar{Z} - \mathbb{E}[\bar{Z}]\right| \geq \delta\right) \leq 2\exp\left(-2\delta^2 k\right)$$
$$\leq 2\exp\left(-2 \cdot 10 \log n\right)$$
$$\leq n^{-10}.$$

So with high probability, we have

$$\left|\bar{Z} - \mathbb{E}[\bar{Z}]\right| \leq \delta$$
$$\frac{m_u}{\binom{n}{2}} - \delta \leq \bar{Z} \leq \frac{m_u}{\binom{n}{2}} + \delta$$
$$m_u - \binom{n}{2}\delta \leq \tilde{Z} \leq m_u + \binom{n}{2}\delta$$

proving the claim. $\qquad\square$

The output of Algorithm 8 is $\tilde{Z} + \binom{n}{2}\delta$ implying $m_u \leq \tilde{Z} + \binom{n}{2}\delta \leq m_u + \delta n^2$ using Claim 4.9. This proves the estimation guarantee of `U-Tool` in Lemma 4.8. We now show that the space used is $O(1/\delta^2)$ words.

**Claim 4.10.** *The space used by Algorithm 8 is* $O(1/\delta^2)$ *words.*

*Proof.* In one iteration, we store a constant number of counters which takes a space of $O(1)$ bits implying that the space used by the algorithm over all iterations is $O(1/\delta^2)$ words. $\qquad\square$

This proves the space-bound of `U-Tool` in Lemma 4.8. Lemmas 4.2, 4.5 and 4.8 together prove Lemma 4.1.

### 4.4 The Algorithm based on Pivot

In this section, we will show the $(3 + \gamma, \delta n^2)$-approximation algorithm for the correlation clustering value, where $\gamma < 1/2$. We will do so by simulating the Local Cluster algorithm from [BGK13]. The Local Cluster algorithm samples $1/\delta$ random vertices in a set $U$ and computes the greedy maximal

independent set (MIS) $M$ of $U$ to get the cluster centers $p_1, p_2, \ldots, p_t$. The clusters generated then are $N^+[p_i] - \cup_{j=1}^{i-1} N^+[p_j]$ and all the remaining vertices (called unclustered vertices) are clustered in their own singleton cluster. [BGK13] proved that the expected cost of this clustering is at most $3 \cdot \mathsf{OPT} + \frac{\delta}{2} n^2$. More formally, they showed the following:

**Proposition 4.11** ([BGK13])**.** *The expected cost of the clustering for Local Cluster is at most* $3 \cdot \mathsf{OPT} + \frac{\delta}{2} n^2$.

Using Proposition 4.11 we first give a $(3, \delta n^2)$ approximation in expectation.

**Theorem 3.** *There is a (dynamic) streaming algorithm that in expectation gives a $(3, \delta n^2)$-approximation for the correlation clustering value and takes space $O\left(\frac{2^{1/\delta} \cdot \log n}{\delta^5}\right)$ words.*

We simulate the Local Cluster algorithm in graph streams. During the stream, we store sketches that help us compute the cost of non-edges within clusters and edges going out of clusters in post-processing. The unclustered vertices are put in their own singleton cluster and we use another sketch to estimate the number of edges incident on the unclustered vertices. Consider the formal description of the algorithm:

---

**Algorithm 9.** Simulation of the Local Cluster Algorithm

**Input:** $G = (V, E^+ \cup E^-)$ in a (dynamic) stream

**Output:** $(3, \delta n^2)$-approximation to the correlation clustering value in expectation

**Pre-Processing:**

1. Sample a set $U$ of $1/\delta$ random vertices

2. Let $\pi$ be a random permutation of the vertices in $U$

**During the Stream:**

1. Store $(+)$ all edges between vertices of $U$

2. For all $u \in U$ and $S \subseteq U$ compute $\mathtt{NE\text{-}Tool}(u, S), \mathtt{E\text{-}Tool}(u, S), \mathtt{U\text{-}Tool}(S)$ with parameter $\delta^2/6$ using the $G^+$ subgraph.

**Post-Processing:**

1. Compute the greedy MIS $M := p_1, p_2, \ldots, p_t$ of $U$ in the order of $\pi$ (using edges stored between vertices of $U$).

2. $S_0 = \emptyset$. $\tilde{Z} = 0$.

3. For $i = 1$ to $t$:

   - $\tilde{Z} = \tilde{Z} + \mathtt{NE\text{-}Tool}(p_i, S_{i-1}) + \mathtt{E\text{-}Tool}(p_i, S_{i-1})$.
   - $S_i := S_{i-1} \cup \{p_i\}$

4. Output $Z := \tilde{Z} + \mathtt{U\text{-}Tool}(M)$

---

Note that for any $p_i \in M$, all the neighbors of $p_i$ that are not in $N^+[S_{i-1}]$ will belong to the cluster of $p_i$. Thus, if a vertex is a neighbor of two vertices in $M$ then it will belong to the cluster of the one that appears earlier in the order $\pi$. We first show that if our tools could estimate without any error we would get exactly the clustering value of the Local Cluster algorithm.

**Claim 4.12.** *The clustering value of the Local Cluster algorithm is equal to* $m_u + \sum_i m_{ne}(p_i, S_{i-1}) + m_e(p_i, S_{i-1})$.

*Proof.* The correlation clustering value is the sum of the number of non-edges within clusters and the number of edges across clusters. We first argue that the number of non-edges within clusters is exactly $\sum_i m_{ne}(p_i, S_{i-1})$.

$m_{ne}(u, S)$ is the number of non-edges within $N^+[u] - N^+[S]$. We will show that a non-edge within a cluster is counted in exactly one term in the summation. Consider any non-edge $(x, y)$ within a cluster. Let $(x, y)$ be in the cluster of $p_i$. This means that $x$ and $y$ are not neighbors of $p_1, p_2, \ldots, p_{i-1}$ because if they were then they would be in a previous cluster and not the cluster of $p_i$. So $(x, y)$ is not counted in $m_{ne}(p_\ell, S_{\ell-1})$ for $\ell < i$. Also, it is counted in $m_{ne}(p_i, S_{i-1})$ since $x$ and $y$ are neighbors of $p_i$ but not of $p_1, p_2, \ldots, p_{i-1}$. Finally, $(x, y)$ is not counted in $m_{ne}(p_\ell, S_{\ell-1})$ for $\ell > i$ because $x, y$ are in $N^+[p_i] \subseteq N^+[S_\ell]$. Also, $m_{ne}$ only counts non-edges within clusters implying that the number of non-edges within clusters is exactly $\sum_i m_{ne}(p_i, S_{i-1})$.

The number of edges across clusters can be divided into two disjoint categories. The number of edges going out of clusters of $p_i$'s and the unclustered edges i.e. the edges between two unclustered vertices. $m_u$ is the number of unclustered edges. Using an almost identical argument to the above we can show that the number of edges going out of clusters of $p_i$'s is exactly $\sum_i m_e(p_i, S_{i-1})$.

Therefore, the sum of these gives the exact correlation clustering value of the Local Cluster algorithm. $\qquad\square$

We now show that Algorithm 9 is a $(3, \delta n^2)$ approximation in expectation.

**Claim 4.13.** *The output $Z$ of Algorithm 9 is at least* OPT *with high probability and is at most* $3\mathsf{OPT} + \delta n^2$ *in expectation.*

*Proof.* Using Claim 4.12 we know that if the tools worked with no error, Algorithm 9 would give the exact clustering cost of Local Cluster. Also, we know that the tools do not underestimate and overestimate by $\delta^2 n^2/6$ with high probability (Lemma 4.1) so we can condition on the high probability events in Lemmas 4.2, 4.5 and 4.8. We first note that $Z$ is at least OPT because of the above conditioning and the fact that the clustering cost of Local Cluster is at least OPT. Using Proposition 4.11 we know that the expected cost of the clustering when choosing $\frac{1}{\delta}$ random pivots is at most $3\mathsf{OPT} + \frac{\delta}{2}n^2$. Therefore, the clustering cost of Algorithm 9 is between OPT and $3\mathsf{OPT} + \frac{\delta}{2}n^2$ plus the overestimate.

We now calculate the overestimate. We use parameter $\delta^2/6$ for the tools thus the additive error in each tool is at most $\delta^2 n^2/6$. There are at most $1/\delta$ pivots implying a total additive error of at most $\delta n^2/3$ over all the copies of NE-Tool and E-Tool. U-Tool has an error of at most $\delta^2 n^2/6$ implying that the overall error is at most $\delta n^2/2 + \delta n^2/3 + \delta^2 n^2/6 \leq \delta n^2$ giving a $(3, \delta n^2)$ approximation in expectation.

We note that we condition on the high probability events for all copies of the tools and union bound over the failure probabilities. Thus, for the overall failure probability to be small we need $2^{1/\delta} \leq poly(n)$. $\qquad\square$

We now show the space-bound of Algorithm 9.

**Claim 4.14.** *The space of Algorithm 9 is* $O\left(\frac{2^{1/\delta} \cdot \log n}{\delta^5}\right)$ *words.*

*Proof.* Each copy of `NE-Tool`, `E-Tool` and `U-Tool` with parameter $\delta^2/6$ takes $O(\log n/\delta^4)$ words of space, and we compute the tools for all $v \in U$ and $S \subseteq U$. Thus, the space used is $O(2^{1/\delta} \cdot \log n/\delta^5)$ words. Storing the $(+)$ edges between vertices in $U$ takes space at most $1/\delta^2$ words which is a lower order term. $\square$

Claim 4.13 and Claim 4.14 together prove Theorem 3. We now prove Theorem 2. We run Algorithm 9 $\frac{60 \log n}{\gamma}$ times and let $Z_{min}$ be the minimum cost over all iterations. $Z_{min}$ is a $(3 + \gamma, \frac{7}{6}\delta n^2)$-approximation with high probability.

**Claim 4.15.** $\mathsf{OPT} \leq Z_{min} \leq (3 + \gamma)\mathsf{OPT} + \frac{7}{6}\delta n^2$ *with high probability.*

*Proof.* Using Claim 4.13 we know that the clustering cost of Algorithm 9 is at least $\mathsf{OPT}$ with high probability, so we condition on this event for all parallel repetitions implying that $\mathsf{OPT} \leq Z_{min}$. By Claim 4.13 we also know that the clustering cost of Algorithm 9 is at most $3\mathsf{OPT} + \delta n^2$ in expectation. Thus, we can run this algorithm $\frac{60 \log n}{\gamma}$ times in parallel and take the minimum cost as the best estimate. Using Markov's inequality, the probability that the true cost exceeds $(1 + \gamma/3) \cdot (3\mathsf{OPT} + \delta n^2) \leq (3 + \gamma)\mathsf{OPT} + (1 + 1/6)\delta n^2$ is at most $1/(1 + \gamma/3)$. We can boost the probability of success by repeating $\frac{60 \log n}{\gamma}$ times:

$$\Pr(\text{failure}) \leq \left(\frac{1}{1 + \gamma/3}\right)^{\frac{60 \log n}{\gamma}}$$
$$\leq \exp(-\gamma/6)^{\frac{60 \log n}{\gamma}} \qquad (1 + x \geq \exp(x/2) \text{ for } x \in [0, 1])$$
$$= n^{-10}.$$

Therefore, $Z_{min}$ is at most $(3 + \gamma)\mathsf{OPT} + \frac{7}{6}\delta n^2$ with high probability. $\square$

Thus, we get a $(3 + \gamma, \frac{7}{6}\delta n^2)$-approximation with high probability. We now prove Theorem 2.

*Proof of Theorem 2.* Each parallel repetition of Algorithm 9 takes $O\left(\frac{2^{1/\delta} \cdot \log n}{\delta^5}\right)$ words of space. Repeating $O(\frac{\log n}{\gamma})$ times and re-scaling $\delta$ by a factor of $\frac{7}{6}$ gives a total space-bound of $O\left(\frac{2^{7/6\delta} \cdot \log^2 n}{\gamma \cdot \delta^5}\right)$ words. $\square$

# 5 A Lower Bound for $O(n^{2-\varepsilon})$ Additive error

In this section, we show that if we *only* allow additive error, any streaming algorithm with poly-logarithm memory cannot cross an error barrier of $\Omega(n^{2-\varepsilon})$ for any constant $\varepsilon$. Here, and throughout, we will refer to this lower bound as the *almost-quadratic lower bound*. The lower bound is weaker than the linear lower bound of Section 6 in terms of the multiplicative factor since it only works for $c = 1$. However, it is much stronger in the additive sense: the upper bounds obtained by our algorithms are $O(n^2)$, and the almost-quadratic lower bound matches this term up to an $O(n^\varepsilon)$ factor – this provides a strong justification of the additive error in our algorithms.

The formal statement of the almost-quadratic lower bound is as follows.

**Theorem 4.** *There exists a constant $C$, such that any single-pass streaming algorithm that estimates the optimal value $\mathsf{OPT}$ of correlation clustering by a $C \cdot n^{2-\varepsilon}$ purely additive error (i.e., an estimated value that is at most $\left(\mathsf{OPT} + C \cdot n^{2-\varepsilon}\right)$) with probability at least $\frac{99}{100}$ has to use a memory of $\Omega(n^\varepsilon)$ bits, even on labeled complete graphs.*

Note that Theorem 4 does *not* require the stream to be dynamic, which is in contrast to our upper bound results that work for dynamic streams. We obtain the almost-quadratic lower bound by a new reduction from the INDEX problem. On the high level, the instance we construct 'hides' an $\Omega(n^{2-\varepsilon})$ gap between the yes and no cases inside a case-invariant $\Omega(n^2)$ cost. The reduction can be viewed as a more involved variant of the space lower bound for the *exact* correlation clustering in a very recent work [AAD+23]. In a nutshell, we modify their construction to 'boost' the gap between yes and no cases, and apply a new trick to separate the *values* of clustering[4].

We now start the formal reduction proof with the following variant of INDEX.

**Problem 7.** Consider a two-player communication problem, where Alice is given a matrix $M \in \{0,1\}$ and Bob is given $i^\star, j^\star \sim [N] \times [N]$. Alice sends a message to Bob, and Bob outputs the value of $M[i^\star, j^\star]$.

It is not hard to prove that Problem 7 requires $\Omega(N^2)$ bits of communication for Bob to output the correct $M[i^\star, j^\star]$ with probability at least $\frac{99}{100}$. We shall now construct an instance that creates the desired gap from Problem 7.

## 5.1 A construction of correlation clustering structural

To continue, we introduce a subroutine for Alice and Bob to construct vertices with a 2-clustering structure, such that for the optimal clustering, Alice's edge will only affect the assignment of one fixed group of vertices. Crucially, the optimal and second-best clustering assignments of the instance only differ on this special group of vertices.

As mentioned, we use a simpler construction in [AAD+23] as our prototype. We start introducing the formal construction with the hard-coded inputs. We insist that the input parameter $N$ is odd in our construction.

---

**Structural hard-coded inputs: An input-invariate sub-graph**

1. Both player create $3C$ vertices for each index $i \in [N]$, where $C$ is a parameter.

2. Connect each collection of $C$ vertices with $(+)$ edges, i.e. make them cliques of size $C$.

3. For each index $i$, put the 3 corresponding collections into 3 groups: $L$, $R^{(1)}$, and $R^{(2)}$. We let $L(i)$ denote the collection of vertices correspond to $(i)$ in $L$, and define $R^{(1)}(i)$ and $R^{(2)}(i)$ analogously.

4. Create $100 \cdot CN$ vertices and name this new group $V$. Divide $V$ into two equal-size partitions of size $50 \cdot CN$ each, and name them $V_\uparrow$ and $V_\downarrow$.

5. Add $(+)$ edges between all vertex pairs inside $V_\uparrow \times V_\uparrow$ and inside $V_\downarrow \times V_\downarrow$; add $(-)$ edges between vertex pairs of $V_\uparrow \times V_\downarrow$.

---

With the structural hard-coded inputs, we can give our complete construction as follows.

---

Dense-Two-Clusters: **A family of graphs from INDEX**

1. Add the structural hard-coded inputs, prescribed as below, to obtain $L$, $R^{(1)}$, $R^{(2)}$, $V_\uparrow$, $V_\downarrow$, and

---

[4]Both ideas are somehow discussed separately by [AAD+23], but it was far from clear whether the two strategies can work together to obtain the desired lower bound – this is a main technical work in our proof.

the corresponding edges.

2. Alice edges:

    (a) For each index $(i,j)$, Alice adds $2 \cdot C^2$ edges as follows:

    - If $M[i,j] = 0$, Alice adds $(+)$ edges between every vertex pairs in $L(i)$ and $R^{(1)}(j)$, and $(-)$ edges between every vertex pairs in $L(i)$ and $R^{(2)}(j)$.
    - Otherwise, if $M[i,j] = 1$, Alice adds $(+)$ edges between every vertex pairs in $L(i)$ and $R^{(2)}(j)$, and $(-)$ edges between every vertex pairs in $L(i)$ and $R^{(1)}(j)$.

3. Bob edges:

    (a) For every index other than $i^*$, Bob divides the vertices in $L$ to $L_\uparrow$ and $L_\downarrow$, such that

    - All vertices inside $L_\uparrow$ and inside $L_\downarrow$ are connected with $(+)$ edges; all vertex pairs in $L_\uparrow \times L_\downarrow$ are connected with $(-)$ edges.
    - Vertices in $L_\uparrow$ are connected to *all vertices* in $V_\uparrow$ with $(+)$ edges and to *all vertices* in $V_\downarrow$ with $(-)$ edges.
    - Vertices in $L_\downarrow$ are connected to *all vertices* in $V_\downarrow$ with $(+)$ edges and to *all vertices* in $V_\uparrow$ with $(-)$ edges.

    (b) For every index other than $j^*$, Bob divides the vertices in $R^{(1)}$ to $R^{(1)}_\uparrow$ and $R^{(1)}_\downarrow$ and $R^{(2)}$ to $R^{(1)}_\uparrow$ and $R^{(2)}_\downarrow$ with the *same* lexigraphical order.

    (c) Bob connects all vertices in $R^{(1)}(j^*)$ with $(+)$ edges to $V_\downarrow$, $R^{(1)}_\downarrow$, and $R^{(2)}_\downarrow$, and $(-)$ edges to $V_\uparrow$, $R^{(1)}_\uparrow$, and $R^{(2)}_\uparrow$.

    (d) Bob connects all vertices in $R^{(2)}(j^*)$ with $(+)$ edges to $V_\uparrow$, $R^{(1)}_\uparrow$ and $R^{(2)}_\uparrow$, and $(-)$ edges to $V_\downarrow$, $R^{(1)}_\downarrow$, $R^{(2)}_\downarrow$.

    (e) Bob connects vertices in $L(i^*)$ with $(+)$ edges to every vertices other than $R^{(1)}$ and $R^{(2)}$.

    (f) Bob connect $(-)$ edges between all vertex pairs in $(L_\uparrow \times R^{(1)}_\downarrow)$, $(L_\uparrow \times R^{(2)}_\downarrow)$, $(L_\downarrow \times R^{(1)}_\uparrow)$, and $(L_\downarrow \times R^{(2)}_\uparrow)$; furthermore, Bob connects $(+)$ edges between all vertex pairs in $(L_\uparrow \times R^{(1)}_\uparrow)$, $(L_\uparrow \times R^{(2)}_\uparrow)$, $(L_\downarrow \times R^{(1)}_\downarrow)$, and $(L_\downarrow \times R^{(2)}_\downarrow)$.

It is straightforward to verify that the graph constructed by Alice and Bob is complete and every edge has a label of either $(+)$ or $(-)$. In what follows, we prove that Dense-Two-Clusters has a neat structure that the optimal clustering always contains exactly 2 clusters, and the optimal and second-optimal clusterings differ only by the cluster of $L(i^*)$. More concretely, we have the following lemma.

**Lemma 5.1.** *The optimal clustering of* Dense-Two-Clusters *contains two clusters:*

- *The first cluster (the* up *cluster) contains the vertices of $L_\uparrow, V_\uparrow, R^{(1)}_\uparrow, R^{(2)}_\uparrow$ (including $R^{(2)}(j^*)$);*

- *The second cluster (the* down *cluster) contains the vertices of $L_\downarrow, V_\downarrow, R^{(1)}_\downarrow, R^{(2)}_\downarrow$ (including $R^{(1)}(j^*)$).*

*Furthermore, the remaining vertex $L(i^*)$ is in the up cluster if $M[i^*, j^*] = 1$, and is in the down cluster if $M[i^*, j^*] = 0$. The second-optimal clustering differs from the optimal clustering by switching $L(i^*)$ to the other cluster, and the cost differs by 2.*

Lemma 5.1 was originally proved by [AAD⁺23]. We give a self-contained proof in our paper, and our proof strategy turns out to be technically more applicable for our purposes in later steps. To begin with, we define the notion of up vertices and down vertices by their edge connectivity to $V_\uparrow$ as follows.

**Definition 7** (Up and down vertices). Fix a graph $G = (L \cup R^{(1)} \cup R^2 \cup V, E)$ sampled from Dense-Two-Clusters, we say a vertex $v \neq L(i^*)$ is an *up vertex* if and only if $|E^+(v, V_\uparrow)| \geq |E^-(v, V_\uparrow)|$, i.e. among the vertices in $V_\uparrow$, $v$ has more $(+)$ edges in $V_\uparrow$ than $(-)$ edges. We say $v$ is a *down vertex* otherwise.

Note that the above definition also covers $R^{(1)}(j^\star)$ as a down vertex and $R^{(2)}(j^\star)$ as an up vertex. Intuitively, a vertex $v$ becomes an up vertex if it is densely connected to the 'up side' with $(+)$ edges, and vice versa. The up and down vertices have very 'local' properties: it has many $(+)$ edges to one side and many $(-)$ edges to the other. More formally, we characterize this property as the following condition.

**Condition 1.** We say two groups of vertices, namely the up and down vertices, satisfy the $\mathsf{Cond}^*$ condition if there exists a global constant $N$ such that:

1. For up-up and down-down vertex pairs, there are

   (a) Every up vertex has no $(-)$ edges and at least $50N$ $(+)$ edges connecting to vertices in $R_\uparrow^{(1)}, R_\uparrow^{(2)}, V_\uparrow$, and at most $N$ $(-)$ edges connecting other up vertices.
   (b) Every down vertex has no $(-)$ edges and at least $50N$ $(+)$ edges connecting to vertices in $R_\downarrow^{(1)}, R_\downarrow^{(2)}, V_\downarrow$, and at most $N$ $(-)$ edges connecting other down vertices.

2. For up-down vertex pairs, there are

   (a) Every up vertex has no $(+)$ edges and at least $50N$ $(-)$ edges connecting to vertices in $R_\downarrow^{(1)}, R_\downarrow^{(2)}, V_\downarrow$, and at most $N$ $(+)$ edges connecting other down vertices.
   (b) Every down vertex has no $(+)$ edges and at least $50N$ $(-)$ edges connecting to vertices in $R_\uparrow^{(1)}, R_\uparrow^{(2)}, V_\uparrow$ and at most $N$ $(+)$ edges connecting other up vertices.

We observe that the graphs sampled from Dense-Two-Clusters satisfy the $\mathsf{Cond}^*$.

**Observation 5.2.** $\mathsf{Cond}^*$ *in Condition 1 holds in any graph sampled from* Dense-Two-Clusters.

We now use Condition 1 and Observation 5.2 to characterize the structure of the optimal clustering. In particular, we first prove that with Condition 1, there can be at most one cluster consisting of only up vertices and similarly for down vertices.

**Claim 5.3.** *Assuming* $\mathsf{Cond}^*$ *in Condition 1, there can be at most one cluster consisting of only up vertices (similarly for down vertices).*

*Proof.* Assume towards a contradiction that there are clusters $C_1$ and $C_2$ consisting of only up vertices. Create a new clustering by merging $C_1$ and $C_2$ into a new cluster $C$. We observe that merging into $C$ does not change the cost induced by the edges with an endpoint in $V - C$. By line 1 of Condition 1, by merging $C_1$ and $C_2$ into $C$, the cost of the new clustering strictly decreases. Thus, the original clustering was not optimal giving a contradiction. Therefore, there can be at most one cluster consisting of only up vertices. The proof for down vertices is analogous. $\square$

Using Claim 5.3 together with Condition 1, we can show that any optimal clustering cannot contain put up and down vertices into the same cluster.

**Claim 5.4.** *Assuming* Cond* *in Condition 1, a cluster in the optimal clustering cannot contain both up vertices and down vertices.*

*Proof.* We first prove the statement for vertices among $V \cup R^{(1)} \cup R^{(2)}$ and vertices among $L$, respectively. Assume towards a contradiction that there is a cluster $C$ that contains both up vertices and down vertices among $V \cup R^{(1)} \cup R^{(2)}$. Create a new clustering by splitting $C$ into clusters $C_\uparrow - L$ containing the up vertices and $C_\downarrow - L$ containing the down vertices. We observe that splitting $C$ does not change the cost induced by the edges with an endpoint in $V - C$. By line 2 of Condition 1, there is a $(-)$ edge from every vertex in $C_\uparrow - L$ to $C_\downarrow - L$ (for vertices in $V \cup R^{(1)} \cup R^{(2)}$), and the cost of the new clustering strictly decreases. Thus, the original clustering was not optimal giving a contradiction. Therefore, a cluster in the optimal clustering cannot contain both up vertices and down vertices for vertices in $V \cup R^{(1)} \cup R^{(2)}$. With the same argument as above, we can prove that the statement holds for vertices among $L$.

We now proceed to prove the statement for all vertices. By Claim 5.3 and the result on vertices among $V \cup R^{(1)} \cup R^{(2)}$, the vertices among $V_\uparrow \cup R_\uparrow^{(1)} \cup R_\uparrow^{(2)}$ must be clustered together, and we let this cluster be $C_\uparrow - L$. We assume towards a contradiction that there is a cluster $C$ that contains both up vertices and down vertices among $L_\downarrow \cup C_\uparrow - L$. As such, we can split $C$ into clusters $C_\uparrow - L$ and vertices in $L_\downarrow$ without affecting the cost induced by the edges with an endpoint in $V - C$. Fix a vertex $v \in L_\downarrow$, although there might be at most $N$ $(+)$ edges between $v$ and $C_\uparrow - L$, there are at least $50N$ $(-)$ edges that leads to a strict cost decrement by this split. As such, we can obtain a contradiction, which proves that all up and down vertices cannot stay in the same cluster in the optimal clustering. $\square$

Using Claim 5.3, Claim 5.4 together with Observation 5.2, we can conclude that the optimal clustering for Dense-Two-Clusters has at most three clusters; one containing $L(i^*)$ ($C_{i^*}$), another containing only up vertices ($C_\uparrow := L_\uparrow \cup V_\uparrow \cup R_\uparrow^{(1)} \cup R_\uparrow^{(2)}$) and the last containing only down vertices ($C_\downarrow := L_\uparrow \cup V_\downarrow \cup R_\downarrow^{(1)} \cup R_\downarrow^{(2)}$). We will now use the edges of $L(i^*)$ to conclude that $L(i^*)$ must be clustered together with either $C_\uparrow$ or $C_\downarrow$.

**Claim 5.5.** *There are exactly 2 clusters in the optimal clustering, and vertex $L(i^*)$ must be clustered either with $C_\uparrow$ or $C_\downarrow$.*

*Proof.* We first argue that the number of clusters is $> 1$. Consider that there is just one cluster $C$. Remove all the up vertices and move them to a new cluster $C_\uparrow$. The decrease in the cost is at least $(50N)^2$ because of the non-edges between $V_\uparrow$ and $V_\downarrow$. The increase in the cost is at most $103N$ because of the edges between $i^*$ and the up vertices. There is a net decrease in cost and thus, the number of clusters is $> 1$. We now assume towards a contradiction that there are 3 clusters $C_{i^*}, C_\uparrow, C_\downarrow$.

We first consider the case where $C_{i^*}$ contains only vertex $i^*$. Move $i^*$ to $C_\uparrow$. The increase in the cost is at most $N$ due to the non-edges to some up vertices. But the decrease in the cost is at least $50N$ because of edges to $V_\uparrow$. Thus, we get a contradiction in this case.

We now consider the case where $C_{i^*}$ also contains vertices other than $i^*$. Let there be $u$ up vertices and $d$ down vertices in $C_{i^*}$ and wlog assume $u > 0$. Move the up vertices in $C_{i^*}$ to $C_\uparrow$. The decrease in cost is $u \cdot d + u \cdot |C_\uparrow|$ where the first term is from the non-edges between the up and down vertices in $C_{i^*}$ and the second term is from the edges between the up vertices in $C_{i^*}$ and $C_\uparrow$. The increase in cost is at most $u$ which is only due to edges between $i^*$ and the up vertices. So there the net decrease in cost is $u \cdot d + u \cdot |C_\uparrow| - u > 0$ (even when $d = 0$). $\square$

**Finalizing the proof of Lemma 5.1.** By Claim 5.3, Claim 5.4, and Claim 5.5, there are exactly two clusters $C_\uparrow$ (containing all up vertices) and $C_\downarrow$ (containing all down vertices) and $i^*$ is a part of one of these depending on its edges to $j_\uparrow^*$ and $j_\downarrow^*$. As such, we can observe $L(i^*)$ joins $C_\uparrow$ if it has $(+)$ edge to $R_\uparrow^{(2)}(j^*)$ and $(-)$ edge to $R_\uparrow^{(1)}(j^*)$, and vice versa. The two scenarios corresponds exactly to $M[i^*, j^*] = 1$ and $M[i^*, j^*] = 0$. Finally, note that if we move the assignment of $L(i^*)$, the cost increases exactly 2, while the cost increases by at least $48N$ if we move any other vertex. As such, the second-best clustering is to keep $C_\uparrow$ and $C_\downarrow$, and switch $L(i^*)$ to the other cluster.

## 5.2 Multi-copy correlation clustering structure

We now slightly tweak our original model by introducing 'duplicates' for each vertex in the Dense-Two-Clusters distribution. In particular, for each vertex in $L$, $R^{(1)}$, $R^{(2)}$ and $V$, we make it as a group of $K$ vertices for some $K \geq 1$ which we specify later. In other works, for each $L(i)$ (resp. $R^{(1)}(i)$, $R^{(2)}(i)$, $V(i)$) in the Dense-Two-Clusters family, we have a group $\mathcal{L}(i)$ (resp. $\mathcal{R}^{(1)}(i)$, $\mathcal{R}^{(2)}(i)$, $\mathcal{V}(i)$) of $K$ vertices. Inside each group of vertices, we connect the vertices as a clique of $(+)$ edges; between different groups, we add $K^2$ edges with the same label as the single labeled edge in Dense-Two-Clusters. We will eventually argue that the best clustering of such a family remains a 2-cluster structure, and the cost gap between the best and the second-best clustering becomes $O(K)$.

We now give a formal description of the augmented family of instances as follows.

---

Duplicate-Dense-Two-Clusters($K$): **A family of graphs from INDEX**

1. Alice and Bob samples a graph $G'$ from Dense-Two-Clusters.

2. For each vertex $L(i)$, the players make a group $\mathcal{L}(i)$ of $K$ copies of vertices. In the same manner, they make each vertex in $R^{(1)}(i)$, $R^{(2)}(i)$ and $V(i)$ to groups $\mathcal{R}^{(1)}(i)$, $\mathcal{R}^{(2)}(i)$ and $\mathcal{V}(i)$ with $K$ vertices in each group.

3. Fix any group $\mathcal{A}$ and two vertices $(u, v) \in \mathcal{A}$, add a $(+)$ edge between $u$ and $v$.

4. For any two groups $\mathcal{A}$ and $\mathcal{B}$ such that $u \in \mathcal{A}$ and $v \in \mathcal{B}$, add an edge whose label is consistent with the edge between the two vertices in $G'$ that induces the groups.

---

Let $G$ be the resulting group sampled from Duplicate-Dense-Two-Clusters($K$), we now prove that the new family of instances retains the two-cluster structure. To proceed formally, we introduce new notation of $\mathcal{L}$, $\mathcal{R}^{(1)}$, $\mathcal{V}$, $\mathcal{R}^{(2)}$ to denote the set of the vertices of all groups induced by the vertices in $L$, $R^{(1)}$, $V$, and $R^{(2)}$. In the same manner, we use $\mathcal{L}_\uparrow$, $\mathcal{R}_\uparrow^{(1)}$, $\mathcal{V}_\uparrow$, $\mathcal{R}_\uparrow^{(2)}$ (resp. $\mathcal{L}_\downarrow$, $\mathcal{R}_\downarrow^{(1)}$, $\mathcal{V}_\downarrow$, $\mathcal{R}_\downarrow^{(2)}$) to denote the vertices in the groups induced by $L_\uparrow, V_\uparrow, R_\uparrow^{(1)}, R_\uparrow^{(2)}$ (resp. $L_\downarrow, V_\downarrow, R_\downarrow^{(1)}, R_\downarrow^{(2)}$). The structure of the optimal clustering can be characterized as follows.

**Lemma 5.6.** *The optimal clustering of* Duplicate-Dense-Two-Clusters($K$) *for any integer $K \geq 1$ contains two clusters:*

- *The first cluster (the* up *cluster) contains the vertices of $\mathcal{L}_\uparrow, \mathcal{V}_\uparrow, \mathcal{R}_\uparrow^{(1)}, \mathcal{R}_\uparrow^{(2)}$ (including $\mathcal{R}^{(2)}(j^*)$);*

- *The second cluster (the* down *cluster) contains the vertices of $\mathcal{L}_\downarrow, \mathcal{V}_\downarrow, \mathcal{R}_\downarrow^{(1)}, \mathcal{R}_\downarrow^{(2)}$ (including $\mathcal{R}^{(1)}(j^*)$).*

*Furthermore, all of the remaining vertices of $\mathcal{L}(i^*)$ is in the up cluster if $M[i^*, j^*] = 1$, and is in the down cluster if $M[i^*, j^*] = 0$.*

Towards the proof of Lemma 5.6, we first show that the optimal clustering for Duplicate-Dense-Two-Clusters($K$) follows the same two-cluster structure on $(\mathcal{L}_\uparrow, \mathcal{V}_\uparrow, \mathcal{R}_\uparrow^{(1)}, \mathcal{R}_\uparrow^{(2)})$ and $(\mathcal{L}_\downarrow, \mathcal{V}_\downarrow, \mathcal{R}_\downarrow^{(1)}, \mathcal{R}_\downarrow^{(2)})$ as happened for Dense-Two-Clusters.

**Claim 5.7.** *In the optimal clustering, there exists a cluster $C_1$ that contains all vertices in $\mathcal{L}_\uparrow, \mathcal{V}_\uparrow, \mathcal{R}_\uparrow^{(1)}, \mathcal{R}_\uparrow^{(2)}$ and a cluster $C_2$ that contains all vertices in $\mathcal{L}_\downarrow, \mathcal{V}_\downarrow, \mathcal{R}_\downarrow^{(1)}, \mathcal{R}_\downarrow^{(2)}$.*

*Proof.* The claim is a natural generalization of Claim 5.3 and Claim 5.4. Similar to Definition 7, we define the up vertices as the vertices (except from the group $\mathcal{L}(i^*)$) whose $(+)$ edges to $\mathcal{L}_\uparrow$ is more than the $(-)$ edges. Note that by copying each vertex $K$ times and adding edges with the rules in Lines 3 and 4, we never add $(+)$ edges between the up vertices and down vertices among the vertices in $\mathcal{R}^{(1)}$, $\mathcal{R}^{(2)}$, and $\mathcal{V}$. Furthermore, for each vertex $u$ in $\mathcal{L}(i)$ for any $i \neq i^*$, suppose w.log. that the vertex is an up vertex, we add at most $K \cdot N$ $(+)$ edges to other down vertices and at most $K \cdot N$ $(-)$ edges to other up vertices. As such, $\mathsf{Cond}^*$ in Condition 1 holds with global constant $K \cdot N$. As such, we can apply Claim 5.3 and Claim 5.4 to obtain the desired claim. $\square$

We now need to handle the clustering of the vertices among the group of $\mathcal{L}(i^*)$, $\mathcal{R}^{(1)}(j^*)$ and $\mathcal{R}^{(2)}(j^*)$. We show that the vertices should be clustered together rather than being split apart, which is sufficient to prove Lemma 5.6 by further applying Claim 5.5.

**Claim 5.8.** *In the optimal clustering of* Duplicate-Dense-Two-Clusters($K$) *for any integer $K \geq 1$, all vertices inside $\mathcal{L}(i^*)$ are in the same cluster.*

*Proof.* Assume towards a contradiction that there are at least 2 clusters with vertices from $\mathcal{L}(i^*)$ (in the optimal clustering). Consider clusters $C_1$ and $C_2$ with multiple vertices $u_1, u_2, \cdots, u_m$ from $\mathcal{L}(i^*)$. Let $I_1$ be all vertices from $\mathcal{L}(i^*)$ in cluster $C_1$ and let $R_1$ be the remaining vertices in $C_1$. Similarly, define $I_2$ and $R_2$ as the vertices in cluster $C_2$ that are inside and outside $\mathcal{L}(i^*)$. We observe that if we move vertices between $C_1$ and $C_2$ the cost to vertices with endpoint in $V - (C_1 \cup C_2)$ remains unchanged. Consider any vertex $u$ in $\mathcal{L}(i^*)$ and calculate the cost $c_1$ of clustering it in $C_1$ with respect to $R_1, R_2$, i.e. the number of $(+)$ edges to $R_2$ plus the number of $(-)$ edges to $R_1$. Similarly calculate the cost $c_2$ of clustering it in $C_2$ with respect to $R_1, R_2$. Suppose w.log. that $c_1 \leq c_2$, we can create a new clustering by moving all copies of $i^*$ in $C_2$ to $C_1$. Since $c_1 \leq c_2$ the cost to $R_1 \cup R_2$ can only decrease (for each $u$ of $\mathcal{L}(i^*)$ moved). Also, the cost induced by the vertex pairs $u_1 \in I_1$ and $u_2 \in I_2$ for any $u_1$ and $u_2$ goes to 0. Thus, the clustering cost strictly decreases, which forms a contradiction to the optimality of the clustering. $\square$

**Finalizing the proof of Lemma 5.6.** By Claim 5.7, there are exactly two clusters $C_\uparrow$ (containing all up vertices) and $C_\downarrow$ (containing all down vertices) and *all* vertices in $\mathcal{L}(i^*)$ are part of one of the clusters depending on its edges to $j_\uparrow^*$ and $j_\downarrow^*$. As such, we can observe vertices of $\mathcal{L}(i^*)$ join $C_\uparrow$ if it has $(+)$ edge to $R_\uparrow^{(2)}(j^*)$ and $(-)$ edge to $R_\uparrow^{(1)}(j^*)$, and vice versa. As such the optimal clustering structure as prescribed in Lemma 5.6 is obtained.

## 5.3  Cost Testing Lower Bound – A Variate of the Hard Instance

So far, we have not discussed the separation of costs in the yes and no instances. Indeed, one can observe that although the optimal clustering changes in a regulated manner depending on the values of $M[i^\star, j^\star]$, the *cost* of the optimal clusters are the same. We now apply a trick to 'tweak' the construction in Section 5.1 such that the clustering structure remains except vertices in the special group $(i^\star, j^\star)$. Such a trick was first employed in [AAD+23]; however, the trick was only applied to a single vertex in their case, and the analysis is much more straightforward. Our technical contribution here is to show that the trick applies even with multiple copies of vertices in each group.

We now introduce the value-testing trick as an algorithm that given an oracle for correlation clustering with $o(n^{2-\Omega(1)})$ (purely) additive error, returns the correct answer of INDEX:

---

**CC-INDEX-ALG: An algorithm for INDEX given Tool-CC$(\varepsilon)$ – an oracle that returns the optimal cost of correlation clustering with $(1, \frac{1}{10^6} \cdot n^{2-\varepsilon})$ error**

1. Alice and Bob construct a graph $G$ from Duplicate-Dense-Two-Clusters$(K)$ with $K = N^{\frac{1}{\varepsilon}-1}$ for arbitrarily small $\varepsilon > 0$.

2. The players run two subroutines:

   (a) In the first subroutine, the players run **Tool-CC$(\varepsilon)$** on graph $G$, and obtain $\mathsf{cost}_1$.

   (b) In the second subroutine, the players run **Tool-CC$(\varepsilon)$** on graph $G'$ as follows:

      i.   Arbitrarily pick $10CN$ vertices from $\mathcal{V}_\downarrow$ on $G$, and name the set as $\mathcal{V}'_\downarrow$.

      ii.  For each vertex $u \in \mathcal{L}(i^\star)$, change the $(+)$ edge between $u$ and $\mathcal{V}'_\downarrow$ to a $(-)$ edge (note that Bob exclusively controls all edges incident on $\mathcal{V}$ so he can do this).

      iii. The players obtain the cost $\mathsf{cost}_2$.

3. If $\mathsf{cost}_1 - \mathsf{cost}_2 \geq 10K^2N - \frac{1}{25}K^2$, return $M[i^\star, j^\star] = 1$; otherwise, return $M[i^\star, j^\star] = 0$.

---

We will eventually show that CC-INDEX-ALG succeeds with the same probability as **Tool-CC$(\varepsilon)$**, which is sufficient to establish the reduction and prove Theorem 4. To this end, we will prove the optimal clustering structure for both subroutines in both cases, and show that the cost gap is greater than $10K^2N - \frac{1}{25}K^2$ if and only if $M[i^\star, j^\star] = 1$. We start with observing the basic clustering structure is preserved in $G'$ for the second subroutine.

**Observation 5.9.** *In the optimal clustering for both $G$ and $G'$, there exists a cluster $C_1$ that contains all vertices in $\mathcal{L}_\uparrow, \mathcal{V}_\uparrow, \mathcal{R}_\uparrow^{(1)}, \mathcal{R}_\uparrow^{(2)}$ and a cluster $C_2$ that contains all vertices in $\mathcal{L}_\downarrow, \mathcal{V}_\downarrow, \mathcal{R}_\downarrow^{(1)}, \mathcal{R}_\downarrow^{(2)}$, and the vertices in $\mathcal{L}(i^\star)$ are in the same clusters.*

*Proof.* The property holds for $G$ simply by Claim 5.7 and Claim 5.8. Furthermore, only edges incident on $\mathcal{L}(i^\star)$ change in $G'$, which does not affect the correctness of Claim 5.7. Finally, we never used any property for edges with endpoint(s) outside $\mathcal{L}(i^\star)$, $\mathcal{R}^{(1)}(j^\star)$ and $\mathcal{R}^{(2)}(j^\star)$ in the proof of Claim 5.8; as such, the property in Claim 5.8 still holds. $\qquad\square$

Based on Observation 5.9, we now establish the optimal clustering for the graph $G'$ in the second subroutine for both the yes and no cases. We can then analyze the cost difference therein.

**Lemma 5.10.** *The optimal clustering of $G'$ for $M[i^\star, j^\star] = 0$ contains two clusters:*

- *The first cluster (the up cluster) contains the vertices of $\mathcal{L}_\uparrow, \mathcal{V}_\uparrow, \mathcal{R}_\uparrow^{(1)}, \mathcal{R}_\uparrow^{(2)}$ (including $\mathcal{R}^{(2)}(j^*)$) and $\mathcal{L}(i^*)$;*

- *The second cluster (the down cluster) contains the vertices of $\mathcal{L}_\downarrow, \mathcal{V}_\downarrow, \mathcal{R}_\downarrow^{(1)}, \mathcal{R}_\downarrow^{(2)}$ (including $\mathcal{R}^{(1)}(j^*)$).*

*Furthermore, the optimal clustering cost of $G'$ ($\mathsf{cost}_2$) decreases by $\left(10K^2N - 2K^2\right)$ in comparison to the optimal clustering cost of $G$ ($\mathsf{cost}_1$).*

*Proof.* We first prove the clustering structure. By Observation 5.9, the only undecided clustering is the cluster that contains $\mathcal{L}(i^\star)$. There are three possible cases: $\mathcal{L}(i^\star)$ forms an individual cluster, $\mathcal{L}(i^\star)$ is clustered with $C_1 = \mathcal{L}_\uparrow \cup \mathcal{V}_\uparrow \cup \mathcal{R}_\uparrow^{(1)} \cup \mathcal{R}_\uparrow^{(2)}$, and $\mathcal{L}(i^\star)$ is clustered with $C_1 = \mathcal{L}_\downarrow \cup \mathcal{V}_\downarrow \cup \mathcal{R}_\downarrow^{(1)} \cup \mathcal{R}_\downarrow^{(2)}$. The only part that induces variate costs between the 3 clustering schemes are the edges with exactly one endpoint in $\mathcal{L}(i^\star)$. As such, it suffices to choose the optimal scheme that minimizes the costs induced by these edges.

If the vertices of $\mathcal{L}(i^\star)$ form an individual cluster, there are at least $90KN$ $(+)$ edges from every vertex in $\mathcal{L}(i^\star)$ to $\mathcal{V}$. As such, the cost induced by edges with exactly one endpoint in $\mathcal{L}(i^\star)$ is at least $90K^2N$. On the other hand, if $\mathcal{L}(i^\star)$ joins the cluster $C_2$, each vertex in $\mathcal{L}(i^\star)$ has at least $50KN$ $(+)$ edges to $C_1$ and $10KN$ $(-)$ edges in $C_2$, which induces a cost of at least $60K^2N$. Finally, if $\mathcal{L}(i^\star)$ joins the cluster $C_1$, it has at most $40KN$ $(+)$ between $\mathcal{V}_\downarrow$. Furthermore, it has at most $3KN$ other edges that can possibly induce any cost. This makes the induced cost to be at most $43K^2N$. As such, clearly the cluster that contains $\mathcal{L}(i^\star)$ should join cluster $C_1$.

We now analyze the optimal cost of correlation clustering on $G'$ compared to $\mathsf{cost}_1$ on $G$. Since only the clustering of $\mathcal{L}(i^\star)$ changes, it suffices to analyze the cost difference induced by the edges with exactly one endpoint in $\mathcal{L}(i^\star)$. We first observe that the costs induced by the following set of edges do *not* change between $\mathsf{cost}_1$ and $\mathsf{cost}_2$:

1. Edges between $\mathcal{L}(i^\star)$ and $\mathcal{L}(j)$ for any $j \neq i^\star$;

2. Edges between $\mathcal{L}(i^\star)$ and $\mathcal{R}^{(1)} - \mathcal{R}^{(1)}(j^\star)$;

3. Edges between $\mathcal{L}(i^\star)$ and $\mathcal{R}^{(2)} - \mathcal{R}^{(2)}(j^\star)$.

As such, we only analyze the edges between $\mathcal{L}(i^\star)$ and $\mathcal{V}$, $\mathcal{L}(i^\star)$ and $\mathcal{R}^{(1)}(j^\star)$, $\mathcal{L}(i^\star)$ and $\mathcal{R}^{(2)}(j^\star)$. In the optimal clustering of $G$, the edges between $\mathcal{L}(i^\star)$ and $\mathcal{V}$ always induce a cost of $50K^2N$. On the other hand, in the optimal clustering of $G'$, the edges between $\mathcal{L}(i^\star)$ and $\mathcal{V}$ induce a cost of $40K^2N$ since there are $10K^2N$ $(-)$ edges that do not induce cost. As such, among the edges between $\mathcal{L}(i^\star)$ and $\mathcal{V}$, the $\mathsf{cost}_2$ is $10K^2N$ smaller than $\mathsf{cost}_1$.

We now look at the edges between $\mathcal{L}(i^\star)$ and $\mathcal{R}^{(1)}(j^\star), \mathcal{R}^{(2)}(j^\star)$. Note that in the optimal clustering of $G$, these edges induce 0 all $(+)$ edges are in the same cluster and all $(-)$ edges are in different ones. On the other hand, for the optimal clustering on $G'$, each set of edges will induce a cost of $K^2$. Therefore, among the edges between $\mathcal{L}(i^\star)$ and $\mathcal{R}^{(1)}(j^\star), \mathcal{R}^{(2)}(j^\star)$, the $\mathsf{cost}_2$ is $2K^2$ larger than $\mathsf{cost}_1$.

Summing up the above terms gives us the conclusion that the cost difference between $\mathsf{cost}_2$ and $\mathsf{cost}_1$ is $10K^2N - 2K^2$. $\qquad\square$

From Lemma 5.10, we can see that the clustering cost reduction comes from the $(-)$ edges between $\mathcal{L}(i^\star)$ and $\mathcal{V}'_\downarrow$. However, since the optimal clustering of $G'$ 'switches' the partition of $\mathcal{L}(i^\star)$, it further incurs some increment of the cost. In contrast, we will show next that in the yes case, i.e. $M[i^\star, j^\star] = 1$, the optimal clustering remains the same as in $G$. As such, the optimal clustering does *not* pay the extra cost induced by the 'switching' of vertices in $\mathcal{L}(i^\star)$. The formal statement of the above intuition is as follows.

**Lemma 5.11.** *The optimal clustering of $G'$ for $M[i^\star, j^\star] = 1$ is the same as the optimal clustering of $G$, i.e.:*

- *The first cluster (the* up *cluster) contains the vertices of $\mathcal{L}_\uparrow, \mathcal{V}_\uparrow, \mathcal{R}_\uparrow^{(1)}, \mathcal{R}_\uparrow^{(2)}$ (including $\mathcal{R}^{(2)}(j^*)$) and $\mathcal{L}(i^*)$;*

- *The second cluster (the* down *cluster) contains the vertices of $\mathcal{L}_\downarrow, \mathcal{V}_\downarrow, \mathcal{R}_\downarrow^{(1)}, \mathcal{R}_\downarrow^{(2)}$ (including $\mathcal{R}^{(1)}(j^*)$).*

*Furthermore, the optimal clustering cost of $G'$ ($\mathsf{cost}_2$) decreases by $10K^2N$ in comparison to the optimal clustering cost of $G$ ($\mathsf{cost}_1$).*

*Proof.* We define $C_1$ and $C_2$ in the same way as in Lemma 5.10, and the clustering structure proof follows from exactly the same argument as the proof of Lemma 5.10. We omit the details to avoid excessive repetition.

We now analyze the optimal cost of correlation clustering on $G'$ compared to $\mathsf{cost}_1$ on $G$. Again, the same as the analysis of Lemma 5.10, we only need to analyze the edges between $\mathcal{L}(i^\star)$ and $\mathcal{V}$, $\mathcal{L}(i^\star)$ and $\mathcal{R}^{(1)}(j^\star)$, $\mathcal{L}(i^\star)$ and $\mathcal{R}^{(2)}(j^\star)$.

Among the edges between $\mathcal{L}(i^\star)$ and $\mathcal{V}$, we can obtain that $\mathsf{cost}_2$ is $10K^2N$ smaller than $\mathsf{cost}_1$ with exactly the same argument as in Lemma 5.10. Furthermore, for the edges between $\mathcal{L}(i^\star)$ and $\mathcal{R}^{(1)}(j^\star), \mathcal{R}^{(2)}(j^\star)$, note that the cluster of $\mathcal{L}(i^\star)$ does *not* change between the optimal clusterings of $G$ and $G'$, as such, the cost induced by these edges stays 0.

Summing up the above terms gives us the conclusion that the cost difference between $\mathsf{cost}_2$ and $\mathsf{cost}_1$ is $10K^2N$. $\qquad\square$

**Finalizing the proof of Theorem 4.** If the oracle for correlation clustering has additive error of $\frac{1}{10^6} \cdot n^{2-\varepsilon}$ and no multiplicative error, the cost estimation errors in both subroutines are at most $\frac{103^2}{10^6} \cdot K^{2-2\varepsilon} N^{2-2\varepsilon} \leq \frac{1}{50} K^2$. As such, if $M[i^\star, j^\star] = 0$, the estimation gap between $\mathsf{cost}_1$ and $\mathsf{cost}_2$ is at most $10K^2N - \frac{24}{25}K^2 < 10K^2N - \frac{1}{25}K^2$. On the other hand, if $M[i^\star, j^\star] = 1$, the estimation gap between $\mathsf{cost}_1$ and $\mathsf{cost}_2$ is at least $10K^2N - \frac{1}{25}K^2$, as desired.

Finally, the communication complexity of the problem is $\Omega(N^2)$, and we have $n = 103KN = O(N^{1/\varepsilon})$. As such, any such **Tool-CC**$(\varepsilon)$ must have a communication complexity of $\Omega(n^{2\varepsilon})$ bits, which implies the desired $\Omega(n^\varepsilon)$ space lower bound for the streaming algorithms.

# 6 A Lower Bound for $O(n)$ Additive error

In this section, we show that any dynamic streaming algorithm that gets a $(c, \varepsilon n)$-approximation for $c < 1.2$ and $O(n)$ additive error needs $\Omega(\sqrt{n})$ bits of space. Here, and throughout, we will call this lower bound the *linear lower bound*. Formally, we have:

**Theorem 5.** *Let $c \in [1, \frac{6}{5})$ and $\varepsilon \in (0, \frac{6}{5} - c)$. Any single-pass streaming algorithm that estimates the optimal value $\mathsf{OPT}$ of correlation clustering by a $(c, \varepsilon n)$-approximation (i.e., an estimated value that is at most $(c \cdot \mathsf{OPT} + \varepsilon \cdot n)$) with probability at least $\frac{99}{100}$ has to use a memory of $\Omega(\sqrt{n})$ bits, even on labeled (complete) graphs.*

Similar to the case in Theorem 4, Theorem 5 does *not* require the stream to be dynamic. Compared with the *almost-quadratic lower bound* we show in Section 5, the lower bound in Theorem 5 is weaker in the additive sense. However, it allows the multiplicative approximation of the algorithm to be $> 1$, while the lower bound in Section 5 only rules out algorithms with purely additive errors.

## 6.1 Gap Cycle Counting with Odd Cycles

Our lower bound uses the celebrated machinery from Boolean Hidden Hypermatching (BHH) and Gap Cycle Counting (GCC) pioneered by [VY11]. The Gap Cycle Counting (GCC) lower bound states that for any algorithm to distinguish whether a graph consists of cycles with length $2t$ or cycles with length $4t$ for some $t \geq 2$, a memory of $\Omega\left(n^{1-\frac{1}{t}}\right)$ bits is necessary. On a high level, our plan is to show that for graphs similar to the ones prescribed in the GCC problem, the values of

correlation clustering are different by an additive gap of $O(n)$. Therefore, by a reduction argument, any algorithm that breaks this barrier of additive gap requires $\omega\left(\text{polylog}(n)\right)$ memory.

The roadblock here is that the original version of Gap Cycle Counting, which only supports *even* cycles, is not sufficient to distinguish the values for correlation clustering. In fact, one can show that for graphs consisting of cycles with any *even* length, the optimal cost for correlation clustering is always $\frac{n}{2}$. To overcome this challenge, we modify the reduction from Boolean Hidden Hypermatching to obtain new hardness results on Gap Cycle Counting with odd cycles. More concretely, we prove the following lemma.

**Lemma 6.1.** *Any single-pass streaming algorithm that distinguishes graphs of cycles with length $(2t+1)$ from cycles with length $2 \cdot (2t+1)$ with probability at least $\frac{99}{100}$ has to use a memory of $\Omega\left(n^{1-\frac{1}{t}}\right)$ bits.*

To prove Lemma 6.1, we need to formally introduce the Boolean Hidden Hypermatching (BHH) problem as follows.

**Definition 8** (Boolean Hidden Hypermatching ($\mathsf{BHH}_{m,t}$), [VY11, LW16a, AKL17]). The Boolean Hidden Hypermatching problem (denoted as $\mathsf{BHH}_{m,t}$) is a one-way communication problem between two players, namely Alice and Bob. Alice is given a boolean vector $x \in \{0,1\}^m$, where where $m = 2kt$ for some integer $k \geq 1$ and Bob gets a perfect $t$-hypermatching $M$ on $m$ vertices, and a boolean vector $w \in \{0,1\}^{\frac{m}{t}}$. Let $Mx$ denote the length $\frac{m}{t}$ boolean vector $(\bigoplus_{1 \leq i \leq t} x_{M_1,i}, \cdots, \bigoplus_{1 \leq i \leq t} x_{M_{m/t},i})$, where $\{M_{1,1}, \cdots, M_{1,t}\}, ..., \{M_{m/t,1}, \cdots, M_{m/t,t}\}$ are the indices corresponds to the vertices matched by hypermatching $M = \{M_1, \cdots, M_{m/t}\}$. It is promised that either $Mx \bigoplus w = 0$ or $Mx \bigoplus w = 1$. The goal of the problem is to output *Yes* when $Mx \bigoplus w = 0$ and *No* when $Mx \bigoplus w = 1$.

Note that we use $m$ to denote the number of vertices in the Boolean Hidden Hypermatching to avoid ambiguity in the reduction, where we are going to construct a new graph with $n$ vertices. It is known that solving $\mathsf{BHH}_{m,t}$ is hard even when Bob's vector is deterministically all zero. More formally, the following result is standard in the literature.

**Proposition 6.2.** *(Communication Complexity of $\mathsf{BHH}_{m,t}$, [VY11]) Let $m = 2kt$ for some integer $k \geq 1$ and $t \geq 2$ as prescribed in Definition 8. Any one-way communication protocol that solves $\mathsf{BHH}_{m,t}$ with probability at least $\frac{99}{100}$ requires $\Omega\left(n^{1-\frac{1}{t}}\right)$ bits in the message. Furthermore, the communication complexity holds even Bob's vector is known to be $\mathbf{0}^{m/t}$.*

We will the hardness result of Proposition 6.2 to prove the hardness of a new version of the Gap Cycle Counting problem, defined as follows.

**Definition 9** (Gap Cycle Counting – Odd Cycle). The Gap Cycle Counting with odd cycle problem is a one-way communication problem between two players, namely Alice and Bob. The input to the players is as follows.

- Both players are given $n$ vertices, of which $\frac{1}{2t+1}$ fraction of the vertices are *special*.

- Alice is given a set of edges $E_A$ which is a matching between the vertices that are *not* special.

- Bob is given a set of edges $E_B$ which consists of two edges on each of the special vertex.

The promise is that $E_A \cup E_B$ is either a disjoint union of cycles with $(2t+1)$ length, or a disjoint union of cycles with $2 \cdot (2t+1)$ length. The goal for the players is to answer *Yes* for the former scenario and *No* for the latter.

We show that the above GCC with odd cycle problem requires the same communication complexity as Definition 8.

**Claim 6.3.** *Any one-way communication protocol that solves the Gap Cycle Counting – Odd Cycle problem with probability at least $\frac{99}{100}$ requires $\Omega\left(n^{1-\frac{1}{t}}\right)$ bits in the message.*

*Proof.* We prove this by a reduction from the $\mathsf{BHH}_{m,t}$ problem. Suppose we have a communication protocol $\mathsf{PROT}$ that solves Gap Cycle Counting – Odd Cycle problem with probability at least $\frac{99}{100}$ and a message of $o\left(n^{1-\frac{1}{t}}\right)$ bits, we show that it is possible to get a communication protocol that solves $\mathsf{BHH}_{m,t}$ with probability $\frac{99}{100}$ and the same message. This contradicts the lower bound of Proposition 6.2. Therefore, the only conclusion is that such a $\mathsf{PROT}$ cannot exist.

The communication protocol to solve $\mathsf{BHH}_{m,t}$ can be given as follows.

---

$\mathsf{PROT}'$: a communication protocol for $\mathsf{BHH}_{m,t}$
**Input:** An instance of $\mathsf{BHH}_{m,t}$.
**Output:** The decision of whether $Mx = 0$ or $Mx = 1$.

1. Alice and Bob both create an all-one vector $y = \mathbf{1}^{2m/t}$, and append it after their own vectors (without communication).

2. Alice locally creates vertices and edges:

   (a) For each $x_i$, create 4 vertices: $u_i$, $v_i$, $u'_i$, $v'_i$.

   (b) If $x_i = 0$, add $(u_i, v_i) \in E_A$ and $(u'_i, v'_i) \in E_A$.

   (c) If $x_i = 1$, add $(u_i, v'_i) \in E_A$ and $(u'_i, v_i) \in E_A$.

   (d) For each $y_j$, create two vertices $\tilde{u}_j, \tilde{u}'_j$, and add no edge to them.

3. Bob locally creates vertices and edges:

   (a) For each vertex $z_i$ in the $\mathsf{BHH}_{m,t}$ instance, create vertices $u_i$, $v_i$, $u'_i$, $v'_i$ in the same manner of Alice.

   (b) For each hyperedge $M_j$, denote $(M_{j,1}, M_{j,2}, \cdots, M_{j,t})$ as the indices of the vertices.

   (c) For $\ell = 1$ to $t - 1$, Bob adds edges between $(v_\ell, u_{\ell+1}) \in E_B$ and $(v'_\ell, u'_{\ell+1}) \in E_B$.

   (d) Bob further create two vertices $\tilde{u}_j, \tilde{u}'_j$ for each $y_j$, and assign to $j$-th matching by adding four edges: $(v_t, \tilde{u}_j) \in E_B$, $(v'_t, \tilde{u}'_j) \in E_B$, $(\tilde{u}_j, u_1) \in E_B$, $(\tilde{u}'_j, u'_1) \in E_B$.

4. The players runs $\mathsf{PROT}$, and answers as the output of $\mathsf{PROT}$, i.e. answer Yes if $\mathsf{PROT}$ returns Yes (graph is of $(2t + 1)$ cycles), and vice versa.

---

An intuitive understanding of the edges added by Alice can be found in Figure 1. Observe that as in the figure, if we always put vertices $u$ and $v$ to the 'outer' circle of the vertices and $u'$ and $v'$ to the 'inner' one, then the edges added by Alice are 'parallel' when $x_i = 0$ and 'crossing' when $x_i = 1$. Bob always adds 'parallel' edges between the corresponding vertices.

It is straightforward to verify that the inputs constructed by Alice and Bob is valid for $\mathsf{PROT}$: the fraction of the special vertices is indeed $\frac{2m/t}{4m + 2m/t} = \frac{1}{2t+1}$. We now show that the graph constructed by the two players is with cycles of length $(2t+1)$ if and only if $Mx = 0$, and cycles of length $2 \cdot (2t+1)$ if and only if $Mx = 0$. Note that for a graph with only 'parallel' edges, each matching in the $\mathsf{BHH}_{m,t}$

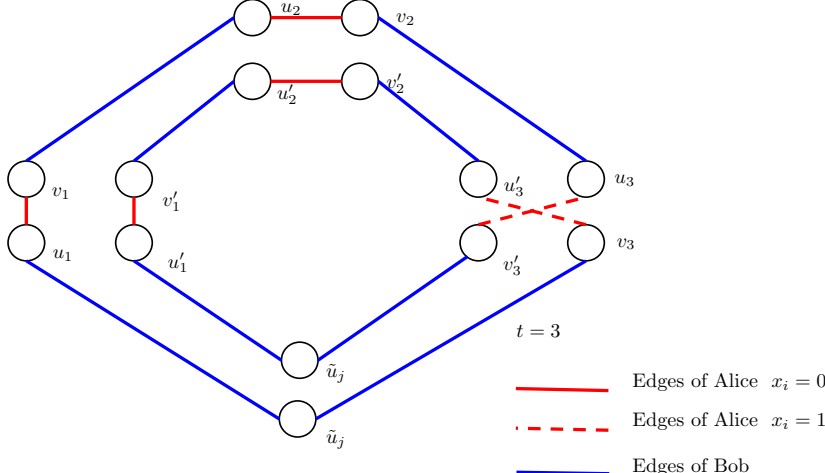

Figure 1: The $\mathsf{BHH}_{m,t}$ instance to Gap Cycle Counting with odd cycles. In the instance of the figure, Alice has $x_3 = 1$ (for matching $j$). Therefore, we have $Mx = 1$ and the graph is one 14-length cycle. One can verify that if one more set of 'cross edges' is added, the graph goes to two separate cycles.

instance induces two cycles of length $(2t+1)$. If one of the vertex pairs switches to 'cross' edges, the tours of the cycles intersect and the two cycles merge into one cycle with length $2 \cdot (2t+1)$. Inductively, one can show that if there is an even number of 'cross' edge connections, the two $(2t+1)$-length cycles remain separate; and if there is an odd number of 'cross' edge connections, the two $(2t+1)$-length cycles merge. Therefore, the type of cycle induced by the $\mathsf{BHH}_{m,t}$ instance depends uniquely on the value of $Mx$ for each matching. Furthermore, the cycles are constructed independently from each matching, and there are no other edges. This gives us the desired correspondence between the length of the cycles and the value of $Mx$.

Since the assumption is that $\mathsf{PROT}$ succeed with probability at least $\frac{99}{100}$, and the reduction in $\mathsf{PROT}'$ is *deterministic*, $\mathsf{PROT}'$ can answer $\mathsf{BHH}_{m,t}$ successfully with the same probability. Therefore, any such $\mathsf{PROT}$ must have a communication complexity of at least $\Omega\left(n^{1-\frac{1}{t}}\right)$. $\qquad\square$

**Proof of Lemma 6.1.** The proof of Lemma 6.1 now follows from a standard reduction by Claim 6.3. Suppose a streaming algorithm that can distinguish graphs of cycles $2t+1$ from graphs of cycles $2 \cdot (2t+1)$ exists, one can use the streaming algorithm to simulate a communication protocol: Alice runs the streaming algorithm, send the memory of the streaming algorithm as the message, and Bob uses the memory to run the rest of the streaming algorithm. Therefore, such a streaming algorithm must use a memory of at least $\Omega\left(n^{1-\frac{1}{t}}\right)$ bits.

## 6.2 The lower bound

We now formally state and prove the additive approximation gap for correlation clustering. To start with, we first show a lower bound for *dynamic* streaming algorithms that allows the operations of the insertion and removal of $(+)$ and $(-)$ edges (see Section 2.3 for the detailed definition). Switching to dynamic streams simplifies the reduction, and we will deal with an additional technical part to prove the lower bound for insertion-only streaming algorithms later.

**Lemma 6.4.** *Let $c \in [1, \frac{6}{5})$ and $\varepsilon \in (0, \frac{6}{5} - c)$. Any single-pass dynamic streaming algorithm that estimates the optimal value* OPT *of correlation clustering by a $(c, \varepsilon n)$-approximation (i.e., an estimated value that is at most $(c \cdot$ OPT $+ \varepsilon \cdot n)$) with probability at least $\frac{99}{100}$ has to use a memory of $\Omega(\sqrt{n})$ bits, even on labeled (complete) graphs.*

*Proof of Lemma 6.4.* We use the simple instance of cycles with length 5 versus cycles of length 10 from the results of Lemma 6.1. We again use the reduction argument: given a dynamic streaming algorithm ALG that achieves a $(c, \varepsilon)$-approximation for the optimal cost of correlation clustering on labeled graphs, we can design a streaming algorithm to distinguish graphs with cycles of length 5 and cycles of length 10 in the following way:

---

1. $(-)$ edges: By the start of the stream, for each vertex pair $(u, v)$, insert a $(-)$ edge.

2. $(+)$ edges: For each arriving edge $(u, v) \in E$, remove the $(-)$ edge between $(u, v)$, and add a $(+)$ edge.

3. Run ALG and output with the following rules: if the cost is less than $\frac{3n}{5}$, output *No* (graph of cycles of length 10); otherwise, output *Yes* (graph of cycles of length 5).

---

We show that provided the correct probability of ALG, we can distinguish the scenarios of length-5 and length-10 cycles with the same probability. Note that by the construction, the $(+)$ edges are the cycle edges. For a graph with $(+)$ edges of length-10 cycles, there is a way to induce a cost of $\frac{n}{2}$: on each cycle, we can put two vertices in the same cluster, and create 5 clusters. In this way, we pay a cost of 5 by splitting $(+)$ edges, and there are $\frac{n}{10}$ such cycles.

On the other hand, when the graph is of disjoint cycles of length 5, we show that the cost is at least $\frac{3n}{5}$. To see this, we first note that an optimal clustering never puts two vertices from different cycles in the same cluster, as there are only $(-)$ edges between them. Among any cycle, the optimal clustering is to put 2 vertices in one cluster, and the other 3 in the same cluster. This induces a cost of 3, and there are $\frac{n}{5}$ such cycles, which gives us the desired solution.

Finally, note that if ALG achieves $(c, \varepsilon)$-approximation in the given parameter range, the value is strictly less than $\frac{3n}{5}$ when the graph is of disjoint length-10 cycles. Therefore, we get the desired reduction. $\qquad \square$

We now generalize the results to insertion-only streams. To this end, we will need the following variation of the Gap Cycle Counting with odd cycles.

**Lemma 6.5.** *Suppose a complete graph has two types of edges: the* normal *edges and the* special *edges. Let $G$ be a graph whose* special *edges form disjoint cycles of length $(2t+1)$, and $\tilde{G}$ be a graph whose* special *edges forms disjoint cycles of length $2 \cdot (2t+1)$. Any single-pass streaming algorithm that distinguishes $G$ from $\tilde{G}$ with probability at least $\frac{99}{100}$ has to use a memory of $\Omega\left(n^{1-\frac{1}{t}}\right)$ bits.*

*Proof.* We can prove Lemma 6.5 by slightly tweaking the Gap Cycle Counting with odd cycle problem of Definition 9. Now suppose the edges $E_A$ and $E_B$ that are given to Alice and Bob are the *special* edges, and we further provide other *normal* edges to make the graph complete. The goal for the players is to distinguish the length of the cycles from the *special* edges.

Let us call this problem GCC-odd-C ('C' stands for 'complete'). There is a straightforward reduction from the $\mathsf{BHH}_{m,t}$ problem to this new variate GCC-odd-C. Suppose we have a communication protocol that solves GCC-odd-C. Then, Alice can mark all the edges she added in the reduction of Claim 6.3 as *special*, and add *normal* edges between all other pairs between $(u_i, v_i, u_i', v_i')$ for each

*i*. Bob can similarly mark all the edges he added in the reduction of Claim 6.3 as *special*, and add *normal* edges between the vertex pairs of different $\ell$ and $(\tilde{u}_j, \tilde{u}'_j)$ whenever there is not a special edge. Such a protocol can solve $\mathsf{BHH}_{m,t}$ with output of GCC-odd-C. Therefore, any communication protocol that solves GCC-odd-C requires $\Omega\left(n^{1-\frac{1}{t}}\right)$ bits communication, and the streaming lower bound follows. $\qquad\square$

**Finalizing the proof of Theorem 5.** We can prove the lower bound for insertion-only streaming algorithms based on Lemma 6.5. Note that now we can mark each *special* edge as a (+) edge, and each *normal* edge as a (−). The graph in GCC-odd-C is complete, which means we can construct an input to the insertion-only streaming algorithm without any extra memory cost. As such, combining the proof of Lemma 6.4 and the result of Lemma 6.5 yields the result.

# 7 Experiments

We describe in this section the experiments of algorithms on the synthetic graph streams generated by the Stochastic Block Model and the Erdos-Renyi random graphs. These experiments show that for a very natural family of graphs, our algorithms can actually achieve a very competitive performance on the estimated cost values, and the performances are often much better than the worst-case theoretical analysis. Furthermore, our algorithms are capable of separating "well-clusterable" vs. "badly-clusterable" instances.

## 7.1 Experimental Settings

**The synthetic data and the graph stream.** As we have discussed, we perform our experiments on the data generated from the well-studied Stochastic Block Model (SBM). For the correlation clustering application, we use the variate of the SBM that plants ground-truth clusters with sizes $\Omega(n)$, samples (+) edge between vertex pairs $(x_i, x_j)$ in the same planted cluster with probability $p > 0.5$, and samples (+) edges between vertex pairs $(x_i, x_j)$ in different clusters with probability $1-p$. The SBM captures a lot of real-world scenarios, including social networks [HLL83], community detection [Abb17b], graph clustering [LW19], and Bioinformatics [MGC21], to name a few.

When the probability $p$ in the above description is sufficiently large, we can assume the ground-truth clusters are the optimal correlation clustering solutions, and obtain the optimal cost by counting the edges. As such, we run our algorithms on SBM instances that are reasonably "clusterable", i.e., we set $p = 0.8$ with vertices $n = 500$, $n = 1000$, and $n = 2000$. Note that since we are dealing with labeled complete graphs, the number of edges scales quadratically w.r.t. $n$, which gives us a non-trivial scale of instances.

Another family of graphs we tested is the well-known Erdos-Renyi random graphs $G(n, p)$. In this model, a (+) edge is added to a vertex pair $(x_i, x_j)$ with probability $p$ independently across edge slots. As such, when $p$ is around 0.5, the graph does not appear any clusterable property, and the cost is very high. Although it is not clear how to exactly compute the costs for the Erdos-Renyi random graphs, we run our experiments by comparing the costs produced by our algorithm from the Erdos-Renyi graphs with the costs from the SBM model with $p = 0.95$, i.e., when the instances are very "well-clusterable" (we use $n = 1000$ for this test). In this way, we can observe our algorithms' ability to distinguish "good" and "bad" instances – a property that can be extremely useful in practice.

**Implementation of the algorithms and experiments.** We implement both the SDD-based and the pivot-based algorithms. On top of following the procedure as described in Section 3 and Section 4, we also use some ad-hoc heuristics in the implementation. For the number of edges and vertices to be sampled, we do *not* scale the number by $\varepsilon.\delta$, and instead pick a fixed constant.

The constant is much smaller than we used in the theoretical analysis, but the performances are nonetheless very well. Furthermore, since our distributions have sufficiently high positive degrees, we do *not* need to add the $\delta n^2$ term. Finally, for the pivot-based algorithm, we slightly relax the requirement to allow 2-pass over the stream; as such, we can use the first pass to perform greedy MIS on the sampled vertex set, and we do not have to duplicate the storage by the $2^{O(1/\delta)}$ factor.

We evaluate the performances mainly based on two metrics: the multiplicative factor of the cost estimation (which we call the "competitive ratio") and the fraction of the edges used. To overcome the possible effects of random seed, we fix our random seed with $0 \sim 14$, and run 15 experiments. We include the error bars and the curves of the competitive ratios and fraction of the edges. For the experiment to distinguish SBM and Erdos-Renyi graphs, we simply plot the two types of costs w.r.t. experiment runs, and give the distributions of the costs.

## 7.2  Experimental Results

We first show that our algorithms are insensitive to the choice of parameters – this property is evident from Figure 2 and Figure 3, where the box plots are obtained by 15 runs with different parameters, but the results are almost identical. As such, we focus on the settings with $\varepsilon = 0.01$ for the SDD-based algorithm and $\delta = 0.1$ for the pivot-based algorithm for the rest of this section.

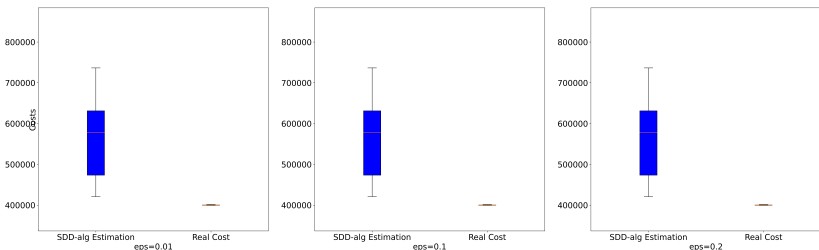

Figure 2: The estimation results for 15 experiment runs with the SDD-based algorithm with parameters $\varepsilon = 0.01, 0.1, 0.2$.

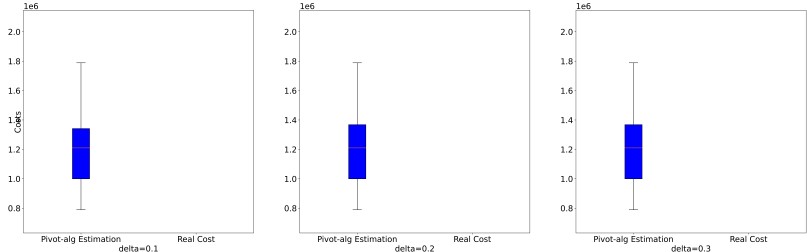

Figure 3: The estimation results for 15 experiment runs with the pivot-based algorithm with parameters $\delta = 0.1, 0.2, 0.3$.

We now discuss our algorithm based on sparse dense decomposition (SDD). Figure 4 to Figure 6 give the plots of the cost estimation for the SDD-based algorithm on $n = 500$, $n = 1000$. and $n = 2000$ sampled from the SBM distribution. The figures show that the approximation factor for this algorithm is roughly between 1 and 2, and the fraction of the edges drops when $n$ increases. For the $n = 2000$ case, the SDD-based algorithm consistently uses less than 4% of the edges.

The task to distinguish between SBM instances with $p = 0.95$ and ER instances with $p = 0.5$ with the SDD-based algorithm is shown in Figure 7. From the figure, it can be observed that the SDD-based algorithm outputs drastically different clustering costs of the SBM instances vs the Erdos-Renyi instances. In addition to having much higher costs on the ER instances for all runs, we can also observe from the left plot that the supports of the costs are *disjoint*. As such, by a simple linear threshold, our SDD algorithm is able to perfectly distinguish between both types of instances while using less than 10% of the edges ($n = 1000$ case).

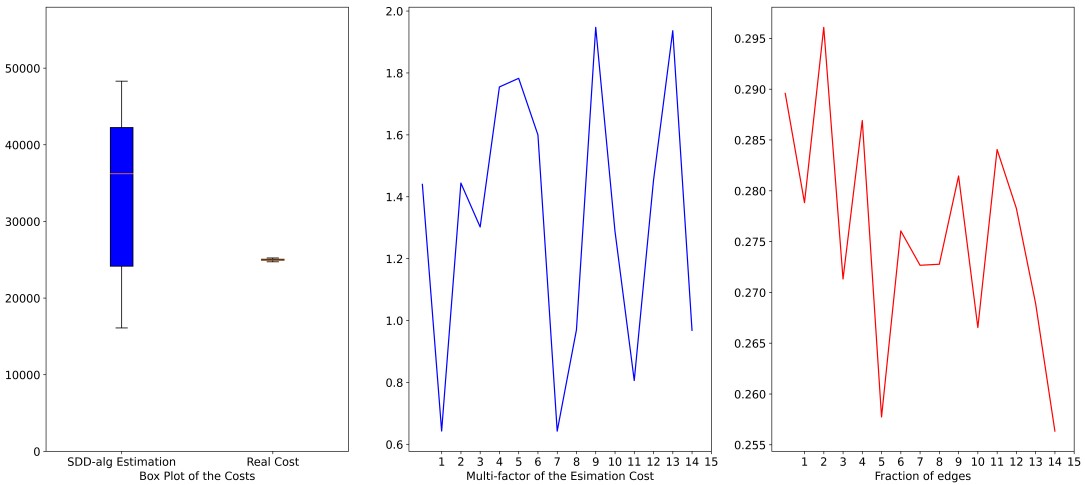

Figure 4: The performances of the SDD-based algorithm on $n = 500$ SBM.

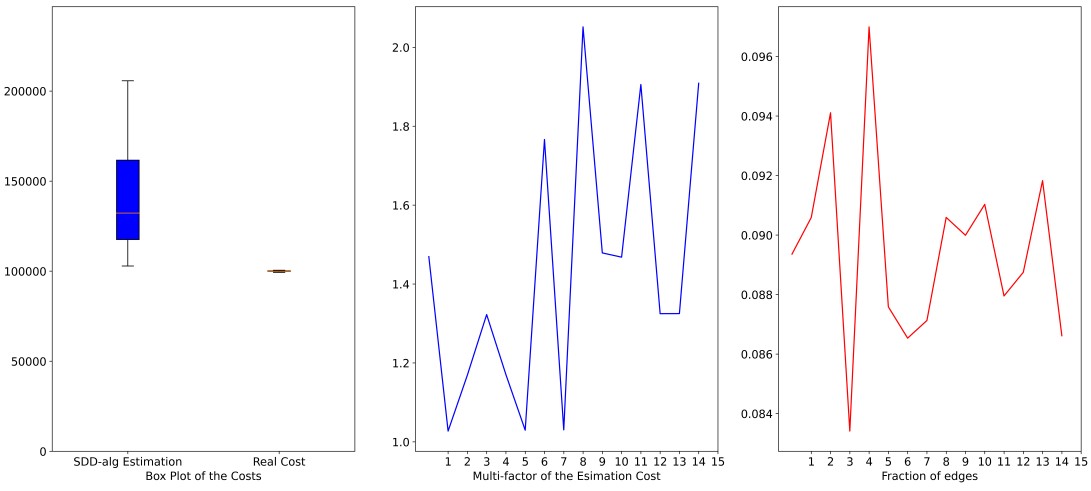

Figure 5: The performances of the SDD-based algorithm on $n = 1000$ SBM.

We now discuss the performances of our algorithm based on pivot. Figure 8 to Figure 10 give the performances for the pivot-based algorithm. The approximation factor for this algorithm is slightly worse than the SDD-based, which are roughly between 2 and 4.5; however, on average we still get

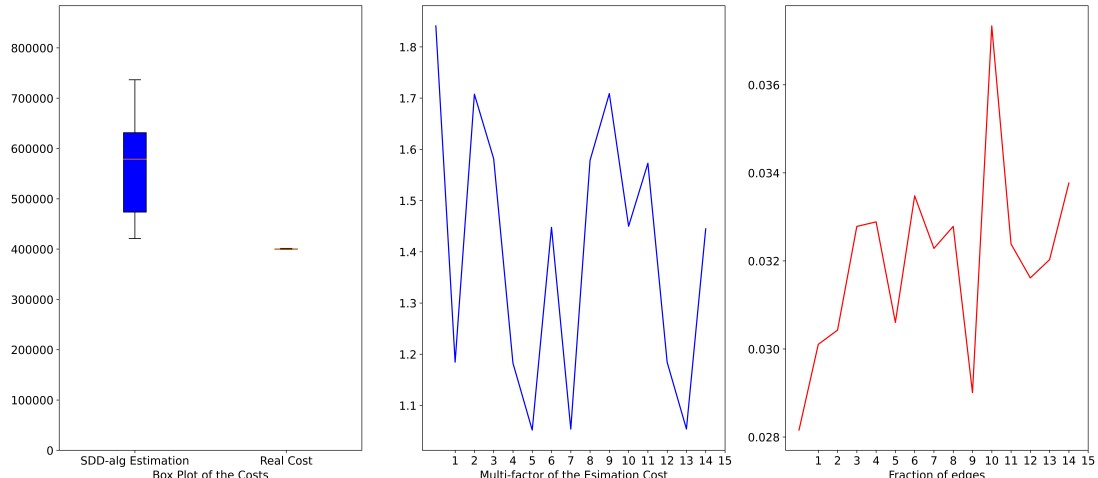

Figure 6: The performances of the SDD-based algorithm on $n = 2000$ SBM.

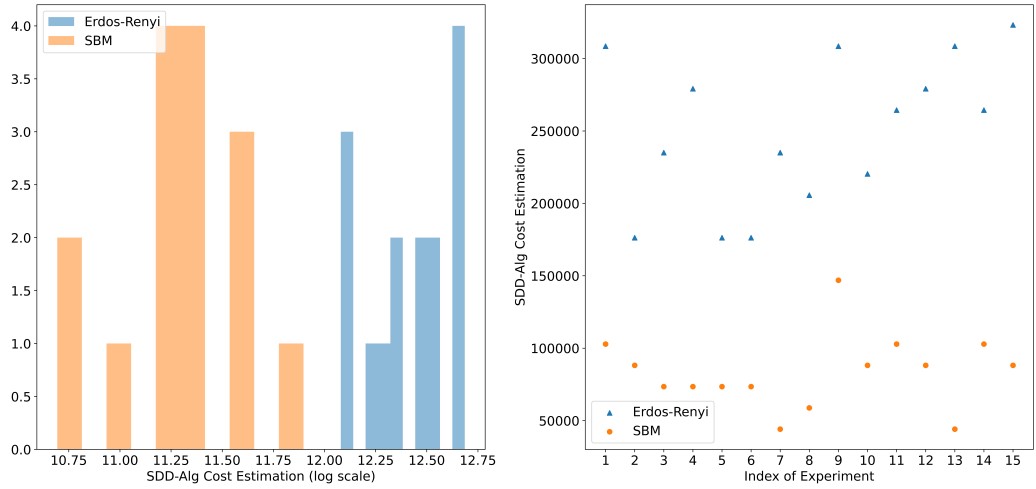

Figure 7: The output costs of the SBM and ER instances by the SDD-based algorithm. Left: cost distributions; Right: costs for each run of the experiment.

an approximation factor less than 3, which is the expected cost of the original pivot algorithm. On the other hand, compared to the SDD-based algorithm, the number of edges stored by the pivot algorithm is even smaller – for the $n = 2000$ case, we only store $\sim 0.04\%$ of the edges. This is partly due to the 2-pass implementation, but it also shows the huge potential of the algorithm in practice.

The pivot-based algorithm can also effectively distinguish between well-clusterable and badly-clusterable instances. Figure 11 shows that the estimated clustering costs of the SBM instances are very different from the estimated clustering costs of the Erdos-Renyi instances. Again, we obtain support-disjoint distributions of costs, and we can find a threshold cost below which we have the SBM instances and above the ER instances. Notably, our pivot algorithm is able to achieve this and can perfectly distinguish between both types of instances by storing $\sim 0.1\%$ of the edges.

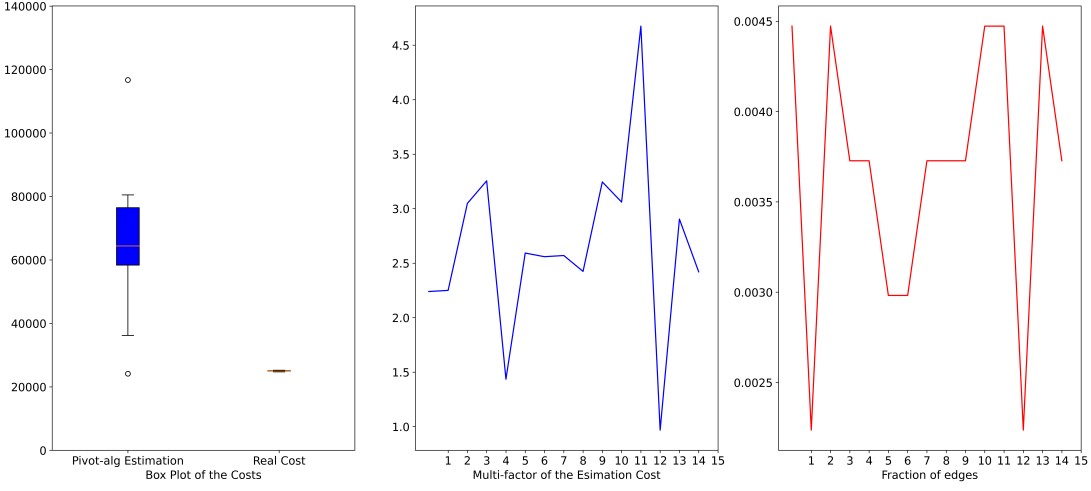

Figure 8: The performances of the pivot-based algorithm on $n = 500$ SBM.

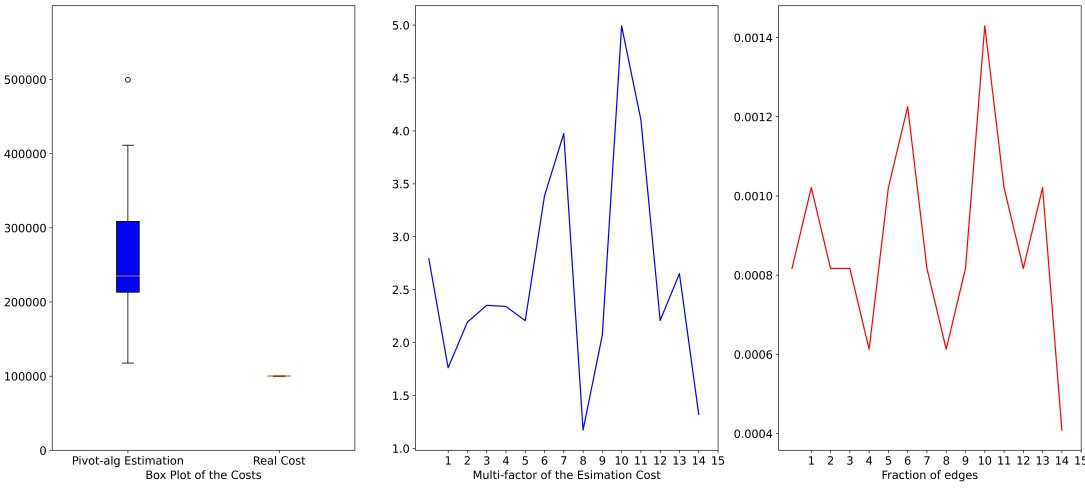

Figure 9: The performances of the pivot-based algorithm on $n = 1000$ SBM.

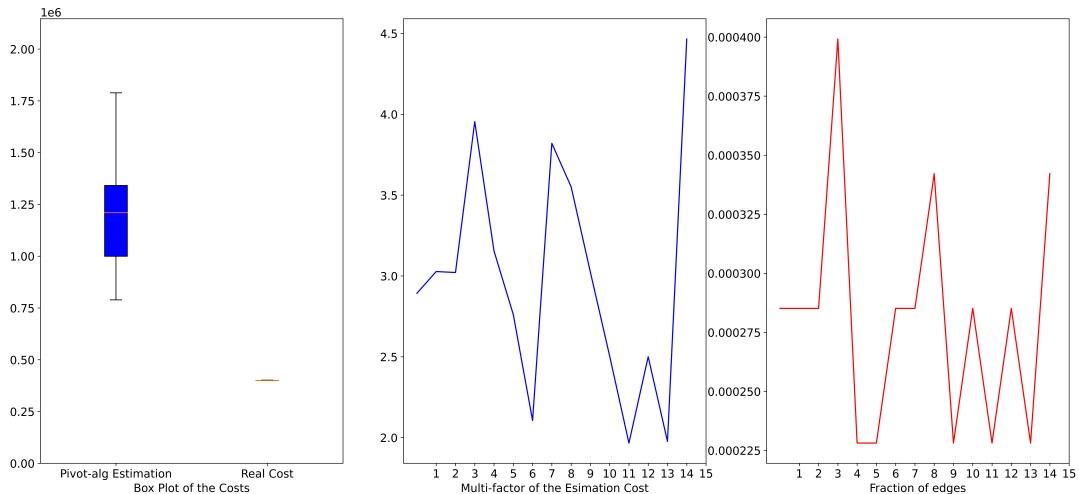

Figure 10: The performances of the pivot-based algorithm on $n = 2000$ SBM.

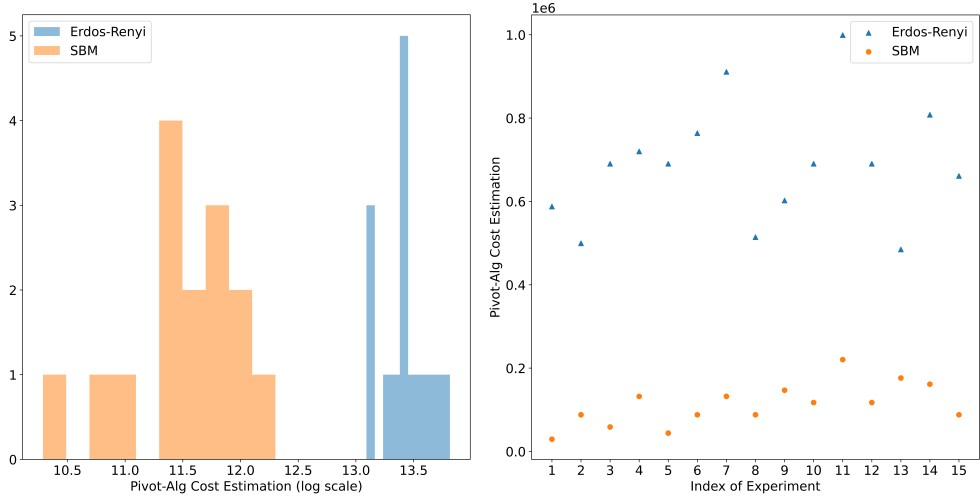

Figure 11: The output costs of the SBM and ER instances by the pivot-based algorithm. Left: cost distributions; Right: costs for each run of the experiment.

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

# A    Proof of Lemma 3.15

We prove Lemma 3.15 that upper-bounds the $m^+_{\beta\varepsilon\text{-sparse}}$ and $\widehat{m}^-_{\beta\varepsilon\text{-dense}}$ with $O(1/\varepsilon) \cdot \mathsf{OPT}$. To begin, we state the lemma again as follows.

**Lemma A.1** (Restatement of Lemma 3.15). *Suppose $G = (V, E)$ is any labeled graph and OPT be the optimal correlation clustering cost, and let $\beta, \varepsilon$ be such that $0 < \beta \leq \frac{1}{2\varepsilon}$, there is*

$$m^+_{\beta\varepsilon\text{-}sparse} \leq \frac{2}{\beta\varepsilon} \cdot \mathsf{OPT};$$

$$\widehat{m}^-_{\beta\varepsilon\text{-}dense} \leq 8 \cdot \mathsf{OPT}.$$

Similar to the argument in [AW22], we resort to a charging lemma to prove the upper bound for a clustering that pays costs of the $\varepsilon$-sparse edges and $\varepsilon$-dense non-edges. In particular, we show that if all edges that induce costs are at least $\varepsilon$-sparse (cf. if the non-edges that induce costs are at most $\varepsilon$-dense), one can find proper charge-sets such that the optimal cluster $\mathcal{O}$ must incur costs on.

We first prove the following lemma that establishes our desired property of the charging scheme.

**Lemma A.2** ([AW22]). *Let $E_c$ be the set of edges $((+)$ and $(-))$ that incur costs. Suppose there is a choice of $\mathrm{ChargeSet}(e)$ for edges $e \in E_c$ in the charging scheme such that for all $z \in V$ and $f \in E(z)$, we have $\mathsf{charge}(z, f) \leq \alpha$ for some $\alpha \geq 1$. Then,*

$$\mathsf{cost}(E_c) \leq 2\alpha \cdot \mathsf{OPT}.$$

*Proof.* We have,

$$
\begin{aligned}
\mathsf{cost}(E_c) &= \sum_{e \in E_c} \mathsf{cost}(e) && \text{(by the definition of the total cost)} \\
&= \sum_{e \in E_c} \sum_{\substack{(z,f) \in \\ \mathrm{ChargeSet}(e)}} |\mathrm{ChargeSet}(e)|^{-1} && \text{(the inner sum is 1)} \\
&= \sum_{z \in V} \sum_{\substack{f \in E(z) \\ \text{and} \\ \mathsf{cost}(\mathsf{OPT},e)=1}} \mathsf{charge}(z, f) && \text{(by summing over charges of all vertex-edge pairs)} \\
&\leq \sum_{z \in V} \alpha \cdot |f \in E(z) \text{ and } \mathsf{cost}(\mathsf{OPT}, e) = 1| && \text{(by the guarantee of the lemma statement)} \\
&= \sum_{f \in E} 2\alpha \cdot \mathsf{cost}(\mathsf{OPT}, e) && \text{(as each edge will be added twice (one by each endpoint))} \\
&= 2\alpha \cdot \mathsf{cost}(\mathsf{OPT}).
\end{aligned}
$$

This concludes the proof. $\square$

By Lemma A.2, we only need to find charge-sets of the edges in $E_c$ so that $\mathsf{charge}(z, f)$ is small for all vertex-edge pairs $(z, f)$. In particular, to prove the bounds on $m^+_{\text{-sparse}} \beta\varepsilon$ in Lemma 3.15, we can set $E_c$ as collection of the $\varepsilon$-sparse $(+)$ edges; on the other hand, to prove the upper bound on $\widehat{m}^-_{\beta\varepsilon\text{-dense}}$, we will use a carefully-defined set of edges together with Observation 3.13 on the top of the charging lemma.

**Part I: Upper bound of $m^+_{\beta\varepsilon\text{-sparse}}$**

We now analyze $m^+_{\beta\varepsilon\text{-sparse}}$ by charging the cost of $\beta\varepsilon$-sparse $(+)$ edges. We assume $\beta = 1$ since we can always scale $\beta$ to get the desired conclusion. We define $\mathsf{sparse\text{-}charge}(z, f)$ as the contribution from the $\varepsilon$-sparse $(+)$ edges to $\mathsf{charge}(z, f)$. We show that,

**Lemma A.3.** *Let $E_c^+$ be the the set of all $\varepsilon$-sparse (+) edges. There is*

$$\left| E_c^+ \right| \leq \frac{2}{\varepsilon} \cdot \mathsf{OPT}.$$

*Proof.* We show that there exist sets ChargeSet($e$) for every $\varepsilon$-sparse edge $e \in E_c^+$, such that for all vertex-edge pairs $(z, f)$, there is

$$\mathsf{sparse\text{-}charge}(z, f) = \frac{1}{\varepsilon}.$$

Define ChargeSet($e$) for any $(u, v) = e$ in $E_c^+ := E_{\varepsilon\text{-sparse}}^+$ as follows:

- Type-1 charges: when $\mathsf{cost}(\mathsf{OPT}, e) = 1$. In this case, we simply set ChargeSet($e$) = $\{(u, e)\}$ itself.

- Type-2 charges: when $\mathsf{cost}(\mathsf{OPT}, e) = 0$. Note that in this case, we have that $\mathcal{O}(u) = \mathcal{O}(v)$ under the optimal clustering $\mathcal{O}$, and let us denote this cluster as $\mathcal{O}_{uv}$. Consider any vertex $w \in N^+(u) \triangle N^+(v)$:

  - Case A: $w \in N^+(u)$ and $w \in N^-(v)$. In this case, there is $\mathsf{cost}(\mathsf{OPT}, (w, v)) = 1$ if $\mathcal{O}(w) = \mathcal{O}_{uv}$, and $\mathsf{costOPT}(w, u) = 1$ if $\mathcal{O}(w) \neq \mathcal{O}_{uv}$.

  - Case B: $w \in N^+(v)$ and $w \in N^-(u)$. In this case, there is $\mathsf{cost}(\mathsf{OPT}, (w, u)) = 1$ if $\mathcal{O}(w) = \mathcal{O}_{uv}$, and $\mathsf{cost}(\mathsf{OPT}, (w, v)) = 1$ if $\mathcal{O}(w) \neq \mathcal{O}_{uv}$.

  Therefore, in both cases, there is exactly one edge $f(w) \in \{(w, u), (w, v)\}$ such that $\mathsf{cost}(\mathsf{OPT}, f(w)) = 1$. Let $z(w) \in \{u, v\}$ be the vertex other than $w$ incident on $f(w)$. We add all pairs $(z(w), f(w))$ to ChargeSet($e$), i.e.,

  $$\mathsf{ChargeSet}(e) = \left\{ (z(w), f(w)) \mid w \in N^+(u) \triangle N^+(v) \right\}.$$

  By definition, the $\varepsilon$-sparse edges have

  $$\left| N^+(v) \triangle N^+(u) \right| \geq \varepsilon \cdot \max \left\{ \deg^+(u), \deg^+(v) \right\};$$

  Therefore, we have that $|\mathsf{ChargeSet}(e)| \geq \varepsilon \cdot \max \left\{ \deg^+(u), \deg^+(v) \right\}$.

Let us now bound the distributed charges. We have three different choices for $(z, f)$ that can belong to ChargeSet($e$) for some edge $e \in E_c^+$ as follows:

- A pair $(u, e)$ charged by a type-1 charge, where $e \in E_c^+$ and $u$ is an endpoint of $e$:
  In this case, $\mathsf{charge}(u, e) = 1$ because there is only a single edge $e$ that can make such a charge.

- A pair $(z(w) = u, f(w))$ charged by a type-2 charge, where $w \in N^+(u) \triangle N^+(v)$:
  For any such charge, we increase $\mathsf{charge}(u, f(w))$ by

  $$|\mathsf{ChargeSet}(e)|^{-1} \leq (\varepsilon \cdot \deg^+(u))^{-1}.$$

  At the same time, such a charge can only be made by edges from $u$ to $v \in N^+(u)$ (so that $(u, v) \in E^+$), which are $\deg^+(u)$ many. Thus, the total charge made in this case leads to $\mathsf{charge}(u, f(w)) = \frac{1}{\varepsilon}$.

- A pair $(v, f(w))$ charged by a type-2 charge, where $w \in N^+(u) \triangle N^+(v)$ and $z(w) = v$:

  For any such charge, we increase $\mathsf{charge}(v, f(w))$ by

  $$|\mathrm{ChargeSet}(e)|^{-1} \leq (\varepsilon \cdot \deg^+(v))^{-1}.$$

  At the same time, such a charge can only be made by edges from $v$ to $u \in N^+(v)$ (so that $(u, v) \in E^+$), which are $\deg^+(v)$ many. Thus, the total charge made in this case leads to $\mathsf{charge}(v, f(w)) = \frac{1}{\varepsilon}$.

Therefore, by Lemma A.2, the total cost of all $\varepsilon$-sparse edges is at most $\frac{2}{\varepsilon} \cdot \mathsf{OPT}$.  $\square$

Note that the size $\varepsilon$-sparse $(+)$ edges is exactly $m^+_{\beta\varepsilon\text{-sparse}}$, which gives the desired property of the first statement of Lemma 3.15.

**Part II: Bounding the charge on $\widehat{E}^-_{\beta\varepsilon\text{-dense}}$ and the value of $\widehat{m}^-_{\beta\varepsilon\text{-dense}}$**

We now turn to the upper bound of $\widehat{m}^-_{\beta\varepsilon\text{-dense}}$, which we prove by showing the upper bound of all $E''$ constructed with a given $\beta$. In particular, we show that,

**Lemma A.4.** *Let $E''$ be obtained by the following process: "add $\deg^+(v)$ non-edge $e$ that are at most $\beta\varepsilon$-dense to $E''$ for each vertex $v$ (or add all such non-edges if the total number is less than $\deg^+(v)$)" (as in Observation 3.13). For any such $E''$, there is*

$$|E''| \leq 4 \cdot \mathsf{OPT}.$$

*Proof.* Fix any such $E''$, let us assign a cost of 1 for each edge $e \in E''$. We show that there exist sets $\mathrm{ChargeSet}(e)$ for every $\beta\varepsilon$-dense $(-)$ edge $e \in E''$, such that for all vertex-edge pairs $(z, f)$, there is

$$\mathsf{dense\text{-}charge}(z, f) = 2.$$

To see this, define $\mathrm{ChargeSet}(e)$ for any $(u, v) = e$ as follows:

- Type-1 charges: when $\mathsf{cost}(\mathsf{OPT}, e) = 1$. In this case, we simply set $\mathrm{ChargeSet}(e) = \{(u, e)\}$ itself.

- Type-2 charges: when $\mathsf{cost}(\mathsf{OPT}, e) = 0$. Note that in this case, we have that $\mathcal{O}(u) \neq \mathcal{O}(v)$ under the optimal clustering $\mathcal{O}$, i.e., the optimal cluster $\mathcal{O}$ does *not* put $u$ and $v$ to the same cluster.

  Consider any vertex $w \in N^+(u) \cap N^+(v)$, if $\mathcal{O}(w) = \mathcal{O}(u)$, then $\mathsf{cost}(\mathsf{OPT}, (w, v)) = 1$; otherwise, if $\mathcal{O}(v) = \mathcal{O}(u)$, then $\mathsf{cost}(\mathsf{OPT}, (w, u)) = 1$. That is to say, there is exactly one edge $f(w) \in \{(w, u), (w, v)\}$ such that $\mathsf{cost}(\mathsf{OPT}, f(w)) = 1$. Let $z(w) \in \{u, v\}$ be the vertex other than $w$ incident on $f(w)$. We add all pairs $(z(w), f(w))$ to $\mathrm{ChargeSet}(e)$, i.e.,

  $$\mathrm{ChargeSet}(e) = \left\{ (z(w), f(w)) \mid w \in N^+(u) \cap N^+(v) \right\}.$$

We now show that the size of the $\mathrm{ChargeSet}(e)$ is large. Note that by definition, every $(-)$ edge $(u, v)$ in the set $E''$ is at most $\beta\varepsilon$-dense. Hence, there is

$$\left| N^+(v) \triangle N^+(u) \right| \leq \beta\varepsilon \cdot \max \left\{ \deg^+(u), \deg^+(v) \right\}.$$

Therefore, we have that

$$
\begin{aligned}
|\text{ChargeSet}(e)| &:= \left| N^+(u) \cap N^+(v) \right| \\
&= \deg^+(u) + \deg^+(v) - \left| N^+(v) \triangle N^+(u) \right| \\
&\geq (1 - \beta \varepsilon) \cdot \max \left\{ \deg^+(u), \deg^+(v) \right\} \\
&\geq \frac{1}{2} \cdot \max \left\{ \deg^+(u), \deg^+(v) \right\}. \qquad \text{(as long as } \beta \leq \tfrac{1}{2\varepsilon})
\end{aligned}
$$

As a result, we can bound the distributed charges in the same manner of the proof of Lemma A.3. We have three different choices for $(z, f)$ that can belong to ChargeSet$(e)$ for some edge $e \in E''$ as follows:

- A pair $(u, e)$ charged by a type-1 charge, where $e \in E''$ and $u$ is an endpoint of $e$:

  In this case, $\mathsf{charge}(u, e) = 1$ because there is only a single $(-)$ edge $e$ that can make such a charge.

- A pair $(z(w) = u, f(w))$ charged by a type-2 charge, where $w \in N^+(u) \cap N^+(v)$:

  For any such charge, we increase $\mathsf{charge}(u, f(w))$ by

  $$
  |\text{ChargeSet}(e)|^{-1} \leq (\frac{1}{2} \cdot \deg^+(u))^{-1}.
  $$

  At the same time, such a charge can only be made by edges from $u$ to $v \in N^-(u)$ (so that $(u, v) \in E^-$), and there are at most $\deg^+(u)$ number of edges ever counted in $E''$. Thus, the total charge made in this case leads to $\mathsf{charge}(u, f(w)) = 2$.

- A pair $(v, f(w))$ charged by a type-2 charge, where $w \in N^+(u) \cap N^+(v)$ and $z(w) = v$:

  For any such charge, we increase $\mathsf{charge}(v, f(w))$ by

  $$
  |\text{ChargeSet}(e)|^{-1} \leq ((\frac{1}{2} \cdot \deg^+(v))^{-1}.
  $$

  At the same time, such a charge can only be made by edges from $v$ to $u \in N^-(v)$ (so that $(u, v) \in E^-$), and there are at most $\deg^+(v)$ number of edges ever counted in $E''$. Thus, the total charge made in this case leads to $\mathsf{charge}(v, f(w)) = 2$.

Therefore, by Lemma A.2, the total cost of all edges in $E''$ is at most $4 \cdot \mathsf{OPT}$.

$\square$

Finally, by Observation 3.13, we have $2 \left| E'' \right| \geq \widehat{m}^-_{\beta \varepsilon\text{-dense}}$, which gives us $\widehat{m}^-_{\beta \varepsilon\text{-dense}} \leq 8 \cdot \mathsf{OPT}$, as desired.