# OpenReview forum: "Streaming Algorithms and Lower Bounds for Estimating Correlation Clustering Cost"
_NeurIPS.cc/2023/Conference — NeurIPS 2023 poster_

### Official Review · Reviewer_E8E2 · 2023-06-28

**Soundness:** 4 excellent
**Presentation:** 3 good
**Contribution:** 3 good
**Rating:** 7
**Confidence:** 3

**Summary:**

The paper studies correlation clustering in the streaming model where the edges of the underlying graph are updated one  at a time. Where previous work has focused on the semi streaming model with $\Omega(n)$ space, they consider more classic streaming with a polylogarithmic number of bits. Here it is not possible to describe the full clustering, so the goal is to estimate the \emph{cost} of the correlation clustering. They consider $(a,b)$-approximation algorithms where if the optimal cost is $OPT$, the algorithm provides an estimate of at least $OPT$ but at most $a\cdot OPT+b$. Their first result is a single-pass streaming algorihtm that is an $(O(1),\delta n^2)$-approximation using $polylog (n)/\delta^5$ space. Here the constant in the $O$-notation is rather large. However, they also present an algorithm which is an $(3,\delta n^2)$-approximation in expectation using $2^{O(1/\delta)} polylog(n)$ space. The first of these results is inspired by the sparse-dense decomposition [AW'22]. At a very high level, using sample based approached, they estimate the number of so called $\varepsilon$-sparse and $\varepsilon$-dense edges and relate these to the optimal clustering cost. This can be done for vertices of degree at least $\delta n$ and the vertices with smaller degrees can contribute at most $\delta n^2$ to the total cost (this is where the additive term comes from). This is by no means trivial and requires substantial technical work. For the second algorithm, they simulate an algorithm from an unpublished manuscript [BGK13].

The paper also provides lower bounds showing that an $(1,n^{2-\varepsilon})$-approximation requires $n^\varepsilon$ space (this is via a reduction from the INDEX problem) and that an $(1.19,O(n))$-approximation requires $\Omega(\sqrt{n})$ space.

The paper also provides experiments demonstrating the performance of their algorithms on the stochastic block model (SBM) and for Erdos-Renyi random graphs (ER). It is of interest that their algorithms are able to distinguish the clusterable setting in SBM from that the ER $G(n,p)$ which are not clusterable at all.

**Strengths:**

It is in interesting problem to study the correlation clustering problem in the low space regime both from a theoretical and practical perspective. The paper is well written and the proofs that I looked at in the supplementary material also seem to be clearly written (I only looked at a few of them so I cannot vouch for correctness, however I gave a 4 for soundness based on the ones I checked). The technical contribution of the paper seems impressive, in particular since they also provide lower bounds for the problem.

**Weaknesses:**

It is a little unclear which ideas are novel and which ideas are inspired from previous work e.g., [AW'22].

**Questions:**

l85: This seems incorrect. To get sublinear space, you put $\delta=n^{-0.19}$ and then the additive error is $n^{1.81}$

l104-110: This section is a little clumsily written and could use a rewriting to make it clearer

l119: What does $0.04\% \sim 3.6\%$ mean?

l276-277: $S_i$ and $m_u$ have not been defined

l347-348: This seems unsurprising. Could it be proved by some union bound approach?

**Limitations:**

The authors have adequately addressed limitations.

---

> ### Author Rebuttal · Authors · 2023-08-09
>
> Thank you for your detailed review and your positive feedback and also for catching all our tiny errors. Below are our responses:
>
> **Point1: Which ideas are new and which are existing.**
> Getting approximate clustering in $\tilde{O}(n)$ space via the algorithms (SDD/Pivot) was already known. The novel idea here lies in estimating the costs of these algorithms within polylog(n) space only (especially since it requires exponentially larger $\Omega(n)$ memory to directly run SDD/Pivot).
>
> The main idea for the SDD algorithm was that we could get an $O(1)$ approximation to the cost using epsilon sparse and epsilon dense edges. We then built the estimators to estimate the number of these edges with an additive error.
>
> The main idea for the pivot algorithm was that we could do the greedy MIS (to find the pivots) after the stream. Then we built estimators to estimate the cost of the clusters.
>
> **Point 2: Questions/MISC comments.**
>
> L85: Yes, you are right we made a calculation error. Thanks for the catch! What we meant to say was setting delta to $n^{0.19}$ would make the space $o(n)$ and the error $o(n^2)$. We will fix this.
>
> L104-110: We agree that this paragraph is not very well written. We will rewrite it in the final version.
>
> L119: What we mean here is the following: The fraction of total edges stored by the Pivot-based algorithm is 0.04%.
> The fraction of total edges stored by the SDD-based algorithm is 3.6%.
>
> L276-277: $S_i$ and $m_u$ are defined in the full version (supplementary material), but when we were creating the conference version, it appears there are some nuaunce ordering issues (e.g., $S_i$ is defined in the algorithm box, which appears before claim 4.2; but in the conference version it appears later). We will fix this in the final version.
>
> L347-348: We can prove that an Erdos-Renyi random graph with p=0.5 has an optimal clustering cost of $\Omega(n^2)$ using charging arguments similar to [AW22]. We are not familiar with the ``union bound approach’’ mentioned in the review – we will be happy to learn more from you about this.

---

### Official Review · Reviewer_nMDo · 2023-07-06

**Soundness:** 3 good
**Presentation:** 4 excellent
**Contribution:** 4 excellent
**Rating:** 7
**Confidence:** 3

**Summary:**

The paper gives polylog space streaming algorithm for approximately computing the value of the cost of correlation clustering. An algorithm for finding the (approximate) correlation clustering in polylog space is known to be not possible. It is also known that approximately computing the cost within a multiplicative factor is not possible in polylog space. This paper designs polylog space streaming algorithm for approximating the cost within a constant multiplicative plus an O(n^2) additive factor. They show that such an additive factor is necessary. The techniques used are sparse-dense decomposition of [AW22] and local correlation clustering of [BGK13].

**Strengths:**

1. The theoretical results are novel, well-motivated, and consistent with the known results. All the bounds are well justified. There aren't significant gaps in the presented results.
2. The paper is well written. The results are well-motivated, related work is clearly stated, and the main ideas are clearly outlined in the allowed space.
3. Experimental results show the capability of the algorithm to distinguish between high and low clusterable instances of random graphs generated in the stochastic block model.

**Weaknesses:**

1. Using some space to motivate the practical aspects of cost estimation (instead of computing the solution) will be useful for the readers. In the current version, this aspect of the motivation is pointed to the previous works.
2. The experimental section also does not help with point (1) since it separates random graphs based on clusterability. A discussion on a real world example may help.


**Questions:**

- Some of the other queries are mentioned above within the other fields.

**Limitations:**

This work is mostly theoretical. There are no negative societal impacts.

---

> ### Author Rebuttal · Authors · 2023-08-09
>
> Thank you for your detailed review and your positive feedback. Our responses are as follows.
>
> **Point 1: Practical motivation.** Thanks for pointing this out, and we will add the discussion of the motivations in our later version. The motivation can be described roughly as follows. In large-scale applications, sometimes we want to know if the instances are worth clustering or not as resources might be limited; or rather, how much better it would be if we do clustering (trying to understand ‘clusterability’ of the instances). If we can test the clustering cost with only $\text{polylog} \ {n}$ space, then such a task can be accomplished in a local machine (as opposed to using a large cluster as in semi-streaming).
> We could also use the value to differentiate between different classes of graphs that have different clustering costs.
>
> We also agree that adding real-world examples will help, and we intend to make changes in the final version.
> In our current paper, we believe that separating random graphs with SBMs *is already* a toy example for real-world applications. Consider two types of social networks: one is more structured (like SBM); the other is more chaotic (random graph-like). Our algorithm allows the users to classify different types of social networks in this regard.

---

> > ### Comment · Reviewer_nMDo · 2023-08-11
> >
> > This is to acknowledge that I have read author rebuttal. I do not have any other questions.

---

### Official Review · Reviewer_DYBZ · 2023-07-06

**Soundness:** 3 good
**Presentation:** 4 excellent
**Contribution:** 4 excellent
**Rating:** 8
**Confidence:** 4

**Summary:**

The authors initiate a study of the correlation clustering problem in streams, in the setting when only the *cost* of the clustering needs to be output. Prior work on correlation clustering in streams focused on the "semi-streaming" model, in which the clustering must be output but a space of $\tilde\Theta(n)$ is allowed. On the other hand, the authors obtain algorithms using space $o(n)$ and even $\mathrm{poly}\log(n)$ when only the cost but not the clustering itself is required. The problem of computing the cost is motivated as a measure of the "clusterability" cost, and can be used to determine whether it makes sense to compute a clustering or not.

The authors provide both upper bounds and lower bounds for this problem. For upper bounds, the authors show that in $\mathrm{poly}\log(n)$ bits of space, one can obtain a mixed additive-multiplicative error guarantee, with $O(1)$ multiplicative error and $o(n^2)$ additive error. The authors also give show that a purely relative error guarantee cannot be obtained in $\mathrm{poly}\log(n)$ space by showing that an approximation with $c = O(1)$ multiplicative error and $\epsilon n$ additive error requires $\Omega(\sqrt n)$ space, for some choices of constant $c$ and $\epsilon$. Another lower bound that the authors give show that an algorithm achieving a purely additive $n^{2-\epsilon}$ error requires $\Omega(n^\epsilon)$ space. Thus, one cannot hope for too much better than the algorithms given by the authors in general. Furthermore, the authors argue that the $o(n^2)$ additive error is in fact highly useful in practice. Indeed, for a typical application of detecting "clustered" communities in a stochastic block model where edges within communities are connected with probability p > 0.5 (say 0.8) and edges between communities with probability p < 0.5 (say 0.2), the $o(n^2)$ additive error algorithm suffices. The practical uses of the algorithm are verified empirically.

In terms of techniques, the authors give two upper bounds, one based on the classic Pivot algorithm and another based on sparse-dense decompositions developed in prior work. In both cases, the authors carefully implement the algorithms in the streaming setting by running the algorithms only on "heavy hitter" vertices and by gathering necessary statistics using sketching techniques.

**Strengths:**

This work provides both a novel streaming problem with many interesting open questions and nice practical motivations, as well as an interesting and highly nontrivial upper bound. The gap between the upper bounds and lower bounds in this work are highly intriguing, and I would expect this work to stimulate many interesting follow-up works to tighten the results. Thus, the problem that is introduced is extremely interesting. The algorithmic techniques used in the upper bound seem to be a combination of prior work (known algorithms for correlation clustering combined with heavy hitters), but making this go through requires a lot of work in the analysis and is highly nontrivial.

**Weaknesses:**

Just a couple of comments on the empirical results (however this is not that significant since the contribution of this work is primarily theoretical):
* The axes in Figure 1 (and all the figures in the appendix in the supplementary material) are illegible. Please take advantage of the space availability in the supplementary material to provide better plots.
* If $n = 1000, 2000$, then a semi-streaming algorithm would hypothetically use only roughly n/n^2 ~ 0.1% of the edges, so 4% or 10% of the edges does not seem that impressive in comparison at first glance. Are there any comparisons to semi-streaming algorithms in terms of empirical performance?

**Questions:**

See weaknesses.

**Limitations:**

The authors have thoroughly discussed the limitations (e.g. the additive error) and justified it (e.g. lower bounds).

---

> ### Author Rebuttal · Authors · 2023-08-09
>
> Thank you so much for your detailed review and your positive feedback. Here are our responses to the questions:
>
> **Point 1: Problems with the axes in the Figures.** Thanks for spotting out the issue, and we will fix the figures by making the text on the axes larger.
>
> **Point 2: Implementation of semi-streaming and the percentage of edges used.** We are not aware of any prior implementations of streaming correlation clustering algorithms. It appears that there are quite some barriers to implementing a complete large-scale semi-streaming correlation clustering system (or even large-scale semi-streaming in general) – some aspects that are assumed to not be problems in theory, e.g., polylog(n) overheads, I/O speed, and the need of external memory, become quite problematic in practical systems.  Take the connected components sketch for an example: the algorithm was known since Ahn et al. [SODA’12], yet the first implementation was only given until quite recently by Tench et al., [SIGMOD’22] (`GraphZeppelin’). The practical implementation of correlation clustering in semi-streaming space seems to be a very interesting problem on its own.
>
> With regard to the percentage of edges saved, we suspect that the leading constant makes the percentage of stored edges rather high when $n$ is not too large (n=1000 or n=2000). Also, the log factors may become large in practice: for instance, when $n=2000$, $n/n^2=0.05$%, but $n \log^{3}n / n^2=65.9$% (log with base 2, a typical semi-streaming bound). These factors may have contributed to the discrepancy between the theoretical bound and the practical performances and would affect semi-streaming algorithms too.

---

### Official Review · Reviewer_YPdG · 2023-07-06

**Soundness:** 2 fair
**Presentation:** 4 excellent
**Contribution:** 1 poor
**Rating:** 3
**Confidence:** 3

**Summary:**

This paper studies the correlation clustering problem in the streaming setting. Unlike previous work they consider algorithms with space much smaller than the number of vertices and only find the cost of the optimal clustering, not the clustering itself. They define an (alpha, beta) approximation to be an additive error of alpha*OPT + beta. Their "result 1" and "result 2" are for any constant delta > 0 streaming algorithms with approximations of (O(1), delta n^2) and (3, delta n^2) respectively and space O~(1/delta^5) and O~(2^O(1/delta)) respectively. Their "result 3" and "result 4" are impossibility results. They also give some experimental results.


**Strengths:**

The paper is well written.

The impossibility results ("result 3" and "result 4") look interesting, though I haven't had time to read the proofs.

**Weaknesses:**

The first and biggest weakness of this paper is the "result 1" and "result 2" in the intro can straightforwardly be beaten using known techniques. In particular one can get a (1, delta n^2) approximation in poly(delta) space and time poly(n) + 2^poly(1/delta). To do this simply take a random sample of k = Theta~(1/delta^4) vertices, solve correlation clustering on the sample with an algorithm with delta k^2 additive error, and then scale up the cost by a factor of (n/k)^2. (If one-sided error is desired as in this paper's definitions you also need to subtract roughly delta n^2 from the estimate.) The key idea is the fact that to get delta n^2 additive error it suffices to consider clusterings with O(1/delta) clusters, which makes correlation clustering an instance of the well-studied MAX-CSP family. A sample size of k being enough to approximate the value of MAX CSPs to an additive error first appeared in a paper by Arora, Frieze, K and Karpinski (AFKK) in the 1990s IIRC. (I'll try to track down the exact citation and some more details in a comment to be added this weekend.) For the approximation algorithm one can either use that general-purpose AFKK algorithm or one of the published PTASs for correlation clustering with a constant number of clusters, e.g. [8] in https://en.wikipedia.org/w/index.php?title=Correlation_clustering&oldid=1147372897 (which includes this problem despite the title not mentioning it). The algorithms I mentioned aren't very practical so for experiments you could use applying a branch and bound solver such as SciP, CPLEX, or Gurobi to the correlation clustering integer program for the sample only, which should scale to a sample size of something like 50 vertices, or use any algorithm for correlation clustering (but with worse performance guarantee). To do the random sampling in a streaming fashion keep track of the k vertices with the smallest hashes seen so far and any edges between them.

Another big weakness of this paper is that correlation clustering instances large enough to not fit on one machine and hence be interesting for streaming algorithms are likely to be sparse, i.e. have o(delta n^2) positive edges. In practice one can therefore probably do better than their result 1 or result 2 (or the algorithm I sketched above) by simply returning the number of positive weight edges, which is a trivial upper-bound on OPT (it's the cost of the all-singletons clustering). I can't rule out the existence of applications where this sort of guarantee would be useful, I'm just not aware of any.

Another weakness of this paper is streaming models are only useful for a somewhat narrow circumstance: instances are small enough that you have enough time to pass all the data through one machine but large enough that you don't have space to store the full data on one machine. Many streaming results also work in a distributed model such as MPC, which is more practically relevant in my experience, but the authors don't discuss distributed models so I can't tell without reading everything in detail.


**Questions:**

Any comments on the weaknesses mentioned in that section?

=== Misc comments not really questions but I'm not sure where else to put them =====

Abstract: replace "improves the additive error further down" with "further reduces the additive error"

Correlation clustering appeared (but by a different name) in a 1989 paper by GROTSCHEL and WAKABAYASI titled "A CUTTING PLANE ALGORITHM FOR A CLUSTERING PROBLEM"
(https://citeseerx.ist.psu.edu/viewdoc/download?doi=10.1.1.468.4826&rep=rep1&type=pdf). I'm guessing there are probably even earlier papers somewhere. So the 2004 paper you cite as introducing the correlation clustering problem actually reintroduced it.

cost(C) is defined twice, first in section 2.1 and then in section 2.2. Also the definition in section 2.1 is incomplete since it doesn't say whether the cost is the number of disagreements or the number of agreements or something else, though the word "cost" does hint that it's probably the number of disagreements.

Using the shape of the brackets to distinguish N^-[] from N^-() and N^+[] from N^+() is a bit weird since when bracket shape is used to distinguish notations the two notations are usually totally unrelated, not variations of the same thing. I would prefer a more traditional notation, e.g. overbar on the N for the version that includes the vertex itself.


**Limitations:**

See weaknesses section for some limitations. No societal impact issues.

---

> ### Author Rebuttal · Authors · 2023-08-09
>
> Thanks for your detailed review and the critical new idea. Please find our responses below.
>
> **Point 1: The suggested $(1, \delta n^2)$-approximation algorithm.** Thank you for raising this observation. If we understand it correctly, the key idea of your suggestion is to reduce the min-disagreement with $\delta \cdot n^2$ additive error to max-agreement correlation clustering with $\delta \cdot n^2$ additive error. Also, it is okay to work with just $1/\delta$ clusters when the additive error is $\delta n^2$. Finally, use some sparsification results to solve the $(1/\delta)$-cluster version of CSP on a small sample. This is a very nice approach that we did not consider previously, and it does indeed seem to us also that an approach along these lines might indeed be plausible.
>
> That being said, we did not find this approach trivial/straightforward, nor could we even be fully convinced at this stage yet that it works without complete proof. If this algorithm was known previously, we have missed it and would appreciate a reference to this result. However, we would like to point out that approaches like this have been tried for (max-agreement) correlation clustering, say paper ``Sublinear Algorithms for MAXCUT and Correlation Clustering’’ by Bhaskara et al. [ICALP’18], and they appear to require substantially more work to address the subtle issues raising in this context, than being a straightforward application of known tools (these tools themselves are also not exactly simple to begin with). In addition, as your review also points out, such an algorithm would require using highly sophisticated “post-processing” algorithms that make the runtime of the algorithm quite high (we suspect at the very least $n^{1/\delta}$-type bound), and the implementation in practice infeasible (although from a purely theoretical result, such a bound fully complements our lower bounds and is obviously of its own interest surely). This is in contrast to our approach which offers practical streaming algorithms with quite straightforward implementations. Thus, we do not see your suggestion as rendering our results trivial and we stand by the contribution of our paper as providing the first truly-sublinear space streaming algorithms for this fundamental problem.
>
> We will be happy to try out the suggested algorithm idea in detail and write a proof in a future version; alternatively, we can also leave the idea since it is not originally from us. We will be happy to hear your opinion on this front.
>
> **Point 2: Concerns about the model.** We were actually rather surprised by the comments about the perceived impracticality of streaming algorithms in general. While we do understand that certain companies, say, Google, may indeed have a strong preference for MapReduce-style computation frameworks, this is certainly not across the board and is quite application-oriented, as is evidenced by the vast amount of work on various streaming platforms. Indeed, streaming algorithms are not designed for processing 20-TB size graphs, but they can target 1-TB size graphs quite easily by making one sequential pass over them very quickly and using RAM for the much smaller memory needed for the algorithm, which gives tremendous speed-up compared to directly running the algorithm on a 1-TB size graph. We refer the reviewer to the papers ''GraphZeppelin: Storage-Friendly Sketching for Connected Components on Dynamic Graph Streams’’ by Tench et al. and ''Practice of Streaming Processing of Dynamic Graphs: Concepts, Models, and Systems’’ by Besta et al. for instances.
>
> That being said, our results **immediately** extend to these other models as well in the same standard way that prior streaming algorithms do.  Since both of our algorithms are based on *linear sketches*, they naturally imply fully scalable MPC algorithms with constant number of rounds – we only need an MPC with $O(n^{c}\cdot \text{polylog} \ {n/\delta})$ space per machine and the number of round is $O(1/c)$. The algorithm has to simply run the linear sketch locally, and broadcast with the standard communication tree, and correctness is guaranteed by the properties of linear sketches. In fact, it appears both of our algorithms can be executed in the stronger PRAM model with $O(1)$ depth and $\text{polylog}\ {n}$ work – we can check the details on whether it works in future versions.
>
> **Point 3: Concerns about the correlation clustering instances being sparse.** We believe there are many streaming applications where large graphs have $\Theta(n^2)$ edges. One toy example is the stochastic block model (SBM) we studied in the experiment section, in which the instances have $\Omega(n^2)$ the intra-block edges alone (which do not even contribute to the cost). The SBM has been studied extensively in the literature, and it captures a wide range of social network applications, e.g., community detection.
>
> The SBM is also a prime example to show that simply returning the number of edges gives a bad approximation for the correlation clustering cost. Note when $p$ is large in the SBM, most edges do not contribute to the cost, and simply counting the number of edges would give a terrible approximation (both in theory and in practice). One can take another example: for Erdos-Renyi graphs with $p=0.5$ and an instance from SBM with $p=0.95$, the numbers of positive edges are roughly the same, but the costs are very different.
>
> **MISC comments.** We appreciate the reviewer’s effort to catch the MISC issues. We will fix them accordingly in future versions of the paper.

---

### Decision · Program_Chairs · 2023-09-21

**Decision:**

Accept (poster)

**Comment:**

Most reviewers agreed that the problem studied is well-motivated and the technical results are solid, except one reviewer who commented that "Result 1" and "Result 2" of this paper can "straightforwardly be beaten using known techniques". The authors addressed this criticism; however, we have not received any further response from that reviewer. Based on my examination of the authors' response, it seems that the criticism cannot be adequately justified without working out the actual proofs.